# Churn Reduction via Distillation

**Heinrich Jiang, Harikrishna Narasimhan, Dara Bahri, Andrew Cotter, Afshin Rostamizadeh**
Google Research
`{heinrichj, hnarasimhan, dbahri, acotter, rostami}@google.com`

## Abstract

In real-world systems, models are frequently updated as more data becomes available, and in addition to achieving high accuracy, the goal is to also maintain a low difference in predictions compared to the base model (i.e. predictive "churn"). If model retraining results in vastly different behavior, then it could cause negative effects in downstream systems, especially if this churn can be avoided with limited impact on model accuracy. In this paper, we show an equivalence between training with distillation using the base model as the teacher and training with an explicit constraint on the predictive churn. We then show that distillation performs strongly for low churn training against a number of recent baselines on a wide range of datasets and model architectures, including fully-connected networks, convolutional networks, and transformers.

## 1 Introduction

Deep neural networks (DNNs) have had profound success at solving some of the most challenging machine learning problems. While much of the focus has been spent towards attaining state-of-art predictive performance, comparatively there has been little effort towards improving other aspects. One such important practical aspect is reducing unnecessary predictive churn with respect to a base model. We define predictive churn as the difference in the prediction of a model relative to a base model on the same datapoints. In a production system, models are often continuously released through an iterative improvement process which cycles through launching a model, collecting additional data and researching ways to improve the current model, and proposing a candidate model to replace the current version of the model serving in production. In order to validate a candidate model, it often needs to be compared to the production model through live A/B tests (it's known that offline performance alone isn't a sufficient, especially if these models are used as part of a larger system where the offline and online metrics may not perfectly align (Deng et al., 2013; Beel et al., 2013)). Live experiments are costly: they often require human evaluations when the candidate and production model disagree to know which model was correct (Theocharous et al., 2015; Deng & Shi, 2016). Therefore, minimizing the unnecessary predictive churn can have a significant impact to the cost of the launch cycle.

It's been observed that training DNN can be very noisy due to a variety of factors including random initialization (Glorot & Bengio, 2010), mini-batch ordering (Loshchilov & Hutter, 2015), data augmentation and processing (Santurkar et al., 2018; Shorten & Khoshgoftaar, 2019), and hardware (Turner & Nowotny, 2015; Bhojanapalli et al., 2021)– in other words running the same procedure multiple times can lead to models with surprisingly amount of disagreeing predictions even though all can have very high accuracies (Bahri & Jiang, 2021). While the stability of the training procedure is a separate problem from lowering predictive churn, such instability can further exacerbate the issue and underscores the difficulty of the problem.

Knowledge distillation (Hinton et al., 2015), which involves having a *teacher* model and mixing its predictions with the original labels has proved to be a useful tool in deep learning. In this paper, we show that this surprisingly is not only an effective tool for churn reduction by using the base model as the teacher, it is also mathematically aligned with learning under a constraint on the churn. Thus, in addition to providing a strong method for low churn training, we also provide insight into the distillation.

Our contributions are as follows:

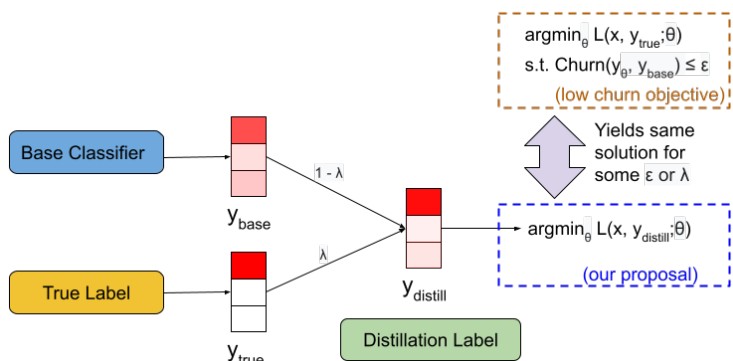

Figure 1: **Illustration of our proposal**: We propose using knowledge distillation with the base model as the teacher (with some mixing parameter $\lambda$) and then training on the distilled label. We show theoretically that training our loss with the distilled label yields approximately the same solution as the original churn constrained optimization problem for some slack $\epsilon$ that depends on $\lambda$ (or vice versa). The significance is that the simple and popular distillation procedure yields the same solution as the original churn problem, without having to deal with the additional complexity that comes with solving constrained optimization problems.

- We show theoretically an equivalence between the low churn training objective (i.e. minimize a loss function subject to a churn constraint on the base model) and using knowledge distillation with the base model as the teacher.
- We show that distillation performs strongly in a wide range of experiments against a number baselines that have been considered for churn reduction.
- Our distillation approach is similar to a previous method called "anchor" (Fard et al., 2016), which trains on the true labels instead of distilled labels for the incorrectly predicted examples by the base model, but outperform this method by a surprising amount. We present both theoretical and experimental results showing that the modification of anchor relative to distillation actually hurts performance.

## 2 RELATED WORKS

**Prediction Churn.** There are few works that address low churn training with respect to a *base model*. Fard et al. (2016) proposed an *anchor loss* which is similar to distillation when the base model's prediction agrees with the original label, and uses a scaled version of the original label otherwise. In our empirical evaluation, we find that this procedure performs considerably worse than distillation. Cotter et al. (2019a); Goh et al. (2016) use constrained optimization by adding a constraint on the churn. We use some of the theoretical insights found in that work to show an equivalence between distillation and the constrained optimization problem. Thus, we are able to bypass the added complexity of using constrained optimization (Cotter et al., 2019b) in favor of distillation, which is a simpler and more robust method.

A related but different notion of churn that has been studied is where the goal is to reduce the training instability. Anil et al. (2018) noted that co-distillation is an effective method. Bahri & Jiang (2021) proposes a locally adaptive variant of label smoothing and Bhojanapalli et al. (2021) propose entropy regularizers and a variant of co-distillation. We tested many of the baselines proposed in these papers and adapt them to our notion of churn and showed that they were not effective at reducing predictive churn w.r.t. a base model.

**Distillation.** Distillation (Ba & Caruana, 2013; Hinton et al., 2015), first proposed to transfer knowledge from larger networks to smaller ones, has become immensely popular. Applications include learning from noisy labels (Li et al., 2017), model compression (Polino et al., 2018), adversarial robustness (Papernot et al., 2016), DNNs with logic rules (Hu et al., 2016), visual relationship detection (Yu et al., 2017), reinforcement learning (Rusu et al., 2015), domain adaptation (Asami et al., 2017) and privacy (Lopez-Paz et al., 2015). Our work adds to the list of applications in which distillation is effective.

The theoretical motivation of distillation however is less established. Lopez-Paz et al. (2015) studied distillation as learning using privileged information (Vapnik & Izmailov, 2015). Phuong & Lampert (2019) establishes fast convergence of the expected risk of a distillation-trained linear classifier. Foster et al. (2019) provides a generalization bound for the student with an assumption that it learns a model close to the teacher. Dong et al. (2019) argued distillation has a similar effect as that of early stopping. Mobahi et al. (2020) showed an equivalence to increasing the regularization strength for kernel methods. Menon et al. (2020) establish a bias-variance trade-off for the student. Our analysis provides a new theoretical perspective of its relationship to churn reduction.

## 3 DISTILLATION FOR CONSTRAINING CHURN

We are interested in a multiclass classification problem with an instance space $\mathcal{X}$ and a label space $[m] = \{1, \ldots, m\}$. Let $D$ denote the underlying data distribution over instances and labels, and $D_{\mathcal{X}}$ denote the corresponding marginal distribution over $\mathcal{X}$. Let $\Delta_m$ denote the $(m-1)$-dimensional simplex with $m$ coordinates. We will use $\mathbf{p} : \mathcal{X} \to \Delta_m$ to denote the underlying conditional-class probabilities, where $p_y(x) = \mathbf{P}(Y = y | X = x)$. We assume that we are provided a base classifier $\mathbf{g} : \mathcal{X} \to \Delta_m$ that predicts a vector of probabilities $\mathbf{g}(x) \in \Delta_m$ for any instance $x$. Our goal is to then learn a new classifier $\mathbf{h} : \mathcal{X} \to \Delta_m$, constraining its churn to be within an acceptable limit.

We will measure the classification performance of a classifier $\mathbf{h}$ using a loss function $\ell : [m] \times \Delta_m \to \mathbb{R}_+$ that maps a label $y \in [m]$ and prediction $\mathbf{h}(x) \in \Delta_m$ to a non-negative number $\ell(y, \mathbf{h}(x))$, and denote the classification risk by $R(\mathbf{h}) := \mathbf{E}_{(x,y) \sim D}[\ell(y, \mathbf{h}(x))]$.

We would ideally like to define predictive churn as the fraction of examples on which $\mathbf{h}$ and $\mathbf{g}$ disagree. For the purpose of designing a tractable algorithm, we will instead work with a softer notion of churn, which evaluates the divergence between their output distributions. To this end, we use a measure of divergence $d : \Delta_m \times \Delta_m \to R_+$, and denote the expected churn between $\mathbf{h}$ and $\mathbf{g}$ by $C(\mathbf{h}) := \mathbf{E}_{x \sim D_{\mathcal{X}}}[d(\mathbf{g}(x), \mathbf{h}(x))]$.

We then seek to minimize the classification risk for $\mathbf{h}$, subject to the expected churn being within an allowed limit $\epsilon > 0$:

$$\min_{\mathbf{h}: \mathcal{X} \to \Delta_m} R(\mathbf{h}) \quad \text{s.t.} \quad C(\mathbf{h}) \leq \epsilon. \tag{1}$$

We consider loss and divergence functions that are defined in terms of a scoring function $\phi : \Delta_m \to \mathbb{R}_+^m$ that maps a distribution to a $m$-dimensional score. Specifically, we will consider scoring functions $\phi$ that are *strictly proper* (Gneiting & Raftery, 2007; Williamson et al., 2016), i.e. for which, given any distribution $\mathbf{u} \in \Delta_m$, the conditional risk $\mathbf{E}_{y \sim \mathbf{u}}[\phi_y(\mathbf{v})]$ is uniquely minimized by $\mathbf{v} = \mathbf{u}$. The following are general forms of the loss and divergence functions employed in this paper:

$$\ell_\phi(y, \mathbf{v}) := \phi_y(\mathbf{v}); \quad d_\phi(\mathbf{u}, \mathbf{v}) := \sum_{y \in [m]} u_y(\phi_y(\mathbf{v}) - \phi_y(\mathbf{u})). \tag{2}$$

The cross-entropy loss and KL-divergence are a special case of this formulation when $\phi_y(\mathbf{v}) = -\log(v_y)$, and the squared loss and the squared $L_2$ distance can be recovered by setting $\phi_y(\mathbf{v}) = \sum_{i \in [m]} (\mathbf{1}(i = y) - v_i)^2$.

### 3.1 BAYES-OPTIMAL CLASSIFIER

We show below that for the loss and divergence functions defined in (2), the optimal-feasible classifier for the constrained problem in (1) is a convex combination of the class probability function $\mathbf{p}$ and the base classifier $\mathbf{g}$.

**Proposition 1.** *Let $(\ell, d)$ be defined as in (2) for a strictly proper scoring function $\phi$. Suppose $\phi(\mathbf{u})$ is strictly convex in $\mathbf{u}$. Then there exists $\lambda^* \in [0, 1]$ such that the following is an optimal-feasible classifier for (1):*

$$\mathbf{h}^*(x) = \lambda^* \mathbf{p}(x) + (1 - \lambda^*)\mathbf{g}(x).$$

*Furthermore, if $\mathbf{u} \cdot \phi(\mathbf{u})$ is $\alpha$-strongly concave over $\mathbf{u} \in \Delta_m$ w.r.t. the $L_q$-norm, then $\lambda^* \leq \sqrt{2\epsilon / \left( \alpha \mathbf{E}_x \left[ \|\mathbf{p}(x) - \mathbf{g}(x)\|_q^2 \right] \right)}$.*

---

**Algorithm 1** Distillation-based Churn Reduction

---

1: **Inputs:** Training sample $S = \{(x_1, y_1), \ldots, (x_n, y_n)\}$, Grid of mixing coefficients $\Lambda = \{\lambda_1, \ldots, \lambda_L\}$, Base classifier $\mathbf{g}$, Constraint slack $\epsilon > 0$
2: Train a classifier $\mathbf{h}_k$ for each $\lambda_k \in \Lambda$ by minimizing the distilled loss in (4):

$$\mathbf{h}_k \in \text{argmin}_{\mathbf{h} \in \mathcal{H}} \, \widehat{\mathcal{L}}_{\lambda_k}(\mathbf{h})$$

3: Find a convex combination of $\mathbf{h}_1, \ldots, \mathbf{h}_L$ by solving following convex program in $L$ variables:

$$\min_{\mathbf{h} \in \text{co}(\mathbf{h}_1, \ldots, \mathbf{h}_L)} \widehat{R}(\mathbf{h}) \ \text{ s.t. } \ \widehat{C}(\mathbf{h}) \leqslant \epsilon$$

and return the solution $\widehat{\mathbf{h}}$

---

The strong concavity condition in Proposition 1 is satisfied by the cross-entropy loss and KL-divergence for $\alpha = 1$ with the $L_1$-norm, and by the squared loss and $L_2$-distance for $\alpha = 2$ with the $L_2$-norm. The bound suggests that the mixing coefficient $\lambda^*$ depends on how close the base classifier is to the class probability function $\mathbf{p}$.

### 3.2 DISTILLATION-BASED APPROACH

Proposition 1 directly motivates the use of a distillation-based approach for solving the churn-constrained optimization problem in (1). We propose treating the base classifier $\mathbf{g}$ as a teacher model, mixing the training labels $y$ with scores from the teacher $\mathbf{g}(x)$, and minimizing a classification loss against the transformed labels:

$$\mathcal{L}_\lambda(h) = \mathbf{E}_{(x,y) \sim D} \left[ (\lambda \mathbf{e}_y + (1 - \lambda)\mathbf{g}(x)) \cdot \phi(\mathbf{h}(x)) \right], \tag{3}$$

where $\mathbf{e}_y \in \{0, 1\}^m$ denotes a one-hot encoding of the label $y \in [m]$ and $\phi$ is a strictly proper scoring function. It is straight-forward to show that when $\lambda = \lambda^*$, the optimal classifier for the above distillation loss takes the same form in Proposition 1, i.e. $\mathbf{h}^*(x) = \lambda^* \mathbf{p}(x) + (1 - \lambda^*)\mathbf{g}(x)$. While the optimal mixing parameter $\lambda^*$ is unknown, we propose treating this as a hyper-parameter and tuning it to reach the desired level of churn.

In practice, we do not have direct access to the distribution $D$ and will need to work with a sample $S = \{(x_1, y_1), \ldots, (x_n, y_n)\}$ drawn from $D$. To this end, we define the empirical risk and the empirical churn as follows:

$$\widehat{R}(\mathbf{h}) = \frac{1}{n} \sum_{i=1}^{n} \ell_\phi(y_i, \mathbf{h}(x_i)); \quad \widehat{C}(\mathbf{h}) = \frac{1}{n} \sum_{i=1}^{n} d_\phi(\mathbf{g}(x_i), \mathbf{h}(x_i)),$$

where $\ell_\phi$ and $d_\phi$ are defined as in (2) for a scoring function $\phi$. Our proposal is to then solve the following empirical risk minimization problem over a hypothesis class $\mathcal{H} \subset \{\mathbf{h} : \mathcal{X} \to \Delta_m\}$ for different values of coefficient $\lambda_k$ chosen from a finite grid $\{\lambda_1, \ldots, \lambda_L\} \subset [0, 1]$:

$$\mathbf{h}_k \in \text{argmin}_{\mathbf{h} \in \mathcal{H}} \, \widehat{\mathcal{L}}_{\lambda_k}(\mathbf{h}) := \frac{1}{n} \sum_{i=1}^{n} (\lambda_k \mathbf{e}_{y_i} + (1 - \lambda_k)\mathbf{g}(x_i)) \cdot \phi(\mathbf{h}(x_i)). \tag{4}$$

To construct the final classifier, we find a convex combination of the $L$ classifiers $\mathbf{h}_1, \ldots, \mathbf{h}_L$ that minimizes $\widehat{R}(\mathbf{h})$ while satisfying the constraint $\widehat{C}(\mathbf{h}) \leqslant \epsilon$, and return an ensemble of the $L$ classifiers. The overall procedure is outlined in Algorithm 1, where we denote the set of convex combinations of classifiers $\mathbf{h}_1, \ldots, \mathbf{h}_L$ by $\text{co}(\mathbf{h}_1, \ldots, \mathbf{h}_L) = \{\mathbf{h} : x \mapsto \sum_{j=1}^{L} \alpha_j \mathbf{h}_j(x) \,|\, \boldsymbol{\alpha} \in \Delta_L\}$.

The post-processing step in Algorithm 1 amounts to solving a simple convex program in $L$ variables. This is needed for technical reasons in our theoretical results, specifically, to translate a solution to a dual-optimal solution to (1) to a primal-feasible solution. In practice, however, we do not construct an ensemble, and instead simply return a single classifier that achieves the least empirical risk while satisfying the churn constraint. In our experiments, we use the cross-entropy loss for training, i.e. set $\phi_y(\mathbf{u}) = -\log(u_y)$.

## 4 THEORETICAL GUARANTEES

We provide optimality and feasibility guarantees for the proposed algorithm and also explain why our approach is better-suited for optimizing accuracy (subject to a churn constraint) compared to the previous churn-reduction method of Fard et al. (2016).

### 4.1 OPTIMALITY AND FEASIBILITY GUARANTEES

We now show that the classifier $\widehat{\mathbf{h}}$ returned by Algorithm 1 approximately satisfies the churn constraint, while achieving a risk close to that of the optimal-feasible classifier in $\mathcal{H}$. This result assumes that we are provided with generalization bounds for the classification risk and churn.

**Theorem 2.** *Let the scoring function $\phi : \Delta_m \to \mathbb{R}_+^m$ be convex, and $\|\phi(\mathbf{z})\|_\infty < B, \forall \mathbf{z} \in \Delta_m$. Let the set of classifiers $\mathcal{H}$ be convex, with the base classifier $\mathbf{g} \in \mathcal{H}$. Suppose $C$ and $R$ enjoy the following generalization bounds: for any $\delta \in (0,1)$, w.p. $\geqslant 1 - \delta$ over draw of $S \sim D^n$, for any $\mathbf{h} \in \mathcal{H}$,*

$$|R(\mathbf{h}) - \widehat{R}(\mathbf{h})| \leqslant \Delta_R(n, \delta); \qquad |C(\mathbf{h}) - \widehat{C}(\mathbf{h})| \leqslant \Delta_C(n, \delta),$$

*for some $\Delta_R(n, \delta)$ and $\Delta_C(n, \delta)$ that is decreasing in $n$ and approaches 0 as $n \to \infty$. Let $\widetilde{\mathbf{h}}$ be an optimal-feasible classifier in $\mathcal{H}$, i.e. $C(\widetilde{\mathbf{h}}) \leqslant \epsilon$ and $R(\widetilde{\mathbf{h}}) \leqslant R(\mathbf{h})$ for all classifiers $\mathbf{h}$ for which $C(\mathbf{h}) \leqslant \epsilon$. Let $\widehat{\mathbf{h}}$ be the classifier returned by Algorithm 1 with $\Lambda = \left\{ \max\{\frac{\epsilon}{\epsilon+2B}, u\} \mid u \in \{\frac{1}{L}, \frac{2}{L}, \ldots, 1\}\right\}$ for some $L \in \mathbb{N}_+$. For any $\delta \in (0,1)$, w.p. $\geqslant 1 - \delta$ over draw of $S \sim D^n$,*

$$\boldsymbol{\mathit{Optimality}} : R(\widehat{\mathbf{h}}) \leqslant R(\widetilde{\mathbf{h}}) + \mathcal{O}\left(\left(1 + \frac{2B}{\epsilon}\right)\left(\Delta_R(n, \delta) + \Delta_C(n, \delta) + \frac{B}{L}\right)\right),$$

$$\boldsymbol{\mathit{Feasibility}} : C(\widehat{\mathbf{h}}) \leqslant \epsilon + \Delta_C(n, \delta).$$

In practice, we expect the churn metric to generalize better than the classification risk, i.e. for $\Delta_C(n, \delta)$ to be smaller than $\Delta_R(n, \delta)$. This is because the classification risk is computed on "hard" labels $y \in [m]$ from the training sample, whereas the churn metric is computed on "soft" labels $\mathbf{g}(\mathbf{x}) \in \Delta_m$ from the base model. The traditional view of distillation (Hinton et al., 2015) suggests that the soft labels from a teacher model come with confidence scores for each example, and thus allow the student to generalize well to unseen new examples. A similar view is also posed by Menon et al. (2020) , who argue that the soft labels from the teacher have "lower variance" than the hard labels from the training sample, and therefore aid in better generalization of the student. To this end, we apply the generalization bound from (Menon et al., 2020, Proposition 2) to the student's churn.

**Proposition 3** (Generalization bound for churn). *Let the scoring function $\phi : \Delta_m \to \mathbb{R}_+^m$ be bounded. For base classifier $\mathbf{g}$, let $\mathcal{U}_\phi \subseteq \mathbb{R}^{\mathcal{X}}$ denote the corresponding class of divergence functions $u(x) = d_\phi(\mathbf{h}(x), \mathbf{g}(x)) = \mathbf{g}(x)^\top(\phi(\mathbf{h}(x)) - \phi(\mathbf{g}(x)))$ induced by classifiers $\mathbf{h} \in \mathcal{H}$. Let $\mathcal{M}_n^C = \mathcal{N}_\infty(\frac{1}{n}, \mathcal{U}_\phi, 2n)$ denote the uniform $L_\infty$ covering number for $\mathcal{U}_\phi$. Fix $\delta \in (0,1)$. Then with probability $\geqslant 1 - \delta$ over draw of $S \sim D^n$, for any $\mathbf{h} \in \mathcal{H}$:*

$$C(\mathbf{h}) \leqslant \widehat{C}(\mathbf{h}) + \mathcal{O}\left(\sqrt{\mathbb{V}_n^C(\mathbf{h})\frac{\log(\mathcal{M}_n^C/\delta)}{n}} + \frac{\log(\mathcal{M}_n^C/\delta)}{n}\right).$$

*where $\mathbb{V}_n^C(\mathbf{h})$ denotes the empirical variance of the divergence values computed on $n$ examples $\{\mathbf{g}(x_i)^\top(\phi(\mathbf{h}(x_i)) - \phi(\mathbf{g}(x_i)))\}_{i=1}^n$; the lower the variance, the tighter is the bound.*

In fact, for certain base classifiers $\mathbf{g}$, generalizing well on "churn" can have the additional benefit of improving classification performance, as shown in Proposition 7 in Appendix B.

### 4.2 ADVANTAGE OVER ANCHOR LOSS

We next compare our distillation loss in (3) with the previous anchor loss of Fard et al. (2016), which uses the base model's prediction only when it agrees with the original label, and uses a scaled version of the original label otherwise. While originally proposed for churn reduction with binary labels, we provide below an analogous version of this loss for a multiclass setup:

$$\mathcal{L}^{\text{anc}}(\mathbf{h}) = \mathbf{E}_{(x,y) \sim D}\left[\mathbf{a} \cdot \phi(\mathbf{h}(x))\right], \tag{5}$$

where

$$\mathbf{a} = \begin{cases} \alpha \mathbf{g}(x) + (1-\alpha)\mathbf{e}_y & \text{if } y = \overline{\text{argmax}}_k \, g_k(x) \\ \eta \mathbf{e}_y & \text{otherwise} \end{cases},$$

for hyper-parameters $\alpha, \eta \in [0,1]$ and a strictly proper scoring function $\phi$. Here, we have used $\overline{\text{argmax}}$ to denote ties being broken in favor of the larger class. While this helps us simplify the exposition, our results can be easily extended to a version of the loss which includes ties.

The anchor loss does not take into account the confidence with which the base model disagrees with the sampled label $y$. For example, if the base model predicts near-equal probabilities for all classes, but happens to assign a slightly higher probability to a class different from $y$, the anchor loss would still completely ignore the base model's score (even though it might be the case that all the labels are indeed equally likely to occur). In some cases, this selective use of the teacher labels can result in a biased objective and may hurt the classifier's accuracy.

To see this, consider an ideal scenario where the base model predicts the true conditional-probabilities $\mathbf{p}(x)$ and the student hypothesis class is universal. In this case, minimizing the churn w.r.t. the base model has the effect of maximizing classification accuracy, i.e. a classifier that has zero churn w.r.t. the base model also produces the least classification error. However, as shown below, even in this ideal setup, minimizing the anchor loss may result in a classifier different from the base model.

**Proposition 4.** *When $\mathbf{g}(x) = \mathbf{p}(x), \forall x$, for any given $\lambda \in [0,1]$, the minimizer for the distillation loss in (3) over all classifiers $h$ is given by:*

$$\mathbf{h}^*(x) = \mathbf{p}(x),$$

*whereas the minimizer of the anchor loss in (5) is given by:*

$$h_j^*(x) = \frac{z_j}{\sum_j z_j} \quad \text{where} \quad z_j = \begin{cases} \alpha p_j^2(x) + (1-\alpha)p_j(x) & \text{if } j = \overline{\text{argmax}}_k \, p_k(x) \\ (\eta + \alpha \max_k p_k(x)) \, p_j(x) & \text{otherwise} \end{cases}.$$

Unless $\alpha = 0$ and $\eta = 1$ (which amounts to completely ignoring the base model) or the base model makes hard predictions on all points, i.e. $p_j(x) \in \{0,1\}, \forall x$, the anchor loss encourages scores that differ from the base model $\mathbf{p}$. For example, when $\alpha = \eta = 1$ (and the base model predicts soft probabilities), the anchor loss has the effect of down weighting the label that the base model is most confident about, and as a result, encourages lower scores on that label and higher scores on all other labels. While one can indeed tweak the two hyper-parameters to reduce the gap between the learned classifier and the base model, our proposal requires only one hyper-parameter $\lambda$, which represents an intuitive trade-off between the one-hot and teacher labels. In fact, irrespective of the choice of $\lambda$, the classifier that minimizes our distillation loss in Proposition 4 mimics the base model $\mathbf{p}$ exactly, and as a result, achieves both zero churn and optimal accuracy.

We shall see in the next section that even on real-world datasets, where the base classifier does not necessarily make predictions close to the true class probabilities (and where the student hypothesis class is not necessarily universal and of limited capacity), our proposal performs substantially better than the anchor loss in minimizing churn at a particular accuracy. Figure 3 provides a further ablation study, effectively interpolating between the anchor and distillation methods, and provides evidence that using the true (hard) label instead of the teacher (soft) label can steadily degrade performance.

## 5 EXPERIMENTS

We now show empirically that distillation is an effective method to train models for both accuracy and low churn. We test our method across a large number of datasets and neural network architectures.

### 5.1 SETUP

**Datasets and architectures**: The following are the datasets we use in our experiments, along with the associated model architectures:

- 12 OpenML datasets using fully-connected neural networks.
- 10 MNIST variants, SVHN, CIFAR10, 40 CelebA tasks using convolutional networks.

| Dataset | cold | warm | s-perturb | mixup | ls | co-dist | anchor | distill |
|---|---|---|---|---|---|---|---|---|
| adult | 6.27 | N/A | 6.05 | 6.57 | N/A | 5.78 | 6.62 | **4.39** |
| bank | 10.04 | 8.43 | 7.8 | 8.25 | 8.89 | 7.55 | 8.77 | **5.58** |
| magic04 | 27.56 | 27.41 | 24.37 | 24.68 | 27.79 | 23.67 | 25.22 | **18.51** |
| phonemes | 10.45 | 10.66 | 10.09 | N/A | 9.02 | 9.3 | 11.14 | **7.4** |
| electricity | 18.16 | 17.53 | 17.23 | 15.69 | 16.19 | 14.94 | 18.22 | **8.99** |
| eeg | 48.02 | 42.96 | 42.04 | 39.98 | 49.98 | 54.99 | 26.99 | **2.0** |
| churn | 27.15 | 25.58 | 22.19 | 20.49 | N/A | 18.71 | 17.59 | **5.51** |
| elevators | 33.34 | 35.87 | 30.41 | 31.47 | 32.91 | 30.53 | 34.38 | **10.44** |
| pollen | 44.03 | N/A | 42.63 | 44.6 | 42.06 | 41.78 | 40.44 | **35.15** |
| phishing | 4.43 | 4.2 | 3.97 | 4.01 | 4.1 | 3.74 | 4.08 | **2.91** |
| wilt | 9.55 | 7.27 | 7.27 | 6.67 | N/A | 7.0 | 7.58 | **4.93** |
| letters | 23.01 | 23.15 | 23.47 | 23.86 | 23.06 | 23.44 | 22.04 | **16.92** |

Table 1: Results for OpenML datasets under churn at cold accuracy metric.

- CIFAR10 and CIFAR100 with ResNet-50, ResNet-101, and ResNet-152.
- IMDB dataset using transformer network.

For each architecture (besides ResNet), we use 5 different sizes. For the fully connected network, we use a simple network with one-hidden layer of $10, 10^2, 10^3, 10^4$, and $10^5$ units, which we call fcn-$x$ where $x$ is the respective size of the hidden layer. For the convolutional neural network, we start with the LeNet5 architecture (LeCun et al., 1998) and scale the number of hidden units by a factor of $x$ for $x = 1, 2, 4, 8, 16$, which we call ConvNet-$x$ for the respective $x$. Finally, we use the basic transformer architecture from Keras tutorial (Keras, 2020) and scale the number of hidden units by $x$ for $x = 1, 2, 4, 8, 16$, which we call Transformer-$x$ for the respective $x$. Code for the models in Keras can be found in the Appendix. For each dataset, we use the standard train/test split if available, otherwise, we fix a random train/test split with ratio 2:1.

**Setup**: For each dataset and neural network, we randomly select from the training set 1000 initial examples, 100 validation examples, and a batch of 1000 examples, and train an initial model using Adam optimizer with default settings on the initial set and early stopping (i.e. stop when there's no improvement on the validation loss after 5 epochs) and default random initialization, and use that model as the base model. Then, for each baseline, we train on the combined initial set and batch (2000 datapoints), again using the Adam optimizer with default settings and the same early stopping scheme and calculate the accuracy and churn against the base model on the test set. We average across 100 runs and provide the error bands in the Appendix. For all the datasets except the OpenML datasets, we also have results for the case of 10000 initial examples, 1000 validation examples, and a batch 1000. We also show results for the case of 100 initial samples, 1000 validation examples, and a batch of 1000 for all of the datasets. Due to space, we show those results in the Appendix. We ran our experiments on a cloud environment. For each run, we used a NVIDIA V100 GPU, which took up to several days to finish all 100 trials.

**Baselines**: We test our method against the following baselines. (1) **Cold start**, where we train the model from scratch with the default initializer. (2) **Warm start**, where we initialize the model's parameters to that of the base model before training. (3) **Shrink-perturb** (Ash & Adams, 2019), which is a method designed to improve warm-starting by initializing the model's weights to $\alpha \cdot \theta_{\text{base}} + (1-\alpha) \cdot \theta_{\text{init}}$ before training, where $\theta_{\text{base}}$ are the weights of the base model, $\theta_{\text{init}}$ is a randomly initialized model, and $\alpha$ is a hyperparameter we tune across $\{0.1, 0.2, ..., 0.9\}$. (4) **Mixup** (Zhang et al., 2017) (a baseline suggested for a different notion of churn (Bahri & Jiang, 2021)), which trains an convex combinations of pairs of datapoints. We search over its hyperparameter $\alpha \in \{0.1, ..., 0.9\}$, as defined in Zhang et al. (2017). (5) **Label smoothing** (Szegedy et al., 2016), which was suggested by Bahri & Jiang (2021) for the variance notion of churn, proceeds by training on a convex combination between the original labels and the base models' soft prediction. We tune across the convex combination weight $\alpha \in \{0.1, 0.2, ..., 0.9\}$. (6) **Co-distillation** (Anil et al., 2018), which was proposed for the variance notion of churn, where we train two warm-started networks that train simultaneously on a loss that is a convex combination on the original loss and a loss on the difference between their predictions. We tune across the convex combination weight $\alpha \in \{0.1, 0.2, ..., 0.9\}$. (7) **Anchor** (Fard

| Dataset | cold | warm | s-perturb | mixup | ls | co-dist | anchor | distill |
|---|---|---|---|---|---|---|---|---|
| mnist | 6.68 | N/A | 6.78 | 5.93 | 5.02 | N/A | 5.21 | **4.81** |
| fashion mnist | 18.48 | N/A | 17.08 | 16.93 | 16.52 | 16.53 | 15.75 | **11.9** |
| emnist balanced | 42.21 | N/A | 37.46 | 37.12 | 35.53 | N/A | 33.64 | **29.41** |
| emnist byclass | 36.4 | N/A | 32.33 | 31.74 | 30.79 | 31.96 | 30.42 | **24.5** |
| emnist bymerge | 34.17 | 30.62 | 30.38 | 29.86 | 28.72 | 29.59 | 27.07 | **21.28** |
| emnist letters | 29.58 | N/A | 26.99 | 26.2 | 24.61 | N/A | 23.22 | **20.16** |
| emnist digits | 6.81 | N/A | 6.95 | 6.09 | 5.29 | N/A | 5.42 | **4.81** |
| emnist mnist | 6.42 | N/A | 6.28 | 5.67 | 4.9 | N/A | 5.21 | **4.49** |
| kmnist | 15.95 | N/A | 14.08 | 13.41 | 12.08 | N/A | 12.0 | **9.9** |
| k49 mnist | 46.35 | N/A | 39.48 | 39.46 | 37.33 | 39.99 | 35.24 | **29.46** |
| svhn | 32.12 | 26.88 | 27.39 | 29.2 | 29.21 | 26.01 | 25.43 | **22.64** |
| cifar10 | 52.01 | 47.57 | 46.36 | 47.17 | 47.92 | 44.61 | 45.75 | **29.13** |

Table 2: Results for MNIST variants, SVHN and CIFAR10 under churn at cold accuracy metric.

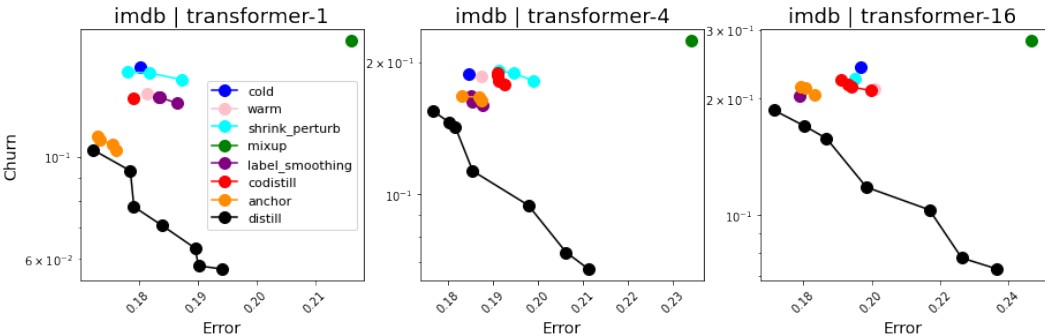

Figure 2: IMDB dataset with Transformer-1, Transformer-4 and Transformer-16. We show the Pareto frontier for each of the baselines. We see that distillation is able to obtain solutions that dominate the other baselines in both churn and accuracy.

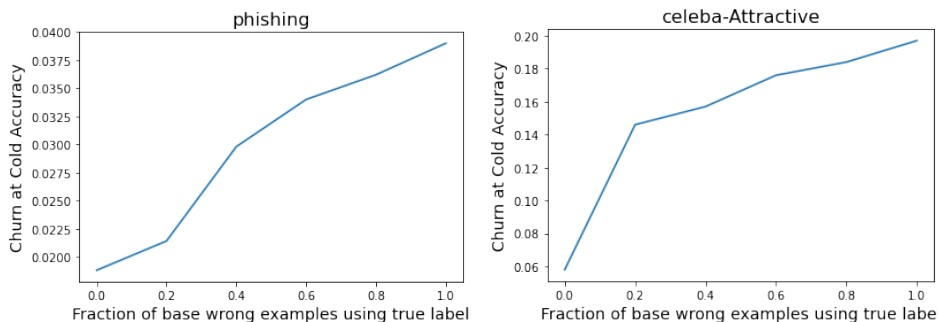

Figure 3: **Distillation vs Anchor Ablation**: We provide an ablation study further showing that using the true labels for wrongly predicted examples by the base model (as done in anchor method) is worse than using distillation for all the examples. We show the performance as we vary the number of wrongly predicted examples that we use the true label instead of the distilled label. The $x$-axis is the fraction of the most (sorted by softmax score) wrongly predicted examples (i.e. 0 is distillation and 1 is anchor method) and $y$-axis is the churn at cold accuracy metric. We show the results for phishing dataset using fcn-1000 and celebA dataset predicting attractiveness using convnet-1, where the average accuracies across the runs of the base model were $93.3\%$ and $69.2\%$, respectively.

et al., 2016), which as noted in Section 4.2, proceeds by optimizing the cross-entropy loss on a modified label: we use the label $\alpha\mathbf{g}(x) + (1-\alpha)\mathbf{e}_y$ when the base model $\mathbf{g}$ agrees with the true label

$y$, and $\eta\mathbf{e}_y$ otherwise. We tune across $\alpha \in \{0.1, 0.2, ..., 0.9\}$ and $\eta \in \{0.5, 0.7, 1\}$. For distillation, we tune the trade-off parameter $\lambda$ across $\{0.1, 0.2, ..., 0.9\}$.

**Metric**: All of the methods will produce a model that we evaluate for both accuracy and churn with respect to the base model on the test set. We consider the hard notion of churn, which measures the average difference in hard predictions w.r.t. the base classifier on a test set. We will see later that there is often-times a trade-off between accuracy and churn, and in an effort to produce one metric for quantitative evaluation, we propose *churn at cold accuracy* metric, which is defined as follows. Each baseline produces a set of models (one for each hyperparameter setting). We take the averaged churn and accuracy across the 100 runs and choose the model with the lowest churn that is at least as accurate as the cold-start model (it's possible that no such model exists for that method). This way, we can identify the method that delivers the lowest churn but still performs at least as well as if we trained on the updated dataset in a vanilla manner. We believe this metric is practically relevant as a practitioner is unlikely to accept a reduction in accuracy to reduce churn.

## 5.2 RESULTS

The detailed results for the following experiments can be found in the Appendix. Given space constraints, we only provide a high level summary in this section

**OpenML datasets with fully-connected networks**: In Table 1 we show the results for the OpenML datasets using the fcn-1000 network. We see that distillation performs the well across the board, and for the other fully connected network sizes, distillation is the best in the majority of cases ($84\%$ of the time for initial batch size 1000 and $52\%$ of time for initial batch size 100).

**MNIST variants, SVHN, and CIFAR10 with convolutional networks**: In Table 2, we show the results for 10 MNIST variants, SVHN and CIFAR10 using convnet-4. We see that distillation performs strongly across the board. We found that distillation performs best in $84\%$ of combinations between dataset and network. When we increase the initial sample size to 10000 and keep the batch size fixed at 1000, then we found that label smoothing starts becoming competitive with distillation, where distillation is best $64\%$ of the time, and label smoothing wins by a small margin all other times. We only saw this phenomenon for a handful of the MNIST variants, which suggests that label smoothing may be especially effective in these situations. When we decreased the initial sample down to 100 and kept the batch size the same, we found that distillation was best $48\%$ of the time, with Anchor being the second best method winning $24\%$ of the time.

For SVHN and CIFAR10, of the 10 combinations, distillation performs the best on all 10 out of the 10. If we increased the initial sample size to 10000 and kept the batch size fixed at 1000, then we find that distillation still performs the best all 10 out of 10 combinations. If we decreased the initial sample size to 100 and kept the same batch size, then distillation performs the best on 8 out of the 10 combinations.

**CelebA with convolutional networks**: Across all 200 combinations of task and network, distillation performs the best $79\%$ of the time. Moreover, if we increased the initial sample size to 10000 and kept the batch size fixed at 1000, distillation is even better, performing the best $91.5\%$ of the time. If we decreased the initial sample size to 100, then distillation is best $96\%$ of the time.

**CIFAR10 and CIFAR100 with ResNet**: Due to the computational costs, we only run these experiments for initial sample size 1000. In all cases (across ResNet-50, ResNet-101 and ResNet-152), we see that distillation outperforms the other baselines.

**IMDB with transformer network**: We experimented for initial batch size 100, 1000, and 10000. We found that distillation performed the best the majority of the time, where the only notable weak performance was in some instances where no baselines were even able to reach the accuracy of the cold starting method. In Figure 2 we show the Pareto frontiers of the various baselines as well as plotting cost of each method as we vary the trade-off between accuracy and churn. We see that not only does distillation do well in churn, but it performs the best at any trade-off between churn and accuracy for the cases shown.

**Conclusion**: We have proposed knowledge distillation as a new practical solution to churn reduction, and provided both theoretical and empirical justifications for the approach.

**Reproducibility Statement**: All details of experimental setup are in the main text, along with descriptions of the baselines and what hyperparameters were swept across. Code can be found in the Appendix. All proofs are in the Appendix.

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

# A PROOFS

## A.1 PROOF OF PROPOSITION 1

**Proposition** (Restated). *Let $(\ell, d)$ be defined as in (2) for a strictly proper scoring function $\phi$. Suppose $\phi(\mathbf{u})$ is strictly convex in $\mathbf{u}$. Then there exists $\lambda^* \in [0, 1]$ such that the following is an optimal-feasible classifier for (1):*

$$\mathbf{h}^*(x) = \lambda^* \mathbf{p}(x) + (1 - \lambda^*)\mathbf{g}(x).$$

*Furthermore, if $\mathbf{u} \cdot \phi(\mathbf{u})$ is $\alpha$-strongly concave over $\mathbf{u} \in \Delta_m$ w.r.t. the $L_q$-norm, then*

$$\lambda^* \leqslant \sqrt{\frac{2\epsilon}{\alpha \, \mathbf{E}_x\left[\|\mathbf{p}(x) - \mathbf{g}(x)\|_q^2\right]}}.$$

*Proof.* Let $\mathbf{h}^*$ denote an optimal feasible solution for (1). We first note that

$$R(\mathbf{h}) = \mathbf{E}_{x,y}\left[\ell(y, \mathbf{h}(x))\right] = \mathbf{E}_x\left[\mathbf{E}_{y|x}\left[\ell(y, \mathbf{h}(x))\right]\right] = \mathbf{E}_x\left[\sum_{i\in[m]} p_i(x)\phi_i(\mathbf{h}(x))\right]$$

and

$$C(\mathbf{h}) = \mathbf{E}_x\left[\sum_{i\in[m]} g_i(x)\left(\phi_i(\mathbf{h}(x)) - \phi_i(\mathbf{g}(x))\right)\right].$$

Because $\phi_i$ is strictly convex in its argument, both $R(\mathbf{h})$ and $C(\mathbf{h})$ are strictly convex in $\mathbf{h}$. In other words, for any $\alpha \in [0, 1]$, and classifiers $\mathbf{h}_1, \mathbf{h}_2$, $R(\alpha\mathbf{h}_1 + (1 - \alpha)\mathbf{h}_2) < \alpha R(\mathbf{h}_1) + (1 - \alpha)R(\mathbf{h}_2)$, and similarly for $C$. Furthermore because $C(\mathbf{g}) = 0 < \epsilon$, the constraint is strictly feasible, and hence strong duality holds for (1) (as a result of Slater's condition being satisfied). Therefore (1) can be equivalently formulated as a max-min problem:

$$\max_{\mu\in\mathbb{R}_+} \min_{\mathbf{h}} R(\mathbf{h}) + \mu C(\mathbf{h}),$$

for which there exists a $\mu^* \in \mathbb{R}_+$ such that $(\mu^*, \mathbf{h}^*)$ is a saddle point. The strict convexity of $R(h)$ and $C(h)$ gives us that $\mathbf{h}^*$ is the unique minimizer of $R(\mathbf{h}) + \mu^* C(\mathbf{h})$. Setting $\lambda^* = \frac{1}{1+\mu^*}$, we equivalently have that $\mathbf{h}^*$ is a unique minimizer of the weighted objective $\lambda^* R(\mathbf{h}) + (1 - \lambda^*)C(\mathbf{h})$.

We next show that the minimizer $\mathbf{h}^*$ is of the required form. Expanding the $R$ and $C$, we have:

$$\lambda^* R(\mathbf{h}) + (1 - \lambda^*)C(\mathbf{h})$$

$$= \mathbf{E}_x\left[\sum_{i\in[m]} \left(\lambda^* p_i(x) + (1 - \lambda^*)g_i(x)\right)\phi_i(\mathbf{h}(x)) - (1 - \lambda^*)g_i(x)\phi_i(\mathbf{g}(x))\right]$$

$$= \mathbf{E}_x\left[\sum_{i\in[m]} \left(\lambda^* p_i(x) + (1 - \lambda^*)g_i(x)\right)\phi_i(\mathbf{h}(x))\right] + \text{a term independent of } \mathbf{h}$$

$$= \mathbf{E}_x\left[\sum_{i\in[m]} \bar{p}_i(x)\,\phi_i(\mathbf{h}(x))\right] + \text{a term independent of } \mathbf{h}, \tag{6}$$

where $\bar{\mathbf{p}}(x) = \lambda^* \mathbf{p}(x) + (1 - \lambda^*)\mathbf{g}(x)$.

Note that it suffices to minimize (6) point-wise, i.e. to choose $\mathbf{h}^*$ so that the term within the expectation $\sum_{i\in[m]} \bar{p}_i(x)\,\phi_i(\mathbf{h}(x))$ is minimized for each $x$. For a fixed $x$, the inner term is minimized when $\mathbf{h}^*(x) = \bar{\mathbf{p}}(x)$. This is because of our assumption that $\phi$ is a strictly proper scoring function, i.e. for any distribution $\mathbf{u}$, the weighted loss $\sum_i u_i\phi_i(\mathbf{v})$ is uniquely minimized by $\mathbf{v} = \mathbf{u}$. Therefore (6) is minimized by $\mathbf{h}^*(x) = \bar{\mathbf{p}}(x) = \lambda^* \mathbf{p}(x) + (1 - \lambda^*)\mathbf{g}(x)$.

To bound $\lambda^*$, we use a result from Williamson et al. (2016); Agarwal (2014) to lower bound $C(\mathbf{h})$ in terms of the norm difference $\|\mathbf{h}(x) - \mathbf{g}(x)\|_q$. Define $\mathcal{Q}(\mathbf{u}) = \inf_{\mathbf{v}\in\Delta_m} \mathbf{u} \cdot \phi(\mathbf{v})$. Because $\phi$ is a proper scoring function, the infimum is attained at $\mathbf{v} = \mathbf{u}$. Therefore $\mathcal{Q}(\mathbf{u}) = \mathbf{u} \cdot \phi(\mathbf{u})$, which recall is assumed to be strongly concave. Also, note that $\mathcal{Q}(\mathbf{u}) = \inf_{\mathbf{v}\in\Delta_m} \mathbf{u} \cdot \phi(\mathbf{v})$ is an infimum of "linear" functions in $\mathbf{u}$, and therefore $\nabla\mathcal{Q}(\mathbf{u}) = \phi(\mathbf{u})$ is a super-differential for $\mathcal{Q}$ at $\mathbf{u}$. See Proposition 7 in Williamson et al. (2016) for more details.

We now re-write $C(\mathbf{h})$ in terms of $\mathcal{Q}$ and lower bound it using the strong concavity property:

$$
\begin{aligned}
C(\mathbf{h}) &= \mathbf{E}_x\Big[\mathbf{g}(x) \cdot (\phi(\mathbf{h}(x)) - \phi(\mathbf{g}(x)))\Big] \\
&= \mathbf{E}_x\Big[\mathbf{h}(x) \cdot \phi(\mathbf{h}(x)) + (\mathbf{g}(x) - \mathbf{h}(x)) \cdot \phi(\mathbf{h}(x)) - \mathbf{g}(x) \cdot \phi(\mathbf{g}(x))\Big] \\
&= \mathbf{E}_x\Big[\mathcal{Q}(\mathbf{h}(x)) + (\mathbf{g}(x) - \mathbf{h}(x)) \cdot \nabla\mathcal{Q}(\mathbf{h}(\mathbf{x})) - \mathcal{Q}(\mathbf{g}(\mathbf{x}))\Big] \\
&\geqslant \mathbf{E}_x\left[\frac{\alpha}{2}\|\mathbf{h}(x) - \mathbf{g}(x)\|_q^2\right],
\end{aligned}
$$

where the last step uses the fact that $\mathcal{Q}$ is $\alpha$-strongly concave over $\mathbf{u} \in \Delta_m$ w.r.t. the $L_q$-norm.

Since the optimal scorer $\mathbf{h}^*$ satisfies the coverage constraint $C(\mathbf{h}^*) \leqslant \epsilon$, we have from the above bound

$$
\mathbf{E}_x\left[\frac{\alpha}{2}\|\mathbf{h}^*(x) - \mathbf{g}(x)\|_q^2\right] \leqslant \epsilon.
$$

Substituting for $\mathbf{h}^*$, we have:

$$
\mathbf{E}_x\left[\frac{(\lambda^*)^2\alpha}{2}\|\mathbf{p}(x) - \mathbf{g}(x)\|_q^2\right] \leqslant \epsilon,
$$

or

$$
(\lambda^*)^2 \leqslant \frac{2\epsilon}{\alpha\mathbf{E}_x\left[\|\mathbf{p}(x) - \mathbf{g}(x)\|_q^2\right]},
$$

which gives us the desired bound on $\lambda^*$. $\qquad\square$

## A.2 Proof of Theorem 2

**Theorem** (Restated). *Let the scoring function $\phi : \Delta_m \to \mathbb{R}_+^m$ be convex, and $\|\phi(\mathbf{z})\|_\infty < B, \forall \mathbf{z} \in \Delta_m$. Let the set of classifiers $\mathcal{H}$ be convex, with the base classifier $\mathbf{g} \in \mathcal{H}$. Suppose $C$ and $R$ enjoy the following generalization bounds: for any $\delta \in (0, 1)$, w.p. $\geqslant 1 - \delta$ over draw of $S \sim D^n$, for any $\mathbf{h} \in \mathcal{H}$,*

$$
|R(\mathbf{h}) - \widehat{R}(\mathbf{h})| \leqslant \Delta_R(n, \delta); \qquad |C(\mathbf{h}) - \widehat{C}(\mathbf{h})| \leqslant \Delta_C(n, \delta),
$$

*for some $\Delta_R(n, \delta)$ and $\Delta_C(n, \delta)$ that is decreasing in $n$ and approaches 0 as $n \to \infty$. Let $\widetilde{\mathbf{h}}$ be an optimal-feasible classifier in $\mathcal{H}$, i.e. $C(\widetilde{\mathbf{h}}) \leqslant \epsilon$ and $R(\widetilde{\mathbf{h}}) \leqslant R(\mathbf{h})$ for all classifiers $\mathbf{h}$ for which $C(\mathbf{h}) \leqslant \epsilon$. Let $\widehat{\mathbf{h}}$ be the classifier returned by Algorithm 1 with $\Lambda = \left\{\max\{\frac{\epsilon}{\epsilon+2B}, u\} \,\big|\, u \in \{\frac{1}{L}, \frac{2}{L}, \ldots, 1\}\right\}$ for some $L \in \mathbb{N}_+$. For any $\delta \in (0, 1)$, w.p. $\geqslant 1 - \delta$ over draw of $S \sim D^n$,*

$$
\textbf{Optimality}: R(\widehat{\mathbf{h}}) \leqslant R(\widetilde{\mathbf{h}}) + \mathcal{O}\left(\left(1 + \frac{2B}{\epsilon}\right)\left(\Delta_R(n, \delta) + \Delta_C(n, \delta) + \frac{B}{L}\right)\right),
$$

$$
\textbf{Feasibility}: C(\widehat{\mathbf{h}}) \leqslant \epsilon + \Delta_C(n, \delta).
$$

We first note that because $\|\phi(\mathbf{z})\|_\infty < B, \forall \mathbf{z} \in \Delta_m$, both $\widehat{R}(\mathbf{h}) < B$ and $\widehat{C}(\mathbf{h}) < B$. Also, because $\phi_i$ is convex, both $\widehat{R}(\mathbf{h})$ and $\widehat{C}(\mathbf{h})$ are convex in $\mathbf{h}$. In other words, for any $\alpha \in [0, 1]$, and classifiers $\mathbf{h}_1, \mathbf{h}_2, \widehat{R}(\alpha\mathbf{h}_1 + (1 - \alpha)\mathbf{h}_2) \leqslant \alpha\widehat{R}(\mathbf{h}_1) + (1 - \alpha)\widehat{R}(\mathbf{h}_2)$, and similarly for $\widehat{C}$. Furthermore, the objective in (4) can be decomposed into a convex combination of the empirical risk and churn:

$$
\begin{aligned}
\widehat{\mathcal{L}}_\lambda(\mathbf{h}) &= \frac{1}{n}\sum_{i=1}^n (\lambda\mathbf{e}_{y_i} + (1 - \lambda)\mathbf{g}(x_i)) \cdot \phi(\mathbf{h}(x_i)) \\
&= \lambda\widehat{R}(\mathbf{h}) + (1 - \lambda)\widehat{C}(\mathbf{h}) + \frac{1 - \lambda}{n}\sum_{i=1}^n \mathbf{g}(x_i) \cdot \phi(\mathbf{g}(x_i)).
\end{aligned}
$$

Therefore minimizing $\widehat{\mathcal{L}}_\lambda(\mathbf{h})$ is equivalent to minimizing the Lagrangian function

$$
\widetilde{\mathcal{L}}_\lambda(\mathbf{h}) = \lambda\widehat{R}(\mathbf{h}) + (1 - \lambda)(\widehat{C}(\mathbf{h}) - \epsilon) \tag{7}
$$

over $\mathbf{h}$. Moreover, each $\mathbf{h}_k$ minimizes $\widetilde{\mathcal{L}}_{\lambda_k}(\mathbf{h})$.

We also note that the churn-constrained optimization problem in (1) can be posed as a Lagrangian game between a player that seeks to minimize the above Lagrangian over $\mathbf{h}$ and a player that seeks to maximize the Lagrangian over $\lambda$. The next two lemmas show that Algorithm 1 can be seen as finding an approximate equilibrium of this two-player game.

**Lemma 5.** *Let the assumptions on $\phi$ and $\mathcal{H}$ in Theorem 2 hold. Let $\widehat{\mathbf{h}}$ be the classifier returned by Algorithm 1 when $\Lambda$ is set to $\Lambda = \big\{ \max\{ \frac{\epsilon}{\epsilon+2B}, u \} \,|\, u \in \{ \frac{1}{L}, \dots, 1 \} \big\}$ of the range $\big[ \frac{\epsilon}{\epsilon+2B}, 1 \big]$ for some $L \in \mathbb{N}_+$. Then there exists a bounded Lagrange multiplier $\bar{\lambda} \in \big[ \frac{\epsilon}{\epsilon+2B}, 1 \big]$ such that $(\widehat{\mathbf{h}}, \bar{\lambda})$ forms an equilibrium of the Lagrangian min-max game:*

$$\bar{\lambda}\widehat{R}(\widehat{\mathbf{h}}) \,+\, (1-\bar{\lambda})(\widehat{C}(\widehat{\mathbf{h}}) - \epsilon) \;=\; \min_{\mathbf{h} \in \mathrm{co}(\mathbf{h}_1, \dots, \mathbf{h}_L)} \bar{\lambda}\widehat{R}(\mathbf{h}) \,+\, (1-\bar{\lambda})(\widehat{C}(\mathbf{h}) - \epsilon)$$

$$\max_{\lambda \in [0,1]} (1-\lambda)(\widehat{C}(\widehat{\mathbf{h}}) - \epsilon) \;=\; (1-\bar{\lambda})(\widehat{C}(\widehat{\mathbf{h}}) - \epsilon).$$

*Proof.* The classifier $\widehat{\mathbf{h}}$ returned by Algorithm 1 is a solution to the following constrained optimization problem over the convex-hull of the classifiers $\mathbf{h}_1, \dots, \mathbf{h}_L$:

$$\min_{\mathbf{h} \in \mathrm{co}(\mathbf{h}_1, \dots, \mathbf{h}_L)} \widehat{R}(\mathbf{h}) \;\; \text{s.t.} \;\; \widehat{C}(\mathbf{h}) \leqslant \epsilon.$$

Consequently, there exists a $\bar{\lambda} \in [0, 1]$ such that:

$$\bar{\lambda}\widehat{R}(\widehat{\mathbf{h}}) \,+\, (1-\bar{\lambda})(\widehat{C}(\widehat{\mathbf{h}}) - \epsilon) \;=\; \min_{\mathbf{h} \in \mathrm{co}(\mathbf{h}_1, \dots, \mathbf{h}_L)} \bar{\lambda}\widehat{R}(\mathbf{h}) \,+\, (1-\bar{\lambda})(\widehat{C}(\mathbf{h}) - \epsilon) \tag{8}$$

$$\max_{\lambda \in [0,1]} (1-\lambda)(\widehat{C}(\widehat{\mathbf{h}}) - \epsilon) \;=\; (1-\bar{\lambda})(\widehat{C}(\widehat{\mathbf{h}}) - \epsilon). \tag{9}$$

To see this, note that the KKT conditions (along with the convexity of $R$ and $C$) give us that there exists a Lagrange multiplier $\bar{\mu} \geqslant 0$ such that

$$\widehat{\mathbf{h}} \in \operatorname*{argmin}_{\mathbf{h} \in \mathrm{co}(\mathbf{h}_1, \dots, \mathbf{h}_L)} \widehat{R}(\mathbf{h}) \,+\, \bar{\mu}(\widehat{C}(\mathbf{h}) - \epsilon) \;\; \text{(stationarity)}$$

$$\bar{\mu}(\widehat{C}(\widehat{\mathbf{h}}) - \epsilon) = 0 \;\; \text{(complementary slackness)}.$$

When $\widehat{C}(\widehat{\mathbf{h}}) \leqslant \epsilon$, $\bar{\mu} = 0$, and so (8) and (9) are satisfied for $\bar{\lambda} = 1$. When $\widehat{C}(\widehat{\mathbf{h}}) = \epsilon$, then (8) and (9) are satisfied for $\bar{\lambda} = \frac{1}{1+\bar{\mu}}$.

It remains to show that that $\bar{\lambda} \in \big[ \frac{\epsilon}{\epsilon+2B}, 1 \big]$. For this, we first show that there exists a $\mathbf{h}' \in \mathrm{co}(\mathbf{h}_1, \dots, \mathbf{h}_L)$ such that $\widehat{C}(\mathbf{h}') \leqslant \epsilon/2$. To see why, pick $\mathbf{h}'$ to be the minimizer of the Lagrangian $\widetilde{\mathcal{L}}_\lambda(\mathbf{h})$ over all $\mathbf{h} \in \mathcal{H}$ for $\lambda = \frac{\epsilon}{\epsilon+2B}$. Because $\widetilde{\mathcal{L}}_\lambda(\mathbf{h}') \leqslant \widetilde{\mathcal{L}}_\lambda(\mathbf{g}) \leqslant \lambda B - (1-\lambda)\epsilon$, where $\mathbf{g}$ is the base classifier that we have assumed is in $\mathcal{H}$, it follows that $\widehat{C}(\mathbf{h}') \leqslant \frac{\lambda}{1-\lambda} B \leqslant \epsilon/2$.

Next, by combining (8) and (9), we have

$$\bar{\lambda}\widehat{R}(\widehat{\mathbf{h}}) \,+\, \max_{\lambda \in [0,1]} (1-\lambda)(\widehat{C}(\widehat{\mathbf{h}}) - \epsilon) \;=\; \min_{\mathbf{h} \in \mathrm{co}(\mathbf{h}_1, \dots, \mathbf{h}_L)} \bar{\lambda}\widehat{R}(\mathbf{h}) \,+\, (1-\bar{\lambda})(\widehat{C}(\mathbf{h}) - \epsilon).$$

Lower bounding the LHS by setting $\lambda = 1$ and upper bounding the RHS by setting $\mathbf{h} = \mathbf{h}'$, we get:

$$\bar{\lambda}\widehat{R}(\widehat{\mathbf{h}}) \;\leqslant\; \bar{\lambda}\widehat{R}(\mathbf{h}') \,-\, (1-\bar{\lambda})\frac{\epsilon}{2},$$

which gives us:

$$\epsilon/2 \;\leqslant\; \bar{\lambda}(\epsilon/2 + \widehat{R}(\mathbf{h}') - \widehat{R}(\widehat{\mathbf{h}})) \;\leqslant\; \bar{\lambda}(\epsilon/2 + B).$$

Hence $\bar{\lambda} \geqslant \frac{\epsilon}{\epsilon+2B}$, which completes the proof. $\qquad\square$

**Lemma 6.** *Let $\widehat{\mathbf{h}}$ be the classifier returned by Algorithm 1 when $\Lambda$ is set to $\Lambda = \big\{ \max\{ \frac{\epsilon}{\epsilon+2B}, u \} \,|\, u \in \{ \frac{1}{L}, \dots, 1 \} \big\}$ of the range $\big[ \frac{\epsilon}{\epsilon+2B}, 1 \big]$ for some $L \in \mathbb{N}_+$. Fix $\delta \in (0, 1)$. Suppose $R$ and $C$ satisfy the generalization bounds in Theorem 2 with error bounds $\Delta_R(n, \delta)$ and $\Delta_C(n, \delta)$ respectively. Then there exists a bounded Lagrange multiplier $\widehat{\lambda} \in \big[ \frac{\epsilon}{\epsilon+2B}, 1 \big]$ such that $(\widehat{\mathbf{h}}, \widehat{\mu})$ forms an approximate equilibrium for the Lagrangian min-max game, i.e. w.p. $\geqslant 1 - \delta$ over draw of sample $S \sim D^n$,*

$$\widehat{\lambda}R(\widehat{\mathbf{h}}) \,+\, (1-\widehat{\lambda})(C(\widehat{\mathbf{h}}) - \epsilon) \;\leqslant\; \min_{\mathbf{h} \in \mathcal{H}} \widehat{\lambda}R(\mathbf{h}) \,+\, (1-\widehat{\lambda})(C(\mathbf{h}) - \epsilon)$$

$$+ \,\mathcal{O}\left( \Delta_R(n, \delta) + \Delta_C(n, \delta) + B/L \right) \tag{10}$$

*and*

$$\max_{\lambda \in [0,1]} (1-\lambda)(C(\widehat{\mathbf{h}}) - \epsilon) \;\leqslant\; (1-\widehat{\lambda})(C(\widehat{\mathbf{h}}) - \epsilon) + \mathcal{O}\left( \Delta_C(n, \delta) + B/L \right). \tag{11}$$

*Proof.* We have from Lemma 5 that there exists $\bar{\lambda} \in [\frac{\epsilon}{\epsilon+2B}, 1]$ such that

$$\bar{\lambda}\widehat{R}(\widehat{\mathbf{h}}) + (1-\bar{\lambda})(\widehat{C}(\widehat{\mathbf{h}}) - \epsilon) = \min_{\mathbf{h}\in\mathrm{co}(\mathbf{h}_1,\ldots,\mathbf{h}_L)} \bar{\lambda}\widehat{R}(\mathbf{h}) + (1-\bar{\lambda})(\widehat{C}(\mathbf{h}) - \epsilon) \tag{12}$$

$$\max_{\lambda\in[0,1]}(1-\lambda)(\widehat{C}(\widehat{\mathbf{h}}) - \epsilon) = (1-\bar{\lambda})(\widehat{C}(\widehat{\mathbf{h}}) - \epsilon). \tag{13}$$

Algorithm 1 works with a discretization $\Lambda = \left\{\max\{\frac{\epsilon}{\epsilon+2B}, u\} \,|\, u \in \{\frac{1}{L}, \ldots, 1\}\right\}$ of the range $[\frac{\epsilon}{\epsilon+2B}, 1]$. Allowing $\widehat{\lambda}$ to denote the closest value to $\bar{\lambda}$ in this set, we have from (12):

$$\begin{aligned}
\widehat{\lambda}\widehat{R}(\widehat{\mathbf{h}}) + (1-\widehat{\lambda})(\widehat{C}(\widehat{\mathbf{h}}) - \epsilon) &\leqslant \min_{\mathbf{h}\in\mathrm{co}(\mathbf{h}_1,\ldots,\mathbf{h}_L)} \widehat{\lambda}\widehat{R}(\mathbf{h}) + (1-\widehat{\lambda})(\widehat{C}(\mathbf{h}) - \epsilon) + \frac{4B}{L} \\
&= \min_{\mathbf{h}\in\mathcal{H}} \widehat{\lambda}\widehat{R}(\mathbf{h}) + (1-\widehat{\lambda})(\widehat{C}(\mathbf{h}) - \epsilon) + \frac{4B}{L},
\end{aligned} \tag{14}$$

where the last step follows from the fact that $\mathrm{co}(\mathbf{h}_1, \ldots, \mathbf{h}_L) \subseteq \mathcal{H}$ and each $\mathbf{h}_k$ was chosen to minimize $(1-\lambda_k)\widehat{R}(\mathbf{h}) + \lambda_k(\widehat{C}(\mathbf{h}) - \epsilon)$ for $\lambda_k \in \Lambda$. Similarly, we have from (13),

$$\max_{\lambda\in[0,1]}(1-\lambda)(C(\widehat{\mathbf{h}}) - \epsilon) \leqslant (1-\widehat{\lambda})(C(\widehat{\mathbf{h}}) - \epsilon) + \frac{B}{L}. \tag{15}$$

What remains is to apply the generalization bounds for $R$ and $C$ to (14) and (15). We first bound the LHS of (14). We have with probability at least $1 - \delta$ over draw of $S \sim D^n$:

$$\begin{aligned}
&\widehat{\lambda}\widehat{R}(\widehat{\mathbf{h}}) + (1-\widehat{\lambda})(\widehat{C}(\widehat{\mathbf{h}}) - \epsilon) \\
&\geqslant \widehat{\lambda}R(\widehat{\mathbf{h}}) + (1-\widehat{\lambda})(C(\widehat{\mathbf{h}}) - \epsilon) - \widehat{\lambda}\Delta_R(n,\delta) - (1-\widehat{\lambda})\Delta_C(n,\delta) \\
&\geqslant \widehat{\lambda}R(\widehat{\mathbf{h}}) + (1-\widehat{\lambda})(C(\widehat{\mathbf{h}}) - \epsilon) - \Delta_R(n,\delta) - \Delta_C(n,\delta),
\end{aligned} \tag{16}$$

where the last step uses the fact that $0 \leqslant \widehat{\lambda} \leqslant 1$. For the RHS, we have with the same probability:

$$\begin{aligned}
&\min_{\mathbf{h}\in\mathcal{H}} \left\{\widehat{\lambda}\widehat{R}(\mathbf{h}) + (1-\widehat{\lambda})(\widehat{C}(\mathbf{h}) - \epsilon)\right\} + 4B/L \\
&\leqslant \min_{\mathbf{h}\in\mathcal{H}} \left\{\widehat{\lambda}R(\mathbf{h}) + (1-\widehat{\lambda})(C(\mathbf{h}) - \epsilon) + 4B/L + \widehat{\lambda}\Delta_R(n,\delta) + (1-\widehat{\lambda})\Delta_C(n,\delta)\right\} \\
&\leqslant \min_{\mathbf{h}\in\mathcal{H}} \left\{\widehat{\lambda}R(\mathbf{h}) + (1-\widehat{\lambda})(C(\mathbf{h}) - \epsilon)\right\} + 4B/L + \Delta_R(n,\delta) + \Delta_C(n,\delta),
\end{aligned}$$

where we again use $0 \leqslant \widehat{\lambda} \leqslant 1$. Combining (14) with (16) and (17) completes the proof for the first part of the lemma. Applying the generalization bounds to (15), we have with the same probability:

$$\begin{aligned}
B/L &\\
&\geqslant \max_{\lambda\in[0,1]}(1-\lambda)(\widehat{C}(\widehat{\mathbf{h}}) - \epsilon) - (1-\widehat{\lambda})(\widehat{C}(\widehat{\mathbf{h}}) - \epsilon) \\
&\geqslant \max_{\lambda\in[0,1]}\left\{(1-\lambda)(C(\widehat{\mathbf{h}}) - \epsilon) - (1-\lambda)\Delta_C(n,\delta)\right\} - (1-\widehat{\lambda})(C(\widehat{\mathbf{h}}) - \epsilon) - (1-\widehat{\lambda})\Delta_C(n,\delta) \\
&\geqslant \max_{\lambda\in[0,1]}(1-\lambda)(C(\widehat{\mathbf{h}}) - \epsilon) - (1-\widehat{\lambda})(C(\widehat{\mathbf{h}}) - \epsilon) - 2\Delta_C(n,\delta),
\end{aligned}$$

which completes the proof for the second part of the lemma. $\qquad\square$

We are now ready to prove Theorem 2.

*Proof of Theorem 2.* To show optimality, we combine (10) and (11) and get:

$$\begin{aligned}
\widehat{\lambda}\widehat{R}(\widehat{\mathbf{h}}) + \max_{\lambda\in[0,1]}(1-\lambda)(C(\widehat{\mathbf{h}}) - \epsilon) &\leqslant \min_{\mathbf{h}\in\mathcal{H}} \widehat{\lambda}\widehat{R}(\mathbf{h}) + (1-\widehat{\lambda})(\widehat{C}(\mathbf{h}) - \epsilon) \\
&\quad + \mathcal{O}\Big(\Delta_R(n,\delta) + \Delta_C(n,\delta) + B/L\Big). \tag{17}
\end{aligned}$$

We then lower bound the LHS in (17) by setting $\lambda = 1$ and upper bound the RHS by setting $\mathbf{h}$ to the optimal feasible solution $\tilde{\mathbf{h}}$, giving us:

$$\widehat{\lambda}R(\widehat{\mathbf{h}}) \leqslant \widehat{\lambda}R(\tilde{\mathbf{h}}) + (1 - \widehat{\lambda})(0) + \mathcal{O}\Big(\Delta_R(n,\delta) + \Delta_C(n,\delta) + \tfrac{B}{L}\Big).$$

Dividing both sides by $\widehat{\lambda}$,

$$R(\widehat{\mathbf{h}}) \leqslant R(\tilde{\mathbf{h}}) + \tfrac{1}{\widehat{\lambda}}\mathcal{O}\Big(\Delta_R(n,\delta) + \Delta_C(n,\delta) + \tfrac{B}{L}\Big).$$

Lower bounding $\widehat{\lambda}$ by $\frac{\epsilon}{\epsilon + 2B}$ gives us the desired optimality result.

The feasibility result directly follows from the fact that Algorithm 1 chooses a $\widehat{\mathbf{h}}$ that satisfies the empirical churn constraint $\widehat{C}(\widehat{\mathbf{h}}) \leqslant \epsilon$, and from the generalization bound for $C$. $\qquad\square$

## A.3 PROOF OF PROPOSITION 4

**Proposition** (Restated). *When $\mathbf{g}(x) = \mathbf{p}(x), \forall x$, for any given $\lambda \in [0,1]$, the minimizer for the distillation loss in (3) over all classifiers $h$ is given by:*

$$\mathbf{h}^*(x) = \mathbf{p}(x),$$

*whereas the minimizer of the anchor loss in (5) is given by:*

$$h_j^*(x) = \frac{z_j}{\sum_j z_j} \quad \text{where} \quad z_j = \begin{cases} \alpha p_j^2(x) + (1-\alpha)p_j(x) & \text{if } j = \overline{\mathrm{argmax}}_k \, p_k(x) \\ (\epsilon + \alpha \max_k p_k(x)) \, p_j(x) & \text{otherwise} \end{cases}.$$

*Proof.* For the first part, we expand (3) with $\mathbf{g}(x) = \mathbf{p}(x)$, and have for any $\lambda \in [0,1]$,

$$\begin{aligned} \mathcal{L}_\lambda(h) &= \mathbf{E}_{(x,y)\sim D}\left[(\lambda\mathbf{e}_y + (1-\lambda)\mathbf{p}(x)) \cdot \phi(\mathbf{h}(x))\right] && (18) \\ &= \lambda\mathbf{E}_{(x,y)\sim D}\left[\mathbf{e}_y \cdot \phi(\mathbf{h}(x))\right] + (1-\lambda)\mathbf{E}_{x\sim D_\mathcal{X}}\left[\mathbf{p}(x) \cdot \phi(\mathbf{h}(x))\right] \\ &= \lambda\mathbf{E}_{x\sim D_\mathcal{X}}\left[\mathbf{E}_{y|x}\left[\mathbf{e}_y\right] \cdot \phi(\mathbf{h}(x))\right] + (1-\lambda)\mathbf{E}_{x\sim D_\mathcal{X}}\left[\mathbf{p}(x) \cdot \phi(\mathbf{h}(x))\right] \\ &= \lambda\mathbf{E}_{x\sim D_\mathcal{X}}\left[\mathbf{p}(x) \cdot \phi(\mathbf{h}(x))\right] + (1-\lambda)\mathbf{E}_{x\sim D_\mathcal{X}}\left[\mathbf{p}(x) \cdot \phi(\mathbf{h}(x))\right] \\ &= \mathbf{E}_{x\sim D_\mathcal{X}}\left[\mathbf{p}(x) \cdot \phi(\mathbf{h}(x))\right]. && (19) \end{aligned}$$

For a fixed $x$, the inner term in (19) is minimized when $\mathbf{h}^*(x) = \mathbf{p}(x)$. This is because of our assumption that $\phi$ is a strictly proper scoring function, i.e. for any distribution $\mathbf{u}$, the weighted loss $\sum_i u_i \phi_i(\mathbf{v})$ is uniquely minimized by $\mathbf{v} = \mathbf{u}$. Therefore (19) is minimized by $\mathbf{h}^*(x) = \mathbf{p}(x), \forall x$.

For the second part, we expand (5) with $\mathbf{g}(x) = \mathbf{p}(x)$, and have:

$$\mathcal{L}^{\mathrm{anc}}(\mathbf{h}) = \mathbf{E}_{(x,y)\sim D}\left[\mathbf{a} \cdot \phi(\mathbf{h}(x))\right],$$

where

$$\mathbf{a} = \begin{cases} \alpha\mathbf{p}(x) + (1-\alpha)\mathbf{e}_y & \text{if } y = \overline{\mathrm{argmax}}_k \, p_k(x) \\ \epsilon\mathbf{e}_y & \text{otherwise} \end{cases},$$

For a given $x$, let us denote $j_x = \overline{\mathrm{argmax}}_k \, p_k(x)$. We then have:

$$\begin{aligned} \mathcal{L}^{\mathrm{anc}}(\mathbf{h}) &= \mathbf{E}_{(x,y)\sim D}\left[(\mathbf{1}(y = j_x)(\alpha\mathbf{p}(x) + (1-\alpha)\mathbf{e}_y) + \epsilon\mathbf{1}(y \neq j_x)\mathbf{e}_y) \cdot \phi(\mathbf{h}(x))\right] \\ &= \mathbf{E}_{x\sim D_\mathcal{X}}\left[\mathbf{E}_{y|x}\left[(\mathbf{1}(y = j_x)(\alpha\mathbf{p}(x) + (1-\alpha)\mathbf{e}_y) + \epsilon\mathbf{1}(y \neq j_x)\mathbf{e}_y) \cdot \phi(\mathbf{h}(x))\right]\right] \\ &= \mathbf{E}_{x\sim D_\mathcal{X}}\left[\sum_k p_k(x)\left((\mathbf{1}(k = j_x)(\alpha\mathbf{p}(x) + (1-\alpha)\mathbf{e}_k) + \epsilon\mathbf{1}(k \neq j_x)\mathbf{e}_k) \cdot \phi(\mathbf{h}(x))\right)\right] \\ &= \mathbf{E}_{x\sim D_\mathcal{X}}\left[p_{j_x}(x)(\alpha\mathbf{p}(x) + (1-\alpha)\mathbf{e}_{j_x}) \cdot \phi(\mathbf{h}(x) + \epsilon\sum_{k \neq j_x} p_k(x)\phi_k(\mathbf{h}(x))\right] \\ &= \mathbf{E}_{x\sim D_\mathcal{X}}\Big[p_{j_x}(x)(\alpha p_{j_x}(x) + (1-\alpha))\,\phi_{j_x}(\mathbf{h}(x)) \end{aligned}$$

$$+ p_{j_x}(x) \sum_{k \neq j_x} \alpha p_k(x)\phi_k(\mathbf{h}(x)) + \epsilon \sum_{k \neq j_x} p_k(x)\phi_k(\mathbf{h}(x))\Bigg]$$

$$= \mathbf{E}_{x \sim D_\mathcal{X}} \Bigg[ p_{j_x}(x)\left(\alpha p_{j_x}(x) + (1-\alpha)\right)\phi_{j_x}(\mathbf{h}(x)) + (\alpha p_{j_x}(x) + \epsilon)\sum_{k \neq j_x} p_k(x)\phi_k(\mathbf{h}(x))\Bigg]$$

$$= \mathbf{E}_{x \sim D_\mathcal{X}}\left[\widetilde{\mathbf{p}}(x) \cdot \phi(\mathbf{h}(x))\right], \tag{20}$$

where

$$\widetilde{p}_s(x) = \begin{cases} \alpha p_s^2(x) + (1-\alpha)p_s(x) & \text{if } s = j_x \\ (\alpha p_{j_x}(x) + \epsilon)p_s(x) & \text{otherwise} \end{cases}$$

$$= \begin{cases} \alpha p_s^2(x) + (1-\alpha)p_s(x) & \text{if } s = \overline{\text{argmax}}_k \, p_k(x) \\ (\alpha \max_k p_k(x) + \epsilon)p_s(x) & \text{otherwise} \end{cases}.$$

For a fixed $x$, the inner term in (20) is minimized when $\mathbf{h}^*(x) = \frac{1}{Z(x)}\widetilde{\mathbf{p}}(x)$, where $Z(x) = \sum_k \widetilde{p}_k(x)$. This follows from the fact that for a fixed $x$, the minimizer of the inner term $\widetilde{\mathbf{p}}(x) \cdot \phi(\mathbf{h}(x))$ is the same as the the minimizer of the scaled term $\frac{1}{Z(x)}\widetilde{\mathbf{p}}(x) \cdot \phi(\mathbf{h}(x))$, and from $\phi$ being a strictly proper scoring function. This completes the proof. $\qquad\square$

## B  ADDITIONAL THEORETICAL RESULTS

### B.1  RELATIONSHIP BETWEEN CHURN AND CLASSIFICATION RISK

For certain base classifiers $\mathbf{g}$, generalizing well on "churn" can have the additional benefit of improving classification performance, as shown by the proposition below.

**Proposition 7.** *Let $(\ell, d)$ be defined as in (2) for a strictly proper scoring function $\phi$. Suppose $\phi(\mathbf{u})$ is strictly convex in $\mathbf{u}$, $\Phi$-Lipschitz w.r.t. $L_1$-norm for each $y \in [m]$, and $\|\phi(\mathbf{z})\|_\infty < B, \forall \mathbf{z} \in \Delta_m$. Let $\lambda^*$ be the optimal mixing coefficient defined in Proposition 1. Let $\Delta_C(n, \delta)$ be the churn generalization bound defined in Theorem 2. Let $\widetilde{\mathbf{h}}$ be an optimal feasible classifier in $\mathcal{H}$ and $\widehat{\mathbf{h}}$ be the classifier returned by Algorithm 1. Then for any $\delta \in (0, 1)$, w.p. $\geqslant 1 - \delta$ over draw of $S \sim D^n$:*

$$R(\widehat{\mathbf{h}}) - R(\widetilde{\mathbf{h}}) \leqslant \epsilon + \Delta_C(n, \delta) + (B + \Phi\lambda^*)\mathbf{E}_{x \sim D_\mathcal{X}}\left[\|\mathbf{p}(x) - \mathbf{g}(x)\|_1\right].$$

*Proof.* Let $\mathbf{h}^*$ be the Bayes optimal classifier, i.e. the optimal-feasible classifier over all classifiers (not just those in $\mathcal{H}$). We have:

$$\begin{aligned} & R(\widehat{\mathbf{h}}) - R(\widetilde{\mathbf{h}}) \\ & \leqslant \quad R(\widehat{\mathbf{h}}) - R(\mathbf{h}^*) \\ & = \quad \mathbf{E}_{x \sim D_\mathcal{X}}\left[\mathbf{E}_{y|x}\left[\mathbf{e}_y \cdot \phi(\widehat{\mathbf{h}}(x))\right]\right] - \mathbf{E}_{x \sim D_\mathcal{X}}\left[\mathbf{E}_{y|x}\left[\mathbf{e}_y \cdot \phi(\mathbf{h}^*(x))\right]\right] \\ & = \quad \mathbf{E}_{x \sim D_\mathcal{X}}\left[\mathbf{p}(x) \cdot \phi(\widehat{\mathbf{h}}(x))\right] - \mathbf{E}_{x \sim D_\mathcal{X}}\left[\mathbf{p}(x) \cdot \phi(\mathbf{h}^*(x))\right] \\ & = \quad \mathbf{E}_{x \sim D_\mathcal{X}}\left[\mathbf{p}(x) \cdot \left(\phi(\widehat{\mathbf{h}}(x)) - \phi(\mathbf{g}(x))\right)\right] - \mathbf{E}_{x \sim D_\mathcal{X}}\left[\mathbf{p}(x) \cdot (\phi(\mathbf{h}^*(x)) - \phi(\mathbf{g}(x)))\right] \\ & \leqslant \quad \mathbf{E}_{x \sim D_\mathcal{X}}\left[\mathbf{p}(x) \cdot \left(\phi(\widehat{\mathbf{h}}(x)) - \phi(\mathbf{g}(x))\right)\right] + \left|\mathbf{E}_{x \sim D_\mathcal{X}}\left[\sum_{y \in [m]} p_y(x)\left(\phi_y(\mathbf{h}^*(x)) - \phi_y(\mathbf{g}(x))\right)\right]\right| \\ & \leqslant \quad \mathbf{E}_{x \sim D_\mathcal{X}}\left[\mathbf{p}(x) \cdot \left(\phi(\widehat{\mathbf{h}}(x)) - \phi(\mathbf{g}(x))\right)\right] + \mathbf{E}_{x \sim D_\mathcal{X}}\left[\sum_{y \in [m]} p_y(x)|\phi_y(\mathbf{h}^*(x)) - \phi_y(\mathbf{g}(x))|\right] \\ & \leqslant \quad \mathbf{E}_{x \sim D_\mathcal{X}}\left[\mathbf{p}(x) \cdot \left(\phi(\widehat{\mathbf{h}}(x)) - \phi(\mathbf{g}(x))\right)\right] + \Phi\mathbf{E}_{x \sim D_\mathcal{X}}\left[\sum_{y \in [m]} p_y(x)\|\mathbf{h}^*(x) - \mathbf{g}(x)\|_1\right] \end{aligned}$$

$$\leqslant \quad \mathbf{E}_{x \sim D_{\mathcal{X}}} \left[ \mathbf{p}(x) \cdot \left( \phi(\widehat{\mathbf{h}}(x)) - \phi(\mathbf{g}(x)) \right) \right] + \Phi \mathbf{E}_{x \sim D_{\mathcal{X}}} \left[ \|\mathbf{h}^*(x) - \mathbf{g}(x)\|_1 \right],$$

where the second-last step follows from Jensen's inequality, and the last step uses the Lipschitz assumption on $\phi_y$.

We further have:

$$R(\widehat{\mathbf{h}}) - R(\widetilde{\mathbf{h}})$$

$$\begin{aligned}
\leqslant \quad & \mathbf{E}_{x \sim D_{\mathcal{X}}} \left[ \mathbf{g}(x) \cdot \left( \phi(\widehat{\mathbf{h}}(x)) - \phi(\mathbf{g}(x)) \right) \right] + \mathbf{E}_{x \sim D_{\mathcal{X}}} \left[ (\mathbf{p}(x) - \mathbf{g}(x)) \cdot \left( \phi(\widehat{\mathbf{h}}(x)) - \phi(\mathbf{g}(x)) \right) \right] \\
& + \Phi \mathbf{E}_{x \sim D_{\mathcal{X}}} \left[ \|\mathbf{h}^*(x) - \mathbf{g}(x)\|_1 \right] \\
\leqslant \quad & \mathbf{E}_{x \sim D_{\mathcal{X}}} \left[ \mathbf{g}(x) \cdot \left( \phi(\widehat{\mathbf{h}}(x)) - \phi(\mathbf{g}(x)) \right) \right] + \mathbf{E}_{x \sim D_{\mathcal{X}}} \left[ \|\mathbf{p}(x) - \mathbf{g}(x)\|_1 \|\phi(\widehat{\mathbf{h}}(x)) - \phi(\mathbf{g}(x))\|_\infty \right] \\
& + \Phi \mathbf{E}_{x \sim D_{\mathcal{X}}} \left[ \|\mathbf{h}^*(x) - \mathbf{g}(x)\|_1 \right] \\
\leqslant \quad & \mathbf{E}_{x \sim D_{\mathcal{X}}} \left[ \mathbf{p}(x) \cdot \left( \phi(\widehat{\mathbf{h}}(x)) - \phi(\mathbf{g}(x)) \right) \right] + B \mathbf{E}_{x \sim D_{\mathcal{X}}} \left[ \|\mathbf{p}(x) - \mathbf{g}(x)\|_1 \right] \\
& + \Phi \mathbf{E}_{x \sim D_{\mathcal{X}}} \left[ \|\mathbf{h}^*(x) - \mathbf{g}(x)\|_1 \right] \\
\leqslant \quad & \mathbf{E}_{x \sim D_{\mathcal{X}}} \left[ \mathbf{g}(x) \cdot \left( \phi(\widehat{\mathbf{h}}(x)) - \phi(\mathbf{g}(x)) \right) \right] + B \mathbf{E}_{x \sim D_{\mathcal{X}}} \left[ \|\mathbf{p}(x) - \mathbf{g}(x)\|_1 \right] \\
& + \lambda^* \Phi \mathbf{E}_{x \sim D_{\mathcal{X}}} \left[ \|\mathbf{p}(x) - \mathbf{g}(x)\|_1 \right] \\
= \quad & C(\widehat{\mathbf{h}}) + (B + \lambda^* \Phi) \mathbf{E}_{x \sim D_{\mathcal{X}}} \left[ \|\mathbf{p}(x) - \mathbf{g}(x)\|_1 \right],
\end{aligned}$$

where the second step applies Hölder's inequality to each $x$, the third step follows from the boundedness assumption on $\phi$, and the fourth step uses the characterization $\mathbf{h}^*(x) = \lambda^* \mathbf{p}(x) + (1 - \lambda^*)\mathbf{g}(x)$, for $\lambda^* \in [0, 1]$ from Proposition 1. Applying Theorem 2 to the churn $C(\widehat{\mathbf{h}})$ completes the proof. $\quad \square$

This result bounds the excess classification risk in terms of the churn generalization bound and the expected difference between the base classifier $\mathbf{g}$ and the underlying class probability function $\mathbf{p}$. When the base classifier is close to $\mathbf{p}$, low values of $\Delta_C(n, \delta)$ result in low classification risk.

### B.2 GENERALIZATION BOUND FOR CLASSIFICATION RISK

As a follow-up to Proposition 3, we also provide generalization bounds for the classification risk in terms of the *empirical variance* of the loss values based on a result from (Menon et al., 2020, Proposition 2).

**Proposition 8** (Generalization bound for classification risk). *Let the scoring function $\phi : \Delta_m \to \mathbb{R}_+^m$ be bounded. Let $\mathcal{V}_\phi \subseteq \mathbb{R}^{\mathcal{X}}$ denote the class of loss functions $v(x, y) = \ell_\phi(y, \mathbf{h}(x)) = \phi_y(\mathbf{h}(x))$ induced by classifiers $\mathbf{h} \in \mathcal{H}$. Let $\mathcal{M}_n^R = \mathcal{N}_\infty(\frac{1}{n}, \mathcal{V}_\phi, 2n)$ denote the uniform $L_\infty$ covering number for $\mathcal{V}_\phi$. Fix $\delta \in (0, 1)$. Then with probability $\geqslant 1 - \delta$ over draw of $S \sim D^n$, for any $\mathbf{h} \in \mathcal{H}$:*

$$R(\mathbf{h}) \leqslant \widehat{R}(\mathbf{h}) + \mathcal{O}\left( \sqrt{\mathbb{V}_n^R(\mathbf{h}) \frac{\log(\mathcal{M}_n^R/\delta)}{n}} + \frac{\log(\mathcal{M}_n^R/\delta)}{n} \right).$$

*where $\mathbb{V}_n^R(\mathbf{h})$ denotes the empirical variance of the loss computed on $n$ examples $\{\phi_{y_i}(\mathbf{h}(x_i))\}_{i=1}^n$.*

## C DEFINITIONS OF NETWORK ARCHITECTURES USED

### C.1 FULLY CONNECTED NETWORK

FCN-x refers to the following model with size set to "x". In other words, it's a simple fully connected network with one hidden layer with x units.

```
def get_fcn(n_columns,
            num_classes=10,
            size=100,
            weight_init=None):
  model = None
```

```
model = tf.keras.Sequential([
    tf.keras.layers.Input(shape=(n_columns,)),
    tf.keras.layers.Dense(size, activation=tf.nn.relu),
    tf.keras.layers.Dense(num_classes, activation="softmax"),
])

model.compile(
    optimizer=tf.keras.optimizers.Adam(),
    loss=tf.keras.losses.CategoricalCrossentropy(),
    metrics=[tf.keras.metrics.categorical_accuracy])
return model
```

## C.2 CONVOLUTIONAL NETWORK

Convnet-x refers to the following model with size set to "x". Convnet-1 is based on the lenet5 architecture LeCun et al. (1998).

```
def get_convnet(
    input_shape=(28, 28, 3),
    size=1,
    num_classes=2,
    weight_init=None):

    model = tf.keras.Sequential()
    model.add(
        tf.keras.layers.Conv2D(
            filters=16 * size,
            kernel_size=(5, 5),
            padding="same",
            activation="relu",
            input_shape=input_shape))
    model.add(tf.keras.layers.MaxPool2D(strides=2))
    model.add(
        tf.keras.layers.Conv2D(
            filters=24 * size,
            kernel_size=(5, 5),
            padding="valid",
            activation="relu"))
    model.add(tf.keras.layers.MaxPool2D(strides=2))
    model.add(tf.keras.layers.Flatten())
    model.add(tf.keras.layers.Dense(128 * size, activation="relu"))
    model.add(tf.keras.layers.Dense(84, activation="relu"))
    model.add(tf.keras.layers.Dense(num_classes, activation="softmax"))

    model.compile(
        optimizer=tf.keras.optimizers.Adam(),
        loss=tf.keras.losses.CategoricalCrossentropy(),
        metrics=[tf.keras.metrics.categorical_accuracy])

    return model
```

## C.3 TRANSFORMER

Transformer-x refers to the following with size set to "x". It is based on keras tutorial on text classification (https://keras.io/examples/nlp/text_classification_with_transformer/ licensed under the Apache License, Version 2.0).

```python
def get_transformer(maxlen,
                    size=1,
                    num_classes=2,
                    weight_init=None):
  model = None

  class TransformerBlock(tf.keras.layers.Layer):

    def __init__(self,
                 embed_dim,
                 num_heads,
                 ff_dim,
                 rate=0.1,
                 weight_init=None):
      super(TransformerBlock, self).__init__()
      self.att = tf.keras.layers.MultiHeadAttention(
          num_heads=num_heads, key_dim=embed_dim)
      self.ffn = tf.keras.Sequential([
            tf.keras.layers.Dense(ff_dim, activation="relu"),
            tf.keras.layers.Dense(embed_dim),
        ])
      self.layernorm1 = tf.keras.layers.LayerNormalization(epsilon=1e-6)
      self.layernorm2 = tf.keras.layers.LayerNormalization(epsilon=1e-6)

    def call(self, inputs, training):
      attn_output = self.att(inputs, inputs)
      #attn_output = self.dropout1(attn_output, training=training)
      out1 = self.layernorm1(inputs + attn_output)
      ffn_output = self.ffn(out1)
      return self.layernorm2(out1 + ffn_output)

  class TokenAndPositionEmbedding(tf.keras.layers.Layer):

    def __init__(
        self,
        maxlen,
        vocab_size,
        embed_dim,
    ):
      super(TokenAndPositionEmbedding, self).__init__()

      self.token_emb = tf.keras.layers.Embedding(
          input_dim=vocab_size, output_dim=embed_dim)
      self.pos_emb = tf.keras.layers.Embedding(
          input_dim=maxlen, output_dim=embed_dim)

    def call(self, x):
      maxlen = tf.shape(x)[-1]
      positions = tf.range(start=0, limit=maxlen, delta=1)
      positions = self.pos_emb(positions)
      x = self.token_emb(x)
      return x + positions

  embed_dim = 32 * size  # Embedding size for each token
  num_heads = 2 * size  # Number of attention heads
  ff_dim = 32 * size  # Hidden layer size in feed forward network inside transformer

  inputs = tf.keras.layers.Input(shape=(maxlen,))
```

```
embedding_layer = TokenAndPositionEmbedding(maxlen, 20000, embed_dim)
x = embedding_layer(inputs)
transformer_block = TransformerBlock(embed_dim, num_heads, ff_dim,
                                     weight_init)
x = transformer_block(x)
x = tf.keras.layers.GlobalAveragePooling1D()(x)

outputs = tf.keras.layers.Dense(num_classes, activation="softmax")(x)

model = tf.keras.Model(inputs=inputs, outputs=outputs)
model.compile(
    optimizer=tf.keras.optimizers.Adam(),
    loss=tf.keras.losses.CategoricalCrossentropy(),
    metrics=[tf.keras.metrics.categorical_accuracy])
return model
```

## D  MODEL TRAINING CODE

```
def model_trainer(get_model,
                  X_train,
                  y_train,
                  X_test,
                  y_test,
                  weight_init=None,
                  validation_data=None,
                  warm=True,
                  mixup_alpha=-1,
                  codistill_alpha=-1,
                  distill_alpha=-1,
                  anchor_alpha=-1,
                  anchor_eps=-1):
  model = get_model()
  if weight_init is not None and warm:
    model.set_weights(weight_init)
  if FLAGS.loss == "squared":
    model.compile(
        optimizer=tf.keras.optimizers.Adam(),
        loss=tf.keras.losses.MeanSquaredError(),
        metrics=[tf.keras.metrics.categorical_accuracy])
  callback = tf.keras.callbacks.EarlyStopping(monitor="val_loss", patience=3)
  history = None

  if distill_alpha >= 0:
    original_model = get_model()
    original_model.set_weights(weight_init)
    y_pred = original_model.predict(X_train)
    y_use = distill_alpha * y_pred + (1 - distill_alpha) * y_train
    history = model.fit(
        x=X_train,
        y=y_use,
        epochs=FLAGS.n_epochs,
        callbacks=[callback],
        validation_data=validation_data)
  elif anchor_alpha >= 0 and anchor_eps >= 0:
    original_model = get_model()
    original_model.set_weights(weight_init)
```

```
        y_pred = original_model.predict(X_train)
        y_pred_hard = np.argmax(y_pred, axis=1)
        y_hard = np.argmax(y_train, axis=1)
        correct = (y_pred_hard == y_hard)
        correct = np.tile(correct, (y_train.shape[1], 1))
        correct = np.transpose(correct)
        correct = correct.reshape(y_train.shape)
        y_use = np.where(correct,
                         anchor_alpha * y_pred + (1 - anchor_alpha) * y_train,
                         y_train * anchor_eps)
        history = model.fit(
            x=X_train,
            y=y_use,
            epochs=FLAGS.n_epochs,
            callbacks=[callback],
            validation_data=validation_data)
    elif mixup_alpha >= 0:
      training_generator = deep_utils.MixupGenerator(
          X_train, y_train, alpha=mixup_alpha)()
      history = model.fit(
          x=training_generator,
          validation_data=validation_data,
          steps_per_epoch=int(X_train.shape[0] / 32),
          epochs=FLAGS.n_epochs,
          callbacks=[callback])
    elif codistill_alpha >= 0:
      teacher_model = get_model()
      if weight_init is not None and warm:
        teacher_model.set_weights(weight_init)
      val_losses = []
      optimizer = tf.keras.optimizers.Adam()
      global_step = 0
      alpha = 0
      codistillation_warmup_steps = 0

      for epoch in range(FLAGS.n_epochs):
        X_train_, y_train_ = sklearn.utils.shuffle(X_train, y_train)
        batch_size = 32
        for i in range(int(X_train_.shape[0] / batch_size)):
          if global_step >= codistillation_warmup_steps:
            alpha = codistill_alpha
          else:
            alpha = 0.
          with tf.GradientTape() as tape:
            X_batch = X_train_[i * 32:(i + 1) * 32, :]
            y_batch = y_train_[i * 32:(i + 1) * 32, :]
            prob_student = model(X_batch, training=True)
            prob_teacher = teacher_model(X_batch, training=True)
            loss = deep_utils.compute_loss(prob_student, prob_teacher, y_batch,
                                           alpha)
            trainable_weights = model.trainable_weights + teacher_model.trainable_weigh
            grads = tape.gradient(loss, trainable_weights)
            optimizer.apply_gradients(zip(grads, trainable_weights))
          global_step += 1
        val_preds = model.predict(validation_data[0])
        val_loss = np.sum(
            deep_utils.cross_entropy(validation_data[1].astype("float32"),
                                     val_preds))
        val_losses.append(val_loss)
```

```
        if len(val_losses) > 3 and min(val_losses[-3:]) > val_losses[-4]:
          break

  else:
    history = model.fit(
        X_train,
        y_train,
        epochs=FLAGS.n_epochs,
        callbacks=[callback],
        validation_data=validation_data)

  y_pred_train = model.predict(X_train)
  y_pred_test = model.predict(X_test)
  return y_pred_train, y_pred_test, model.get_weights()
```

## E   ADDITIONAL EXPERIMENTAL RESULTS

### E.1   ADDITIONAL OPENML RESULTS

#### E.1.1   INITIAL SAMPLE 100, BATCH SIZE 1000, VALIDATION SIZE 100

In Tables 3 and 4, we show the churn at cold accuracy metric across network sizes (fcn-10, fcn-100, fcn-1000, fcn-10000, fcn-100000). Table 5 shows the standard error bars. They are obtained by fixing the dataset and model, and taking the 100 accuracy and churn results from each baseline and calculating the standard error, which is the standard deviation of the mean. We then report the average standard error across the baselines We see that distillation is the best $52\%$ of the time.

#### E.1.2   INITIAL SAMPLE 1000, BATCH SIZE 1000, VALIDATION SIZE 100

In Tables 6 and 7, we show the churn at cold accuracy metric across network sizes (fcn-10, fcn-100, fcn-1000, fcn-10000, fcn-100000). We see that distillation consistently performs strongly across datasets and sizes of networks. Table 8 shows the standard error bars. We see that distillation is the best $84\%$ of the time.

### E.2   ADDITIONAL MNIST VARIANT RESULTS

#### E.2.1   INITIAL SAMPLE SIZE 100, BATCH SIZE 1000, VALIDATION SIZE 100

We show full results in Table 9. We see that distillation is the best for 24 out of the 50 combinations of dataset and network. Error bands can be found in Table 10.

#### E.2.2   INITIAL SAMPLE SIZE 1000, BATCH SIZE 1000, VALIDATION SIZE 100

We show full results in Table 11. We see that distillation is the best for 42 out of the 50 combinations of dataset and network. Error bands can be found in Table 12.

#### E.2.3   INITIAL SAMPLE SIZE 10000, BATCH SIZE 1000, VALIDATION SIZE 1000

We show full results in Table 13. We see that in this situation, label smoothing starts becoming competitive with distillation with either of them being the best. Distillation is the best for 32 out of the 50 combinations of dataset and network, and losing marginally to label smoothing in other cases. See Table 14 for error bands.

### E.3   ADDITIONAL SVHN AND CIFAR RESULTS

#### E.3.1   INITIAL SAMPLE 100, BATCH SIZE 1000, VALIDATION SIZE 100

Results are in Table 15, where we see that distillation is best on 8 out of 10 combinations of dataset and network. Error bands can be found in Table 16.

### E.3.2 INITIAL SAMPLE 1000, BATCH SIZE 1000, VALIDATION SIZE 100

The results can be found in Table 17. We include the error bands here in Table 18. Distillation is best in all combinations.

### E.3.3 INITIAL SAMPLE 10000, BATCH SIZE 1000, VALIDATION SIZE 1000

Results are in Table 19, where we see that distillation is best on all combinations of dataset and network. Error bands can be found in Table 20.

### E.4 ADDITIONAL CELEBA RESULTS

### E.4.1 INITIAL SAMPLE 100, BATCH SIZE 1000, VALIDATION SIZE 100

Tables 21, 22, 23, and 24 show the performance of CelebA tasks when we instead use an initial sample size of 100. We see that across the 200 combinations of task and network, distillation is the best 192 of time, or 96% of the time. The error bands can be found in Table 25.

### E.4.2 INITIAL SAMPLE 1000, BATCH SIZE 1000, VALIDATION SIZE 100

We show some additional CelebA results for initial sample 1000 and batch size 1000 in Tables 26, 27, 28, and 29 which show performance for each dataset across convnet-1, convnet-2, convnet-4, convent-8, convnet-16. This gives us $40 \cdot 5 = 200$ results, of which distillation performs the best 158 out of those settings, or 79% of the time. The error bands can be found in Table 30.

### E.4.3 INITIAL SAMPLE SIZE 10000, BATCH SIZE 1000, VALIDATION SIZE 1000

Tables 31, 32, 33, and 34 show the performance of CelebA tasks when we instead use an initial sample size of 10000. We see that across the 200 combinations of task and network, distillation is the best 183 of time, or 91.5% of the time. The error bands can be found in Table 35.

### E.5 CIFAR10 AND CIFAR100 ON RESNET

Results can be found in Table 36. We see that distillation outperforms in every case.

### E.6 ADDITIONAL IMDB RESULTS

In Table 37, we show the results for the IMDB dataset and transformer networks for initial batch sizes of 100, 1000 and 10000 with the batch size fixed at 1000. The error bands can be found in Table 38. We see that for initial sample size of 100, distillation performs poorly for the smaller networks as the process of distillation hurts the performance with a weak teacher trained on only 100 examples, but performs well for the larger networks. For initial sample size of 1000 and 10000, distillation is the clear winner losing in only one instance. We show the full Pareto frontiers and cost curves in Figure 4.

| Dataset | network | cold | warm | s-perturb | mixup | ls | co-dist | anchor | distill |
|---|---|---|---|---|---|---|---|---|---|
| adult | fcn-10 | 12.75 | N/A | 11.55 | 12.3 | N/A | **10.85** | N/A | N/A |
| | fcn-100 | 11.8 | N/A | 10.86 | 11.14 | 11.11 | **10.0** | N/A | N/A |
| | fcn-1000 | 11.78 | 12.2 | 11.45 | 11.84 | 12.39 | 10.67 | 12.16 | **8.55** |
| | fcn-10000 | 14.03 | 13.18 | 13.28 | 13.1 | 12.8 | 13.1 | 14.22 | **9.7** |
| | fcn-100000 | 14.13 | 13.28 | 13.45 | 13.49 | 13.74 | 13.32 | N/A | **8.43** |
| bank | fcn-10 | 10.24 | 9.03 | 9.17 | 9.07 | 8.48 | 8.17 | **6.28** | 6.72 |
| | fcn-100 | 8.19 | N/A | 6.79 | 6.81 | 8.28 | 6.45 | 7.85 | **3.09** |
| | fcn-1000 | 10.1 | 10.71 | 9.96 | 9.59 | 10.29 | 8.93 | 10.38 | **4.35** |
| | fcn-10000 | 12.81 | N/A | 11.04 | 11.76 | 12.97 | 10.86 | 11.42 | **7.4** |
| | fcn-100000 | 10.78 | 10.97 | 9.79 | 9.78 | 8.36 | 9.21 | 10.33 | **5.48** |
| COMPAS | fcn-10 | 21.71 | 18.29 | 18.19 | 17.98 | 19.06 | **14.32** | 20.76 | N/A |
| | fcn-100 | 17.59 | 16.34 | 16.42 | 16.17 | 16.85 | 14.44 | 14.03 | **12.0** |
| | fcn-1000 | 19.08 | 18.37 | 17.69 | 17.57 | 18.64 | 16.78 | N/A | **11.76** |
| | fcn-10000 | 23.13 | 23.02 | 22.23 | 21.83 | N/A | **21.6** | N/A | N/A |
| | fcn-100000 | 24.79 | N/A | **24.12** | 24.2 | N/A | 24.84 | N/A | N/A |
| magic04 | fcn-10 | 30.42 | 25.96 | 27.12 | 26.58 | 26.88 | 25.71 | 25.88 | **23.58** |
| | fcn-100 | 32.15 | N/A | 28.3 | 28.22 | N/A | **25.61** | N/A | N/A |
| | fcn-1000 | 32.35 | N/A | 29.75 | 29.64 | N/A | 28.59 | 31.94 | **20.93** |
| | fcn-10000 | 30.84 | 31.0 | 28.5 | 29.57 | 29.28 | **27.07** | 29.09 | 27.49 |
| | fcn-100000 | 27.56 | 27.75 | 25.81 | 26.37 | 25.25 | 25.12 | 26.73 | **23.65** |
| phonemes | fcn-10 | 18.64 | 16.77 | **16.73** | N/A | N/A | N/A | 18.0 | N/A |
| | fcn-100 | 18.15 | N/A | 17.33 | N/A | N/A | N/A | N/A | **16.23** |
| | fcn-1000 | 18.97 | 19.36 | 18.25 | N/A | N/A | 18.46 | 20.76 | **13.24** |
| | fcn-10000 | 20.6 | 20.56 | 20.32 | 19.68 | 19.71 | 20.35 | 22.28 | **16.5** |
| | fcn-100000 | 19.66 | 19.9 | 18.97 | 18.3 | 18.58 | 18.93 | 20.93 | **12.82** |
| electricity | fcn-10 | 38.8 | 39.6 | 36.8 | 35.8 | 39.8 | 33.8 | N/A | **30.4** |
| | fcn-100 | **33.45** | N/A | N/A | N/A | N/A | N/A | N/A | N/A |
| | fcn-1000 | 40.29 | N/A | 33.43 | 29.76 | 33.14 | 35.14 | 42.52 | **27.52** |
| | fcn-10000 | 35.78 | N/A | 35.81 | 28.63 | 32.81 | 34.11 | 38.63 | **26.33** |
| | fcn-100000 | 33.92 | N/A | N/A | **32.96** | 36.46 | 37.71 | N/A | N/A |
| eeg | fcn-10 | **45.92** | N/A | N/A | N/A | N/A | N/A | N/A | N/A |
| | fcn-100 | 52.8 | 47.31 | 46.88 | 47.56 | N/A | 41.82 | 32.66 | **25.33** |
| | fcn-1000 | 54.73 | N/A | 44.22 | 48.4 | 57.85 | N/A | 27.38 | **2.12** |
| | fcn-10000 | **50.59** | N/A | N/A | N/A | N/A | N/A | N/A | N/A |
| | fcn-100000 | 46.57 | 50.91 | 44.54 | 43.34 | 40.63 | 44.43 | 45.86 | **29.3** |

Table 3: Results for OpenML datasets for initial sample size 100 under churn at cold accuracy metric across different sizes of fully connected networks. Part 1 of 2.

| Dataset | network | cold | warm | s-perturb | mixup | ls | co-dist | anchor | distill |
|---|---|---|---|---|---|---|---|---|---|
| churn | fcn-10 | 19.57 | N/A | 16.9 | 18.82 | N/A | N/A | N/A | **7.61** |
| | fcn-100 | 23.51 | 16.9 | 15.33 | 17.67 | N/A | 13.66 | 15.0 | **3.16** |
| | fcn-1000 | 22.2 | N/A | 18.2 | 18.6 | N/A | 14.68 | 14.5 | **4.48** |
| | fcn-10000 | 24.8 | 24.1 | 22.33 | 24.74 | N/A | 20.79 | **18.97** | 20.69 |
| | fcn-100000 | 23.15 | 18.68 | 18.52 | 16.93 | 20.82 | **16.85** | 18.88 | 18.03 |
| elevators | fcn-10 | 35.52 | N/A | 32.43 | 30.78 | N/A | **29.12** | N/A | 29.5 |
| | fcn-100 | 34.06 | N/A | 32.25 | 30.54 | N/A | 29.27 | 23.89 | **21.37** |
| | fcn-1000 | 39.06 | 39.64 | 35.92 | 37.35 | N/A | 34.22 | 30.56 | **21.32** |
| | fcn-10000 | 39.8 | 39.68 | 38.0 | 35.04 | 41.94 | 35.19 | 39.02 | **33.96** |
| | fcn-100000 | **33.84** | N/A | 34.13 | N/A | N/A | 35.67 | N/A | N/A |
| pollen | fcn-10 | 48.06 | 36.85 | 46.15 | 34.7 | 36.74 | 33.85 | 18.54 | **2.07** |
| | fcn-100 | 46.97 | N/A | 44.94 | 41.89 | 41.4 | 40.91 | **28.01** | 36.61 |
| | fcn-1000 | 47.06 | N/A | N/A | N/A | N/A | N/A | 36.93 | **5.37** |
| | fcn-10000 | 45.85 | N/A | 45.65 | 46.06 | 47.11 | 45.81 | **39.53** | N/A |
| | fcn-100000 | 45.77 | N/A | 46.53 | 48.12 | N/A | 48.91 | 43.12 | **40.57** |
| phishing | fcn-10 | 9.74 | N/A | 8.97 | 8.76 | 9.18 | 8.65 | N/A | **8.12** |
| | fcn-100 | 7.44 | N/A | 7.48 | 6.91 | N/A | N/A | 7.28 | **6.69** |
| | fcn-1000 | 8.25 | N/A | N/A | 7.9 | **7.85** | 8.11 | N/A | N/A |
| | fcn-10000 | 9.21 | 9.45 | 8.91 | 8.7 | 8.56 | 8.61 | 8.53 | **6.48** |
| | fcn-100000 | 10.2 | N/A | 9.95 | 9.74 | **8.85** | 9.76 | 9.89 | N/A |
| wilt | fcn-10 | 7.44 | N/A | **6.48** | N/A | N/A | N/A | N/A | N/A |
| | fcn-100 | 3.85 | N/A | 3.39 | 3.15 | N/A | **2.42** | 3.81 | 3.05 |
| | fcn-1000 | 6.45 | 4.98 | 3.83 | 3.41 | N/A | 0.88 | 1.61 | **0.15** |
| | fcn-10000 | 5.08 | 4.22 | 3.21 | 1.58 | N/A | 0.56 | 1.89 | **0.01** |
| | fcn-100000 | 7.69 | 3.98 | 4.67 | 3.45 | 4.19 | 3.11 | 3.7 | **0.22** |
| letters | fcn-10 | 91.44 | 91.67 | 91.33 | N/A | 92.0 | 90.89 | 92.22 | **90.56** |
| | fcn-100 | **63.1** | N/A | 63.6 | N/A | 63.5 | N/A | N/A | N/A |
| | fcn-1000 | 59.6 | 60.1 | 59.1 | 58.57 | 58.9 | N/A | 59.13 | **54.43** |
| | fcn-10000 | 61.67 | N/A | N/A | N/A | **60.05** | N/A | N/A | N/A |
| | fcn-100000 | 61.62 | N/A | 61.78 | **60.53** | 60.78 | 61.88 | 61.72 | N/A |

Table 4: Results for OpenML datasets for initial sample size 100 under churn at cold accuracy metric across different sizes of fully connected networks. Part 2 of 2.

| | fcn-10 | | fcn-100 | | fcn-1000 | | fcn-10000 | | fcn-100000 | |
|---|---|---|---|---|---|---|---|---|---|---|
| Dataset | Error | Churn | Error | Churn | Error | Churn | Error | Churn | Error | Churn |
| adult | 0.35 | 0.36 | 0.32 | 0.39 | 0.38 | 0.41 | 0.43 | 0.6 | 0.53 | 0.61 |
| bank | 0.25 | 0.64 | 0.25 | 0.48 | 0.42 | 0.78 | 0.57 | 0.96 | 0.53 | 0.81 |
| COMPAS | 0.48 | 0.77 | 0.45 | 0.68 | 0.47 | 0.67 | 0.54 | 0.91 | 0.52 | 0.94 |
| magic04 | 0.54 | 0.99 | 0.63 | 1.05 | 0.88 | 1.37 | 0.9 | 1.68 | 0.71 | 1.4 |
| phonemes | 0.42 | 0.54 | 0.36 | 0.48 | 0.41 | 0.57 | 0.41 | 0.66 | 0.44 | 0.74 |
| electricity | 0.94 | 2.06 | 0.67 | 1.49 | 0.55 | 1.43 | 0.62 | 1.62 | 0.62 | 1.82 |
| eeg | 0.65 | 3.23 | 0.59 | 4.59 | 0.59 | 4.71 | 0.59 | 4.69 | 0.47 | 4.64 |
| churn | 1.17 | 1.49 | 1.72 | 2.29 | 2.02 | 2.93 | 2.13 | 3.22 | 1.34 | 2.72 |
| elevators | 0.54 | 1.06 | 0.74 | 1.48 | 0.93 | 1.89 | 0.97 | 1.97 | 0.81 | 1.77 |
| pollen | 0.51 | 0.75 | 0.46 | 0.89 | 0.44 | 1.16 | 0.45 | 1.25 | 0.42 | 1.4 |
| phishing | 0.27 | 0.32 | 0.28 | 0.27 | 0.31 | 0.31 | 0.37 | 0.44 | 0.45 | 0.51 |
| wilt | 0.45 | 0.62 | 0.57 | 0.83 | 1.12 | 1.43 | 1.2 | 1.41 | 0.94 | 1.63 |
| letters | 0.62 | 0.78 | 0.53 | 0.69 | 0.53 | 0.66 | 0.51 | 0.69 | 0.51 | 0.59 |

Table 5: OpenML Error Bands for initial sample size 100: Average standard errors for error and churn across baselines for each dataset and network across 100 runs.

| Dataset | network | cold | warm | s-perturb | mixup | ls | co-dist | anchor | distill |
|---|---|---|---|---|---|---|---|---|---|
| adult | fcn-10 | 4.96 | N/A | **4.58** | N/A | N/A | N/A | N/A | N/A |
| | fcn-100 | 5.49 | N/A | 4.87 | N/A | 5.3 | 4.39 | 4.51 | **3.53** |
| | fcn-1000 | 6.27 | N/A | 6.05 | 6.57 | N/A | 5.78 | 6.62 | **4.39** |
| | fcn-10000 | 8.8 | N/A | 8.71 | 8.72 | N/A | N/A | N/A | **4.68** |
| | fcn-100000 | 10.36 | 9.47 | 9.38 | 9.28 | 9.29 | 9.1 | N/A | **3.13** |
| bank | fcn-10 | 4.29 | N/A | 3.99 | 4.23 | 3.35 | 2.57 | 4.19 | **2.39** |
| | fcn-100 | 6.23 | N/A | 5.32 | 5.72 | 6.32 | 4.87 | N/A | **1.48** |
| | fcn-1000 | 10.04 | 8.43 | 7.8 | 8.25 | 8.89 | 7.55 | 8.77 | **5.58** |
| | fcn-10000 | 10.04 | 9.19 | 9.15 | 8.72 | 8.75 | 8.68 | 9.25 | **3.75** |
| | fcn-100000 | 7.81 | 8.02 | 7.86 | 8.28 | **6.86** | 7.35 | 8.51 | 7.29 |
| magic04 | fcn-10 | 17.97 | 13.42 | 12.59 | 12.95 | 13.69 | 11.37 | 13.04 | **5.34** |
| | fcn-100 | 22.4 | 21.2 | 19.47 | 20.4 | N/A | 18.9 | 20.7 | **10.94** |
| | fcn-1000 | 27.56 | 27.41 | 24.37 | 24.68 | 27.79 | 23.67 | 25.22 | **18.51** |
| | fcn-10000 | 26.83 | 23.97 | 23.01 | 23.19 | 25.72 | 22.85 | 24.0 | **19.97** |
| | fcn-100000 | 18.04 | 18.89 | 16.15 | 17.49 | 16.08 | 16.68 | 17.76 | **8.73** |
| phonemes | fcn-10 | 12.05 | 10.03 | 10.41 | N/A | N/A | 10.1 | 10.5 | **7.26** |
| | fcn-100 | 9.37 | 8.79 | 8.69 | N/A | N/A | 8.91 | 9.28 | **7.11** |
| | fcn-1000 | 10.45 | 10.66 | 10.09 | N/A | 9.02 | 9.3 | 11.14 | **7.4** |
| | fcn-10000 | 13.04 | 13.16 | 13.26 | N/A | 12.62 | 12.45 | 14.3 | **8.14** |
| | fcn-100000 | 14.08 | 14.1 | 14.0 | 13.16 | 12.97 | 12.91 | 14.79 | **8.58** |
| electricity | fcn-10 | 16.27 | 14.56 | 14.97 | 13.54 | 14.24 | 14.16 | 15.77 | **10.36** |
| | fcn-100 | 17.11 | 15.42 | 15.73 | 14.63 | 15.39 | **13.98** | 17.16 | 15.25 |
| | fcn-1000 | 18.16 | 17.53 | 17.23 | 15.69 | 16.19 | 14.94 | 18.22 | **8.99** |
| | fcn-10000 | 19.94 | 19.47 | 18.64 | 17.38 | 18.15 | 17.01 | 20.53 | **10.18** |
| | fcn-100000 | 20.68 | 20.23 | 19.14 | 18.2 | 19.44 | 18.47 | 19.53 | **5.21** |
| eeg | fcn-10 | 47.44 | 35.23 | 36.92 | 33.12 | 38.18 | 33.34 | 28.04 | **13.54** |
| | fcn-100 | 41.01 | N/A | N/A | 39.82 | N/A | 44.6 | **33.45** | N/A |
| | fcn-1000 | 48.02 | 42.96 | 42.04 | 39.98 | 49.98 | 54.99 | 26.99 | **2.0** |
| | fcn-10000 | 41.02 | 50.38 | 44.65 | 37.09 | 49.09 | 38.15 | 30.02 | **1.01** |
| | fcn-100000 | 27.73 | 20.25 | 19.75 | 19.67 | 24.75 | 19.72 | 22.67 | **17.89** |

Table 6: Results for OpenML datasets with initial sample size 1000 under churn at cold accuracy metric across different sizes of fully connected networks. Part 1 of 2.

| Dataset | network | cold | warm | s-perturb | mixup | ls | co-dist | anchor | distill |
|---------|---------|------|------|-----------|-------|-----|---------|--------|---------|
| churn | fcn-10 | 21.61 | 17.23 | 17.85 | 15.69 | N/A | **14.79** | 17.52 | 15.5 |
| | fcn-100 | 26.42 | N/A | 20.34 | 21.44 | 26.07 | 16.68 | 18.32 | **4.13** |
| | fcn-1000 | 27.15 | 25.58 | 22.19 | 20.49 | N/A | 18.71 | 17.59 | **5.51** |
| | fcn-10000 | 27.84 | 29.72 | 22.21 | 21.26 | N/A | 20.22 | 22.92 | **18.39** |
| | fcn-100000 | 14.51 | 11.64 | 11.27 | 10.72 | 10.96 | 11.0 | 11.53 | **8.57** |
| elevators | fcn-10 | 24.34 | 19.75 | 20.38 | 18.83 | 19.88 | 16.72 | 21.77 | **11.64** |
| | fcn-100 | 30.83 | 29.56 | 29.18 | 28.81 | 29.97 | 26.77 | 30.48 | **13.68** |
| | fcn-1000 | 33.34 | 35.87 | 30.41 | 31.47 | 32.91 | 30.53 | 34.38 | **10.44** |
| | fcn-10000 | 34.79 | 34.36 | 29.77 | 30.9 | 32.76 | 28.95 | 31.85 | **11.29** |
| | fcn-100000 | 23.23 | N/A | 22.86 | N/A | **22.57** | 24.38 | N/A | N/A |
| pollen | fcn-10 | 46.05 | N/A | **23.42** | N/A | 31.2 | N/A | 33.21 | N/A |
| | fcn-100 | 42.82 | 35.15 | 36.11 | 33.8 | 35.65 | 34.58 | 39.95 | **10.93** |
| | fcn-1000 | 44.03 | N/A | 42.63 | 44.6 | 42.06 | 41.78 | 40.44 | **35.15** |
| | fcn-10000 | 45.94 | N/A | 41.64 | 41.04 | 40.78 | 42.25 | 41.95 | **6.74** |
| | fcn-100000 | 45.72 | N/A | 43.77 | 43.16 | 43.31 | 41.33 | 43.03 | **13.41** |
| phishing | fcn-10 | **3.25** | N/A | N/A | N/A | N/A | N/A | N/A | N/A |
| | fcn-100 | 3.95 | N/A | 3.68 | 3.42 | 3.2 | 3.21 | 3.45 | **2.52** |
| | fcn-1000 | 4.43 | 4.2 | 3.97 | 4.01 | 4.1 | 3.74 | 4.08 | **2.91** |
| | fcn-10000 | 5.29 | 5.2 | 5.15 | 5.09 | 4.69 | 5.07 | 5.07 | **4.53** |
| | fcn-100000 | 5.93 | 5.79 | 5.38 | 5.51 | 5.03 | 5.12 | 5.38 | **3.49** |
| wilt | fcn-10 | 4.1 | 2.61 | 2.87 | 2.76 | N/A | 2.14 | 2.9 | **1.53** |
| | fcn-100 | 4.5 | 4.68 | 3.89 | 3.96 | 4.93 | 3.49 | 3.96 | **3.02** |
| | fcn-1000 | 9.55 | 7.27 | 7.27 | 6.67 | N/A | 7.0 | 7.58 | **4.93** |
| | fcn-10000 | 11.56 | 10.07 | 9.67 | 9.51 | N/A | **9.2** | 10.13 | 9.68 |
| | fcn-100000 | 5.42 | 5.22 | 5.0 | 4.53 | 4.63 | 4.43 | 4.64 | **3.43** |
| letters | fcn-10 | 38.44 | N/A | 25.92 | N/A | N/A | N/A | N/A | **23.56** |
| | fcn-100 | 22.74 | 20.92 | 21.31 | N/A | N/A | N/A | 20.81 | **19.05** |
| | fcn-1000 | 23.01 | 23.15 | 23.47 | 23.86 | 23.06 | 23.44 | 22.04 | **16.92** |
| | fcn-10000 | 27.44 | 26.48 | 26.29 | 26.1 | 24.79 | 24.86 | 24.73 | **18.97** |
| | fcn-100000 | 30.33 | 29.57 | 28.76 | 28.23 | 26.96 | 27.89 | 27.82 | **20.71** |

Table 7: Results for OpenML datasets with initial sample size 1000 under churn at cold accuracy metric across different sizes of fully connected networks. Part 2 of 2.

| | fcn-10 | | fcn-100 | | fcn-1000 | | fcn-10000 | | fcn-100000 | |
|---------|-------|-------|-------|-------|-------|-------|-------|-------|-------|-------|
| Dataset | Error | Churn | Error | Churn | Error | Churn | Error | Churn | Error | Churn |
| adult | 0.35 | 0.22 | 0.36 | 0.28 | 0.35 | 0.35 | 0.44 | 0.51 | 0.46 | 0.55 |
| bank | 0.2 | 0.26 | 0.28 | 0.38 | 0.37 | 0.6 | 0.52 | 0.77 | 0.51 | 0.69 |
| COMPAS | 0.45 | 0.36 | 0.44 | 0.46 | 0.48 | 0.66 | 0.55 | 0.73 | 0.56 | 0.76 |
| magic04 | 0.45 | 0.67 | 0.6 | 1.09 | 0.85 | 1.58 | 0.81 | 1.51 | 0.67 | 0.99 |
| phonemes | 0.43 | 0.37 | 0.38 | 0.33 | 0.41 | 0.43 | 0.41 | 0.54 | 0.42 | 0.55 |
| electricity | 0.59 | 0.83 | 0.47 | 0.84 | 0.51 | 1.04 | 0.58 | 1.31 | 0.57 | 1.38 |
| eeg | 0.63 | 2.89 | 0.59 | 4.57 | 0.59 | 4.71 | 0.59 | 4.62 | 0.4 | 4.02 |
| churn | 1.08 | 1.73 | 1.74 | 2.58 | 2.02 | 2.98 | 1.91 | 3.0 | 0.87 | 2.19 |
| elevators | 0.5 | 1.0 | 0.74 | 1.55 | 0.89 | 1.79 | 0.85 | 1.8 | 0.82 | 1.5 |
| pollen | 0.48 | 0.73 | 0.45 | 0.94 | 0.45 | 1.34 | 0.42 | 1.35 | 0.43 | 1.41 |
| phishing | 0.26 | 0.16 | 0.26 | 0.2 | 0.26 | 0.24 | 0.32 | 0.34 | 0.39 | 0.41 |
| wilt | 0.32 | 0.39 | 0.57 | 0.72 | 1.12 | 1.55 | 1.08 | 1.94 | 0.72 | 1.2 |
| letters | 0.67 | 0.74 | 0.47 | 0.5 | 0.51 | 0.56 | 0.54 | 0.61 | 0.58 | 0.69 |

Table 8: OpenML Error Bands with initial sample size 1000: Average standard errors for error and churn across baselines for each dataset and network across 100 runs.

| Dataset | network | cold | warm | s-perturb | mixup | ls | co-dist | anchor | distill |
|---|---|---|---|---|---|---|---|---|---|
| mnist | convnet-1 | 38.0 | 36.1 | 36.3 | 36.4 | N/A | 36.6 | **34.5** | N/A |
| | convnet-2 | 25.4 | N/A | **24.0** | 24.5 | 24.8 | N/A | N/A | N/A |
| | convnet-4 | 20.7 | N/A | 19.4 | 19.0 | 18.9 | N/A | 17.5 | **17.3** |
| | convnet-8 | 23.4 | N/A | 23.0 | 22.7 | 22.1 | N/A | **21.2** | N/A |
| | convnet-16 | 27.0 | N/A | 27.1 | 25.8 | 25.8 | N/A | **25.4** | N/A |
| fashion mnist | convnet-1 | 36.9 | 34.4 | 34.7 | 32.8 | 33.9 | 33.7 | 32.8 | **28.9** |
| | convnet-2 | 34.0 | 32.9 | 33.0 | 31.5 | 31.0 | 31.2 | 31.8 | **28.0** |
| | convnet-4 | 31.8 | N/A | N/A | 31.1 | 30.8 | 30.8 | **30.6** | N/A |
| | convnet-8 | 28.9 | N/A | 29.7 | 27.9 | 27.3 | 27.5 | 27.9 | **24.1** |
| | convnet-16 | 35.0 | N/A | 32.5 | 33.0 | 32.7 | 32.4 | 33.5 | **26.2** |
| emnist balanced | convnet-1 | 93.4 | 91.7 | 92.0 | 91.6 | N/A | **91.0** | 91.7 | N/A |
| | convnet-2 | 87.0 | 84.2 | 84.4 | 84.5 | 84.9 | **84.1** | 84.4 | N/A |
| | convnet-4 | 85.9 | 82.6 | 82.0 | 81.7 | 82.0 | 82.2 | 82.0 | **76.4** |
| | convnet-8 | 84.8 | 82.0 | 82.2 | 82.1 | 82.2 | 82.0 | 81.7 | **74.3** |
| | convnet-16 | 88.6 | N/A | 87.5 | 87.4 | 87.3 | N/A | 87.4 | **82.2** |
| emnist byclass | convnet-1 | 73.5 | 71.75 | 70.5 | 69.75 | 70.25 | 69.25 | 70.0 | **65.75** |
| | convnet-2 | 68.8 | 64.6 | 65.8 | 63.6 | **63.2** | 64.2 | 63.6 | N/A |
| | convnet-4 | 64.8 | 62.6 | 62.0 | 60.0 | 59.2 | 61.2 | 61.4 | **52.6** |
| | convnet-8 | 67.5 | 63.25 | 64.25 | 64.5 | 63.25 | 64.25 | 61.25 | **51.5** |
| | convnet-16 | 63.0 | 63.33 | 59.67 | 58.67 | 57.67 | 61.33 | 57.33 | **50.67** |
| emnist bymerge | convnet-1 | 76.5 | 77.5 | **75.0** | 75.5 | 77.25 | 75.5 | **75.0** | N/A |
| | convnet-2 | 71.8 | N/A | 67.4 | 66.0 | 67.4 | 67.8 | 67.6 | **62.0** |
| | convnet-4 | 61.4 | N/A | **58.0** | 59.0 | 59.0 | 61.4 | 58.2 | N/A |
| | convnet-8 | 65.25 | 61.75 | 58.75 | 60.75 | 60.0 | 59.75 | 59.75 | **57.25** |
| | convnet-16 | 65.33 | 59.67 | 60.33 | 59.67 | 59.33 | 60.33 | 57.67 | **50.67** |
| emnist letters | convnet-1 | 77.4 | 75.9 | 76.4 | 75.3 | N/A | **74.5** | 75.7 | N/A |
| | convnet-2 | 68.8 | 67.7 | 66.4 | 66.8 | 66.6 | 66.6 | **65.6** | N/A |
| | convnet-4 | 61.9 | N/A | 59.9 | 59.8 | 59.4 | 60.5 | 58.9 | **55.2** |
| | convnet-8 | 63.4 | N/A | 62.7 | 62.0 | 61.0 | N/A | **60.5** | N/A |
| | convnet-16 | 66.5 | N/A | 66.1 | 66.2 | **65.2** | N/A | **65.2** | N/A |
| emnist digits | convnet-1 | 33.8 | 32.0 | 32.1 | 31.9 | 31.5 | 31.6 | 30.9 | **29.6** |
| | convnet-2 | 23.0 | N/A | 22.9 | N/A | **22.8** | N/A | N/A | N/A |
| | convnet-4 | 23.3 | 22.2 | 22.9 | 21.7 | 21.4 | N/A | **20.2** | N/A |
| | convnet-8 | 18.8 | N/A | 19.7 | 19.1 | 18.3 | N/A | 16.9 | **16.8** |
| | convnet-16 | 21.89 | N/A | 23.44 | 22.56 | 21.56 | N/A | 20.67 | **19.56** |
| emnist mnist | convnet-1 | 33.3 | 31.4 | 31.5 | **31.0** | N/A | 32.6 | N/A | N/A |
| | convnet-2 | 22.6 | 22.2 | 22.2 | 22.3 | 22.1 | N/A | **21.4** | N/A |
| | convnet-4 | 19.6 | N/A | N/A | **19.0** | 19.5 | N/A | N/A | N/A |
| | convnet-8 | 21.6 | N/A | 20.9 | 21.8 | 20.4 | N/A | **20.0** | N/A |
| | convnet-16 | 22.8 | N/A | N/A | 22.1 | 21.9 | N/A | **21.1** | N/A |
| kmnist | convnet-1 | 53.4 | 49.8 | 50.6 | 50.4 | 51.2 | 49.2 | **47.5** | N/A |
| | convnet-2 | 42.7 | 40.4 | 40.9 | 40.9 | 41.1 | 40.7 | 37.9 | **37.1** |
| | convnet-4 | 40.4 | N/A | 37.5 | 38.7 | 37.8 | N/A | 37.3 | **35.5** |
| | convnet-8 | 39.9 | N/A | 40.3 | 38.1 | 38.3 | N/A | 37.2 | **34.2** |
| | convnet-16 | 41.2 | N/A | N/A | 40.3 | 39.5 | N/A | **38.8** | N/A |
| k49 mnist | convnet-1 | 93.5 | 89.8 | 89.7 | 89.1 | 89.9 | **87.9** | 88.6 | N/A |
| | convnet-2 | 86.2 | 83.7 | 83.8 | 83.7 | 83.9 | 83.1 | 83.3 | **76.8** |
| | convnet-4 | 83.4 | N/A | 82.6 | 81.4 | 81.4 | 81.1 | 80.9 | **72.5** |
| | convnet-8 | 76.5 | 73.8 | 74.1 | 73.3 | 73.0 | 71.8 | 70.8 | **62.1** |
| | convnet-16 | 79.44 | 78.11 | 76.22 | 76.11 | 75.89 | 76.11 | 77.0 | **65.11** |

Table 9: Results for MNIST variants with initial sample 100 under churn at cold accuracy metric across different sizes of convolutional networks.

| Dataset | convnet-1 | | convnet-2 | | convnet-4 | | convnet-8 | | convnet-16 | |
|---|---|---|---|---|---|---|---|---|---|---|
| | Error | Churn | Error | Churn | Error | Churn | Error | Churn | Error | Churn |
| mnist | 0.48 | 0.52 | 0.38 | 0.41 | 0.37 | 0.39 | 0.41 | 0.43 | 0.43 | 0.45 |
| fashion mnist | 0.52 | 0.58 | 0.56 | 0.59 | 0.55 | 0.59 | 0.56 | 0.59 | 0.59 | 0.62 |
| emnist balanced | 0.59 | 0.61 | 0.69 | 0.7 | 0.73 | 0.73 | 0.77 | 0.76 | 0.68 | 0.69 |
| emnist byclass | 0.58 | 0.65 | 0.68 | 0.72 | 0.71 | 0.73 | 0.6 | 0.64 | 0.59 | 0.72 |
| emnist bymerge | 0.56 | 0.62 | 0.66 | 0.68 | 0.7 | 0.76 | 0.58 | 0.64 | 0.55 | 0.58 |
| emnist letters | 0.59 | 0.63 | 0.69 | 0.69 | 0.75 | 0.74 | 0.73 | 0.74 | 0.68 | 0.69 |
| emnist digits | 0.44 | 0.47 | 0.34 | 0.37 | 0.43 | 0.47 | 0.44 | 0.48 | 0.45 | 0.49 |
| emnist mnist | 0.44 | 0.46 | 0.35 | 0.36 | 0.39 | 0.41 | 0.41 | 0.46 | 0.4 | 0.43 |
| kmnist | 0.54 | 0.59 | 0.56 | 0.6 | 0.59 | 0.62 | 0.62 | 0.67 | 0.66 | 0.71 |
| k49 mnist | 0.63 | 0.62 | 0.72 | 0.68 | 0.74 | 0.75 | 0.76 | 0.75 | 0.69 | 0.68 |

Table 10: MNIST Error Bands with initial sample size 100: Average standard errors for error and churn across baselines for each dataset and network across 100 runs.

| Dataset | network | cold | warm | s-perturb | mixup | ls | co-dist | anchor | distill |
|---|---|---|---|---|---|---|---|---|---|
| mnist | convnet-1 | 11.18 | N/A | 10.33 | 9.75 | 9.67 | 10.05 | 8.85 | **8.57** |
| | convnet-2 | 6.96 | N/A | 6.86 | 6.62 | 5.82 | N/A | 5.81 | **5.67** |
| | convnet-4 | 6.68 | N/A | 6.78 | 5.93 | 5.02 | N/A | 5.21 | **4.81** |
| | convnet-8 | 7.13 | N/A | N/A | 5.71 | **4.95** | N/A | N/A | 5.22 |
| | convnet-16 | 7.91 | N/A | 8.25 | 6.39 | **5.35** | N/A | 6.95 | 6.35 |
| fashion mnist | convnet-1 | 22.17 | 18.72 | 19.4 | 18.76 | 18.07 | 18.13 | 17.51 | **11.88** |
| | convnet-2 | 19.85 | 18.17 | 17.93 | 17.44 | 17.49 | 16.86 | 16.48 | **12.16** |
| | convnet-4 | 18.48 | N/A | 17.08 | 16.93 | 16.52 | 16.53 | 15.75 | **11.9** |
| | convnet-8 | 17.57 | N/A | 16.35 | 16.07 | 14.62 | 15.36 | 15.1 | **9.05** |
| | convnet-16 | 17.98 | N/A | 17.23 | 16.88 | 15.9 | 16.21 | 15.87 | **11.11** |
| emnist balanced | convnet-1 | 52.34 | 41.04 | 41.86 | 43.79 | 41.68 | 40.97 | **38.62** | N/A |
| | convnet-2 | 45.04 | N/A | 39.29 | 39.45 | 37.78 | 38.43 | 35.3 | **32.44** |
| | convnet-4 | 42.21 | N/A | 37.46 | 37.12 | 35.53 | N/A | 33.64 | **29.41** |
| | convnet-8 | 40.95 | N/A | 35.82 | 35.66 | 33.96 | N/A | 32.5 | **26.39** |
| | convnet-16 | 42.74 | N/A | 39.11 | 37.3 | 35.13 | N/A | N/A | **29.27** |
| emnist byclass | convnet-1 | 44.18 | 36.18 | 36.35 | 36.15 | 34.9 | 34.96 | 35.62 | **27.07** |
| | convnet-2 | 38.39 | 33.21 | 34.04 | 33.68 | 32.56 | 33.2 | 31.41 | **24.17** |
| | convnet-4 | 36.4 | N/A | 32.33 | 31.74 | 30.79 | 31.96 | 30.42 | **24.5** |
| | convnet-8 | 36.72 | N/A | 32.9 | 31.7 | 30.17 | N/A | N/A | **25.51** |
| | convnet-16 | 36.81 | 33.06 | 33.11 | 31.59 | 29.64 | 31.21 | 30.68 | **21.81** |
| emnist bymerge | convnet-1 | 42.59 | N/A | 34.44 | 34.76 | 34.12 | 33.76 | **32.58** | N/A |
| | convnet-2 | 37.35 | N/A | 32.87 | 32.52 | 31.43 | N/A | **30.74** | N/A |
| | convnet-4 | 34.17 | 30.62 | 30.38 | 29.86 | 28.72 | 29.59 | 27.07 | **21.28** |
| | convnet-8 | 33.59 | N/A | 30.56 | 29.89 | **27.63** | N/A | N/A | N/A |
| | convnet-16 | 35.46 | N/A | 31.65 | 30.77 | 28.83 | N/A | 28.27 | **22.9** |
| emnist letters | convnet-1 | 38.03 | 30.14 | 30.95 | 31.53 | 30.39 | 30.26 | 27.41 | **24.84** |
| | convnet-2 | 31.65 | N/A | 28.36 | 27.72 | 26.37 | N/A | 24.26 | **21.88** |
| | convnet-4 | 29.58 | N/A | 26.99 | 26.2 | 24.61 | N/A | 23.22 | **20.16** |
| | convnet-8 | 29.52 | N/A | 28.08 | 26.19 | 24.39 | N/A | 22.73 | **20.94** |
| | convnet-16 | 30.15 | N/A | 27.36 | 26.39 | 24.28 | N/A | 23.77 | **20.67** |
| emnist digits | convnet-1 | 9.16 | N/A | 8.71 | 8.29 | 8.04 | N/A | 7.62 | **7.31** |
| | convnet-2 | 6.43 | N/A | 6.38 | 5.98 | 5.17 | N/A | 5.12 | **5.02** |
| | convnet-4 | 6.81 | N/A | 6.95 | 6.09 | 5.29 | N/A | 5.42 | **4.81** |
| | convnet-8 | 6.74 | N/A | N/A | 6.14 | 5.11 | N/A | 5.9 | **4.91** |
| | convnet-16 | 6.93 | N/A | N/A | 5.72 | **4.72** | N/A | N/A | 5.61 |
| emnist mnist | convnet-1 | 9.16 | N/A | 8.28 | 8.3 | 7.71 | 8.62 | 7.39 | **7.34** |
| | convnet-2 | 5.7 | N/A | 6.02 | 5.4 | 4.94 | N/A | 4.95 | **4.59** |
| | convnet-4 | 6.42 | N/A | 6.28 | 5.67 | 4.9 | N/A | 5.21 | **4.49** |
| | convnet-8 | 6.39 | N/A | 6.64 | 5.16 | 4.48 | N/A | 5.28 | **4.23** |
| | convnet-16 | 7.09 | N/A | 7.14 | 6.03 | 5.15 | N/A | 6.11 | **4.97** |
| kmnist | convnet-1 | 23.31 | 18.22 | 18.25 | 19.1 | 18.97 | 18.56 | 16.68 | **16.08** |
| | convnet-2 | 16.54 | N/A | 15.35 | 14.98 | 13.79 | N/A | 13.12 | **12.18** |
| | convnet-4 | 15.95 | N/A | 14.08 | 13.41 | 12.08 | N/A | 12.0 | **9.9** |
| | convnet-8 | 16.89 | N/A | 15.17 | 14.19 | 12.68 | N/A | 12.72 | **10.67** |
| | convnet-16 | 18.0 | N/A | 16.57 | 15.37 | 13.49 | N/A | 13.64 | **11.97** |
| k49 mnist | convnet-1 | 56.23 | 44.13 | 44.89 | 46.74 | 46.12 | 43.13 | 41.79 | **40.3** |
| | convnet-2 | 48.22 | 40.43 | 41.02 | 40.89 | 38.97 | 40.5 | 36.31 | **31.52** |
| | convnet-4 | 46.35 | N/A | 39.48 | 39.46 | 37.33 | 39.99 | 35.24 | **29.46** |
| | convnet-8 | 47.84 | N/A | 41.35 | 40.98 | 38.23 | N/A | 36.39 | **29.13** |
| | convnet-16 | 49.02 | 42.1 | 42.1 | 41.44 | 38.45 | 41.59 | 38.16 | **30.74** |

Table 11: Results for MNIST variants under churn at cold accuracy metric across different sizes of convolutional networks with initial sample size 1000..

| Dataset | convnet-1 | | convnet-2 | | convnet-4 | | convnet-8 | | convnet-16 | |
|---|---|---|---|---|---|---|---|---|---|---|
| | Error | Churn | Error | Churn | Error | Churn | Error | Churn | Error | Churn |
| mnist | 0.48 | 0.52 | 0.38 | 0.41 | 0.37 | 0.39 | 0.41 | 0.43 | 0.43 | 0.45 |
| fashion mnist | 0.52 | 0.58 | 0.56 | 0.59 | 0.55 | 0.59 | 0.56 | 0.59 | 0.59 | 0.62 |
| emnist balanced | 0.59 | 0.61 | 0.69 | 0.7 | 0.73 | 0.73 | 0.77 | 0.76 | 0.68 | 0.69 |
| emnist byclass | 0.58 | 0.65 | 0.68 | 0.72 | 0.71 | 0.73 | 0.6 | 0.64 | 0.62 | 0.76 |
| emnist bymerge | 0.56 | 0.62 | 0.66 | 0.68 | 0.7 | 0.76 | 0.58 | 0.64 | 0.59 | 0.61 |
| emnist letters | 0.59 | 0.63 | 0.69 | 0.69 | 0.75 | 0.74 | 0.73 | 0.74 | 0.68 | 0.69 |
| emnist digits | 0.44 | 0.47 | 0.34 | 0.37 | 0.43 | 0.47 | 0.44 | 0.48 | 0.45 | 0.49 |
| emnist mnist | 0.44 | 0.46 | 0.35 | 0.36 | 0.39 | 0.41 | 0.41 | 0.46 | 0.4 | 0.43 |
| kmnist | 0.54 | 0.59 | 0.56 | 0.6 | 0.59 | 0.62 | 0.62 | 0.67 | 0.66 | 0.71 |
| k49 mnist | 0.63 | 0.62 | 0.72 | 0.68 | 0.74 | 0.75 | 0.76 | 0.75 | 0.69 | 0.68 |

Table 12: MNIST Error Bands with initial sample size 1000: Average standard errors for error and churn across baselines for each dataset and network across 100 runs.

| Dataset | network | cold | warm | s-perturb | mixup | ls | co-dist | anchor | distill |
|---|---|---|---|---|---|---|---|---|---|
| mnist | convnet-1 | 4.84 | 4.22 | 4.01 | 3.73 | 3.34 | 3.9 | 3.81 | **3.32** |
| | convnet-2 | 4.16 | 3.78 | 3.61 | 3.22 | **2.86** | 3.46 | 3.3 | 2.96 |
| | convnet-4 | 4.06 | 3.66 | 3.49 | 3.09 | **2.72** | 3.4 | 3.2 | 2.93 |
| | convnet-8 | 4.29 | 4.01 | 3.81 | 3.35 | **2.99** | 3.79 | 3.54 | 3.19 |
| | convnet-16 | 4.47 | 4.04 | 3.97 | 3.44 | **3.08** | 3.84 | 3.65 | 3.2 |
| fashion mnist | convnet-1 | 14.91 | 12.32 | 12.5 | 12.26 | 11.98 | 11.65 | 11.46 | **7.81** |
| | convnet-2 | 13.82 | 11.96 | 11.9 | 11.77 | 11.41 | 11.29 | 11.12 | **7.5** |
| | convnet-4 | 13.04 | 11.52 | 11.42 | 11.3 | 10.93 | 10.77 | 10.64 | **6.78** |
| | convnet-8 | 12.92 | 11.44 | 11.52 | 11.18 | 10.98 | 10.8 | 10.85 | **7.06** |
| | convnet-16 | 13.56 | 12.13 | 12.0 | 11.81 | 11.38 | 11.55 | 11.29 | **7.61** |
| emnist balanced | convnet-1 | 25.34 | 20.86 | 20.83 | 20.8 | 19.89 | 19.97 | 19.48 | **14.26** |
| | convnet-2 | 23.33 | 20.18 | 20.09 | 19.45 | 18.38 | 19.11 | 18.67 | **13.62** |
| | convnet-4 | 24.71 | 19.93 | 19.76 | 19.07 | 17.5 | 19.0 | 18.07 | **13.0** |
| | convnet-8 | 22.49 | 20.17 | 20.12 | 19.21 | 17.62 | 19.35 | 18.48 | **13.22** |
| | convnet-16 | 22.48 | 20.09 | 20.3 | 19.35 | 17.48 | 19.52 | 18.35 | **13.4** |
| emnist byclass | convnet-1 | 23.66 | N/A | 19.92 | 19.46 | 18.93 | 18.69 | N/A | **13.34** |
| | convnet-2 | 21.7 | N/A | 19.25 | 18.73 | 17.64 | 18.18 | N/A | **12.32** |
| | convnet-4 | 21.45 | N/A | 19.01 | 18.71 | 17.41 | 18.09 | 18.39 | **11.76** |
| | convnet-8 | 21.47 | N/A | 19.42 | 18.72 | 17.08 | 18.13 | N/A | **11.9** |
| | convnet-16 | 21.83 | N/A | 19.63 | 18.82 | 17.28 | 18.68 | 18.78 | **12.71** |
| emnist bymerge | convnet-1 | 21.1 | 17.78 | 17.35 | 17.44 | 16.67 | 16.61 | 16.98 | **12.09** |
| | convnet-2 | 19.27 | 17.02 | 16.74 | 16.33 | 15.41 | 15.84 | 16.21 | **11.23** |
| | convnet-4 | 18.37 | 16.72 | 16.41 | 15.85 | 14.65 | 15.42 | 15.91 | **10.74** |
| | convnet-8 | 19.01 | 17.08 | 17.12 | 16.31 | 15.05 | 16.18 | 16.44 | **10.95** |
| | convnet-16 | 18.91 | N/A | 17.49 | 16.56 | 15.08 | 16.5 | 16.66 | **11.11** |
| emnist letters | convnet-1 | 17.17 | 14.29 | 14.25 | 13.82 | 13.03 | 13.56 | 13.09 | **10.47** |
| | convnet-2 | 15.44 | 13.62 | 13.51 | 12.9 | 11.74 | 12.96 | 12.39 | **9.82** |
| | convnet-4 | 15.25 | 13.58 | 13.32 | 12.67 | 11.45 | 12.99 | 12.27 | **9.33** |
| | convnet-8 | 15.16 | 13.25 | 13.33 | 12.52 | 11.32 | 12.76 | 12.17 | **9.14** |
| | convnet-16 | 15.19 | 13.63 | 13.33 | 12.55 | 11.18 | 12.99 | 12.42 | **9.4** |
| emnist digits | convnet-1 | 3.98 | 3.43 | 3.28 | 3.0 | **2.64** | 3.21 | 3.06 | 2.82 |
| | convnet-2 | 3.64 | 3.36 | 3.13 | 2.91 | **2.53** | 3.1 | 2.95 | 2.7 |
| | convnet-4 | 3.59 | 3.24 | 3.17 | 2.77 | **2.37** | 3.07 | 2.88 | 2.56 |
| | convnet-8 | 3.89 | 3.41 | 3.33 | 2.88 | **2.58** | 3.3 | 3.16 | 2.73 |
| | convnet-16 | 4.0 | 3.56 | 3.52 | 2.96 | **2.63** | 3.34 | 3.24 | 2.74 |
| emnist mnist | convnet-1 | 4.09 | 3.5 | 3.39 | 3.14 | **2.84** | 3.29 | 3.17 | 2.93 |
| | convnet-2 | 3.69 | 3.33 | 3.11 | 2.78 | **2.42** | 3.02 | 2.91 | 2.65 |
| | convnet-4 | 3.64 | 3.36 | 3.22 | 2.86 | **2.5** | 3.15 | 3.04 | 2.74 |
| | convnet-8 | 3.76 | 3.4 | 3.41 | 2.92 | **2.55** | 3.35 | 3.13 | 2.73 |
| | convnet-16 | 3.94 | 3.6 | 3.51 | 2.98 | **2.63** | 3.41 | 3.23 | 2.93 |
| kmnist | convnet-1 | 8.99 | 7.31 | 7.2 | 6.76 | 6.18 | 7.07 | 6.64 | **6.17** |
| | convnet-2 | 7.91 | 6.96 | 6.73 | 6.17 | **5.48** | 6.63 | 6.28 | 5.62 |
| | convnet-4 | 7.69 | 6.8 | 6.66 | 6.07 | **5.24** | 6.57 | 6.22 | 5.42 |
| | convnet-8 | 7.93 | 6.71 | 6.74 | 6.03 | **5.26** | 6.6 | 6.23 | 5.52 |
| | convnet-16 | 8.07 | 6.99 | 6.89 | 6.02 | **5.35** | 6.72 | 6.29 | 5.81 |
| k49 mnist | convnet-1 | 27.34 | 21.79 | 21.76 | 21.62 | 20.83 | 20.8 | 19.27 | **17.11** |
| | convnet-2 | 23.82 | 19.97 | 20.03 | 19.27 | 17.75 | 19.42 | 18.02 | **15.99** |
| | convnet-4 | 22.73 | 19.42 | 19.34 | 18.48 | 16.57 | 18.93 | 17.43 | **15.4** |
| | convnet-8 | 22.33 | 19.22 | 19.29 | 18.21 | 16.21 | 18.69 | 17.1 | **15.14** |
| | convnet-16 | 22.37 | 19.47 | 19.23 | 18.02 | 15.91 | 18.94 | 17.19 | **15.46** |

Table 13: Results for MNIST variants with initial sample 10000 under churn at cold accuracy metric across different sizes of convolutional networks.

| Dataset | convnet-1 | | convnet-2 | | convnet-4 | | convnet-8 | | convnet-16 | |
|---|---|---|---|---|---|---|---|---|---|---|
| | Error | Churn | Error | Churn | Error | Churn | Error | Churn | Error | Churn |
| mnist | 0.34 | 0.36 | 0.31 | 0.32 | 0.38 | 0.39 | 0.43 | 0.44 | 0.46 | 0.47 |
| fashion mnist | 0.27 | 0.3 | 0.27 | 0.3 | 0.36 | 0.39 | 0.37 | 0.39 | 0.37 | 0.49 |
| emnist balanced | 0.23 | 0.25 | 0.31 | 0.34 | 0.34 | 0.37 | 0.36 | 0.39 | 0.36 | 0.38 |
| emnist byclass | 0.23 | 0.3 | 0.26 | 0.32 | 0.29 | 0.35 | 0.29 | 0.33 | 0.31 | 0.35 |
| emnist bymerge | 0.25 | 0.29 | 0.3 | 0.34 | 0.32 | 0.37 | 0.29 | 0.34 | 0.31 | 0.35 |
| emnist letters | 0.31 | 0.34 | 0.42 | 0.44 | 0.45 | 0.47 | 0.46 | 0.49 | 0.47 | 0.48 |
| emnist digits | 0.26 | 0.27 | 0.29 | 0.3 | 0.36 | 0.37 | 0.49 | 0.5 | 0.44 | 0.45 |
| emnist mnist | 0.29 | 0.31 | 0.28 | 0.29 | 0.41 | 0.42 | 0.48 | 0.49 | 0.43 | 0.44 |
| kmnist | 0.39 | 0.4 | 0.49 | 0.51 | 0.53 | 0.54 | 0.6 | 0.61 | 0.62 | 0.63 |
| k49 mnist | 0.41 | 0.42 | 0.45 | 0.46 | 0.54 | 0.55 | 0.54 | 0.54 | 0.57 | 0.56 |

Table 14: MNIST Error Bands with initial sample size 10000: Average standard errors for error and churn across baselines for each dataset and network across 100 runs.

| Dataset | network | cold | warm | s-perturb | mixup | ls | co-dist | anchor | distill |
|---|---|---|---|---|---|---|---|---|---|
| svhn | convnet-1 | 79.2 | N/A | N/A | 77.2 | 78.4 | N/A | 80.8 | **70.8** |
| | convnet-2 | 81.2 | 79.9 | N/A | 80.3 | 81.2 | 82.6 | 83.4 | **74.1** |
| | convnet-4 | 80.3 | 81.5 | 74.6 | 79.1 | 80.2 | 72.4 | 84.7 | **64.6** |
| | convnet-8 | 70.22 | 80.78 | 72.78 | 70.22 | 62.56 | 69.33 | 71.33 | **59.89** |
| | convnet-16 | 41.33 | 70.83 | 41.0 | 52.5 | 52.33 | 53.67 | 42.83 | **34.5** |
| cifar10 | convnet-1 | 77.9 | N/A | 76.3 | N/A | 76.0 | **75.5** | N/A | N/A |
| | convnet-2 | 74.7 | 77.1 | 73.3 | 74.6 | 73.0 | 72.3 | 74.1 | **70.1** |
| | convnet-4 | 71.7 | 70.4 | 70.8 | 73.6 | 70.9 | 69.0 | N/A | **61.5** |
| | convnet-8 | **75.5** | N/A | N/A | N/A | N/A | N/A | N/A | N/A |
| | convnet-16 | 79.4 | 79.5 | 79.9 | 76.7 | 78.2 | 78.1 | 82.8 | **69.9** |

Table 15: Results for SVHN and CIFAR datasets with initial sample size 100 under churn at cold accuracy metric across different sizes of convolutional networks..

| Dataset | convnet-1 | | convnet-2 | | convnet-4 | | convnet-8 | | convnet-16 | |
|---|---|---|---|---|---|---|---|---|---|---|
| | Error | Churn | Error | Churn | Error | Churn | Error | Churn | Error | Churn |
| svhn | 0.67 | 1.33 | 0.73 | 1.37 | 0.88 | 1.59 | 1.35 | 2.18 | 1.87 | 3.25 |
| cifar10 | 0.4 | 0.89 | 0.39 | 0.85 | 0.41 | 0.9 | 0.4 | 1.0 | 0.43 | 1.1 |
| cifar100 | 0.39 | 0.93 | 0.4 | 0.92 | 0.4 | 0.9 | 0.41 | 1.02 | 0.45 | 1.09 |

Table 16: SVHN and CIFAR with initial sample size 100 Error Bands: Average standard errors for error and churn across baselines for each dataset and network across 100 runs.

| Dataset | network | cold | warm | s-perturb | mixup | ls | co-dist | anchor | distill |
|---|---|---|---|---|---|---|---|---|---|
| svhn | convnet-1 | 31.1 | N/A | 24.44 | 25.97 | 27.33 | 23.3 | 23.54 | **21.26** |
| | convnet-2 | 29.49 | 24.26 | 24.05 | 25.88 | 26.48 | 23.23 | 21.41 | **16.73** |
| | convnet-4 | 32.12 | 26.88 | 27.39 | 29.2 | 29.21 | 26.01 | 25.43 | **22.64** |
| | convnet-8 | 42.22 | 36.14 | 34.78 | 37.41 | 36.91 | 34.82 | 35.46 | **28.55** |
| | convnet-16 | 50.94 | 46.12 | 37.26 | 37.87 | 42.62 | 44.44 | 41.01 | **29.65** |
| cifar10 | convnet-1 | 50.82 | N/A | 43.71 | 44.4 | 45.66 | 41.53 | 44.37 | **29.45** |
| | convnet-2 | 52.16 | N/A | 51.11 | 47.64 | 48.83 | 44.72 | N/A | **39.89** |
| | convnet-4 | 52.01 | 47.57 | 46.36 | 47.17 | 47.92 | 44.61 | 45.75 | **29.13** |
| | convnet-8 | 52.65 | N/A | 47.42 | 47.39 | 48.29 | 44.34 | 47.07 | **34.17** |
| | convnet-16 | 53.24 | 46.97 | 48.43 | 48.2 | 48.79 | 44.83 | 47.59 | **34.54** |

Table 17: Results for SVHN and CIFAR under churn at cold accuracy metric across network sizes.

| Dataset | convnet-1 | | convnet-2 | | convnet-4 | | convnet-8 | | convnet-16 | |
|---|---|---|---|---|---|---|---|---|---|---|
| | Error | Churn | Error | Churn | Error | Churn | Error | Churn | Error | Churn |
| svhn | 0.53 | 0.77 | 0.61 | 0.97 | 0.6 | 1.45 | 0.73 | 2.19 | 1.4 | 2.67 |
| cifar10 | 0.41 | 0.65 | 0.41 | 0.62 | 0.39 | 0.62 | 0.39 | 0.64 | 0.41 | 0.66 |

Table 18: SVHN and CIFAR with initial sample size 1000 Error Bands: Average standard errors for error and churn across baselines for each dataset and network across 100 runs.

| Dataset | network | cold | warm | s-perturb | mixup | ls | co-dist | anchor | distill |
|---|---|---|---|---|---|---|---|---|---|
| svhn | convnet-1 | 15.43 | N/A | 11.29 | 12.08 | 12.01 | 10.63 | 9.94 | **6.37** |
| | convnet-2 | 14.29 | N/A | 12.43 | 12.15 | 11.43 | 10.97 | N/A | **6.99** |
| | convnet-4 | 14.25 | N/A | 13.27 | 12.16 | 11.38 | 11.16 | 9.34 | **7.43** |
| | convnet-8 | 14.61 | N/A | 11.9 | 11.92 | 11.04 | 11.08 | 9.09 | **7.09** |
| | convnet-16 | 24.35 | 16.97 | 16.81 | 16.0 | 15.89 | 17.04 | 15.39 | **9.68** |
| cifar10 | convnet-1 | 41.28 | N/A | 28.82 | 29.9 | 29.61 | 27.47 | 28.5 | **14.08** |
| | convnet-2 | 39.5 | N/A | 31.67 | 31.23 | 31.96 | 28.75 | N/A | **16.03** |
| | convnet-4 | 39.47 | N/A | 33.4 | 32.9 | 31.18 | N/A | N/A | **18.7** |
| | convnet-8 | 39.67 | N/A | 32.78 | 31.53 | 30.87 | 30.56 | 25.42 | **16.87** |
| | convnet-16 | 40.75 | N/A | 33.94 | N/A | 31.6 | N/A | N/A | **20.12** |

Table 19: Results for SVHN and CIFAR datasets with initial sample size 10000 under churn at cold accuracy metric across different sizes of convolutional networks..

| Dataset | convnet-1 | | convnet-2 | | convnet-4 | | convnet-8 | | convnet-16 | |
|---|---|---|---|---|---|---|---|---|---|---|
| | Error | Churn | Error | Churn | Error | Churn | Error | Churn | Error | Churn |
| svhn | 0.28 | 0.29 | 0.33 | 0.35 | 0.43 | 0.45 | 0.49 | 0.51 | 0.86 | 1.8 |
| cifar10 | 0.16 | 0.26 | 0.16 | 0.26 | 0.17 | 0.26 | 0.18 | 0.24 | 0.39 | 0.67 |

Table 20: SVHN and CIFAR with initial sample size 10000 Error Bands: Average standard errors for error and churn across baselines for each dataset and network across 100 runs.

| Dataset | network | cold | warm | s-perturb | mixup | ls | co-dist | anchor | distill |
|---|---|---|---|---|---|---|---|---|---|
| 5 o Clock Shadow | convnet-1 | 5.08 | N/A | 3.41 | 4.45 | 1.11 | 1.9 | 5.48 | **0.04** |
| | convnet-2 | 5.57 | N/A | 3.29 | 4.28 | 2.23 | 2.86 | N/A | **0.16** |
| | convnet-4 | 4.97 | N/A | 3.69 | 4.05 | 2.11 | 2.44 | N/A | **0.08** |
| | convnet-8 | 4.84 | N/A | 2.95 | 2.86 | 1.03 | 2.08 | N/A | **0.01** |
| | convnet-16 | 4.49 | 3.81 | 2.32 | 2.02 | 0.37 | 1.74 | 4.57 | **0.06** |
| Arched Eyebrows | convnet-1 | 19.39 | N/A | 18.04 | 14.6 | N/A | 13.77 | N/A | **1.97** |
| | convnet-2 | 20.83 | 20.17 | 18.86 | 17.21 | 15.36 | 14.97 | 21.2 | **1.32** |
| | convnet-4 | 17.66 | N/A | 15.71 | 13.74 | 15.2 | 13.17 | N/A | **0.97** |
| | convnet-8 | 16.99 | N/A | 14.54 | 11.22 | 11.68 | 12.7 | N/A | **0.79** |
| | convnet-16 | 16.96 | 15.49 | 12.35 | 8.59 | 6.26 | 9.77 | N/A | **0.34** |
| Attractive | convnet-1 | 36.39 | 35.27 | 35.45 | 34.43 | 37.04 | 32.97 | 36.42 | **30.17** |
| | convnet-2 | 37.0 | 36.5 | 34.84 | 36.24 | 36.7 | 34.59 | 38.84 | **27.0** |
| | convnet-4 | 37.9 | 37.46 | 36.38 | 36.73 | 37.22 | 35.93 | 40.48 | **29.28** |
| | convnet-8 | 40.86 | N/A | 40.14 | 39.71 | 39.36 | 38.49 | 41.6 | **32.93** |
| | convnet-16 | 44.17 | 41.65 | 40.81 | 42.31 | 42.15 | 41.38 | 43.67 | **37.37** |
| Bags Under Eyes | convnet-1 | 11.22 | N/A | 7.77 | 9.09 | 5.04 | 4.2 | N/A | **0.13** |
| | convnet-2 | 11.46 | 11.6 | 9.13 | 9.67 | 6.2 | 6.13 | 11.69 | **0.25** |
| | convnet-4 | 12.14 | 10.92 | 8.06 | 7.75 | 4.39 | 4.69 | 10.57 | **0.16** |
| | convnet-8 | 10.01 | N/A | 6.74 | 7.91 | 3.21 | 4.16 | N/A | **0.2** |
| | convnet-16 | 7.36 | 8.4 | 5.18 | 5.8 | 2.31 | 2.63 | N/A | **0.01** |
| Bald | convnet-1 | 0.62 | N/A | 0.41 | 0.43 | **0.0** | 0.26 | **0.0** | **0.0** |
| | convnet-2 | 0.58 | N/A | 0.5 | N/A | 0.06 | 0.31 | 0.6 | **0.04** |
| | convnet-4 | 0.71 | N/A | 0.45 | N/A | 0.04 | 0.39 | 0.75 | **0.02** |
| | convnet-8 | 0.74 | N/A | 0.25 | 0.5 | 0.03 | 0.33 | 0.53 | **0.0** |
| | convnet-16 | 0.78 | N/A | 0.3 | 0.35 | **0.0** | 0.24 | 0.53 | **0.0** |
| Bangs | convnet-1 | 11.62 | N/A | N/A | 10.06 | 10.09 | **9.88** | N/A | N/A |
| | convnet-2 | 12.77 | N/A | 12.02 | 9.5 | 10.32 | 10.65 | 12.16 | **6.97** |
| | convnet-4 | 12.11 | 11.35 | 11.4 | 9.55 | 9.64 | 10.7 | 11.63 | **7.34** |
| | convnet-8 | 12.67 | N/A | 11.79 | 10.02 | 9.66 | 11.08 | 12.41 | **5.84** |
| | convnet-16 | 13.61 | N/A | 12.92 | 10.69 | 11.7 | 11.51 | 12.76 | **8.21** |
| Big Lips | convnet-1 | 3.89 | 4.1 | 0.78 | 1.36 | N/A | 1.11 | N/A | **0.06** |
| | convnet-2 | 5.01 | N/A | 1.99 | 2.0 | N/A | 2.65 | N/A | **0.31** |
| | convnet-4 | 3.79 | N/A | 2.19 | 2.28 | N/A | 2.33 | N/A | **0.21** |
| | convnet-8 | 2.1 | N/A | 1.36 | 0.75 | N/A | 1.45 | N/A | **0.05** |
| | convnet-16 | 2.56 | N/A | 0.89 | 0.37 | 0.94 | 0.79 | N/A | **0.01** |
| Big Nose | convnet-1 | 14.24 | N/A | 11.52 | 12.31 | 10.3 | 6.85 | N/A | **0.36** |
| | convnet-2 | 15.33 | 14.56 | 11.85 | 13.72 | 11.49 | 10.19 | 14.97 | **0.73** |
| | convnet-4 | 11.32 | N/A | N/A | N/A | N/A | 10.14 | N/A | **2.4** |
| | convnet-8 | 13.76 | 12.49 | 10.95 | 11.26 | 8.83 | 7.81 | 14.23 | **0.29** |
| | convnet-16 | 11.7 | 12.16 | 9.26 | 9.43 | 7.12 | 6.84 | 12.15 | **0.4** |
| Black Hair | convnet-1 | 19.43 | 20.03 | 20.82 | N/A | 19.7 | **18.78** | N/A | N/A |
| | convnet-2 | 21.93 | 20.7 | 20.07 | 20.92 | 20.48 | 18.96 | 21.73 | **16.49** |
| | convnet-4 | 22.01 | 20.72 | 20.07 | 20.71 | 20.8 | 20.15 | 22.05 | **15.07** |
| | convnet-8 | 23.02 | 21.72 | 20.8 | 21.92 | 20.93 | 19.46 | 20.49 | **15.59** |
| | convnet-16 | 21.57 | 21.94 | 20.44 | 21.33 | 19.86 | 18.84 | 22.31 | **15.74** |
| Blond Hair | convnet-1 | 11.79 | 11.13 | 11.38 | 9.73 | 11.27 | 10.67 | 12.26 | **7.31** |
| | convnet-2 | 12.76 | N/A | 12.23 | 10.69 | 11.86 | 11.43 | 13.67 | **6.62** |
| | convnet-4 | 11.69 | N/A | 11.4 | 10.44 | 11.51 | 10.81 | 12.37 | **6.97** |
| | convnet-8 | 12.18 | N/A | N/A | 10.13 | 11.38 | 11.36 | N/A | **8.66** |
| | convnet-16 | 13.51 | 11.94 | 11.85 | 9.83 | 12.26 | 11.8 | 12.64 | **7.26** |

Table 21: Results for CelebA tasks under churn at cold accuracy metric across different sizes of convolutional networks with initial sample 100. Part 1 of 4.

| Dataset | network | cold | warm | s-perturb | mixup | ls | co-dist | anchor | distill |
|---|---|---|---|---|---|---|---|---|---|
| Blurry | convnet-1 | 0.03 | N/A | 0.03 | 0.01 | N/A | N/A | 0.01 | **0.0** |
| | convnet-2 | **0.0** | N/A | **0.0** | **0.0** | **0.0** | **0.0** | N/A | **0.0** |
| | convnet-4 | 0.03 | N/A | **0.0** | **0.0** | N/A | **0.0** | N/A | **0.0** |
| | convnet-8 | 0.14 | 0.07 | **0.0** | **0.0** | 0.01 | **0.0** | 0.06 | **0.0** |
| | convnet-16 | 0.02 | N/A | 0.02 | 0.02 | 0.02 | 0.02 | N/A | **0.01** |
| Brown Hair | convnet-1 | 14.11 | N/A | 12.46 | 6.42 | 9.48 | 8.39 | N/A | **0.46** |
| | convnet-2 | 14.79 | 12.54 | 11.45 | 5.93 | 9.5 | 8.38 | 15.33 | **0.27** |
| | convnet-4 | 13.88 | 14.21 | 12.12 | 6.01 | 9.22 | 8.71 | 16.76 | **0.47** |
| | convnet-8 | 14.21 | N/A | 12.64 | 5.19 | 7.52 | 8.54 | N/A | **0.4** |
| | convnet-16 | 13.14 | 13.45 | 9.99 | 4.18 | 3.1 | 7.23 | N/A | **0.07** |
| Bushy Eyebrows | convnet-1 | 5.5 | 4.55 | 2.81 | 3.27 | 2.66 | 2.33 | 6.1 | **0.06** |
| | convnet-2 | 5.05 | N/A | 4.44 | 4.89 | 3.4 | 3.27 | N/A | **0.16** |
| | convnet-4 | 5.15 | N/A | 3.95 | 3.7 | 3.5 | 2.94 | N/A | **0.17** |
| | convnet-8 | 6.17 | N/A | 3.42 | 3.14 | 1.83 | 2.45 | N/A | **0.01** |
| | convnet-16 | 4.2 | 4.03 | 2.54 | 1.59 | 1.62 | 1.99 | N/A | **0.08** |
| Chubby | convnet-1 | 1.01 | N/A | 0.66 | 1.03 | 0.15 | 0.39 | N/A | **0.0** |
| | convnet-2 | 1.49 | 1.2 | 0.58 | 0.89 | 0.11 | 0.46 | 1.5 | **0.0** |
| | convnet-4 | 1.31 | 1.52 | 0.89 | 1.16 | 0.21 | 0.63 | 1.6 | **0.02** |
| | convnet-8 | 1.35 | 1.6 | 0.93 | 0.97 | 0.3 | 0.59 | N/A | **0.03** |
| | convnet-16 | 0.94 | N/A | 0.38 | 0.58 | 0.06 | 0.32 | N/A | **0.0** |
| Double Chin | convnet-1 | 0.81 | N/A | 0.41 | 0.7 | 0.1 | 0.27 | N/A | **0.0** |
| | convnet-2 | 0.91 | N/A | 0.48 | 0.98 | 0.08 | 0.21 | N/A | **0.0** |
| | convnet-4 | 1.14 | N/A | 0.66 | 0.9 | 0.21 | 0.53 | 1.12 | **0.08** |
| | convnet-8 | 0.82 | N/A | 0.37 | 0.48 | 0.04 | 0.38 | 0.85 | **0.01** |
| | convnet-16 | 0.7 | N/A | 0.17 | 0.43 | 0.07 | 0.18 | N/A | **0.0** |
| Eyeglasses | convnet-1 | 4.21 | 4.1 | 4.07 | 3.82 | 2.6 | 3.48 | 4.41 | **2.1** |
| | convnet-2 | 4.33 | N/A | 3.95 | 3.82 | 3.5 | 3.57 | 4.49 | **2.43** |
| | convnet-4 | 4.22 | N/A | 4.03 | 3.76 | 2.93 | 3.34 | 4.34 | **2.26** |
| | convnet-8 | 4.48 | 3.97 | 3.78 | 3.96 | 2.91 | 3.38 | 4.09 | **2.38** |
| | convnet-16 | 4.76 | N/A | 3.64 | 3.91 | 2.34 | 2.86 | 4.1 | **2.25** |
| Goatee | convnet-1 | 1.83 | N/A | 1.24 | 1.78 | 0.18 | 0.38 | 2.18 | **0.02** |
| | convnet-2 | 1.67 | N/A | 1.15 | 1.35 | 0.24 | 0.72 | N/A | **0.01** |
| | convnet-4 | 2.23 | 1.6 | 0.9 | 1.43 | 0.38 | 0.46 | 1.61 | **0.07** |
| | convnet-8 | 1.48 | N/A | 0.82 | 1.08 | 0.14 | 0.33 | 1.74 | **0.01** |
| | convnet-16 | 1.01 | 1.03 | 0.59 | 0.48 | 0.09 | 0.29 | 1.04 | **0.0** |
| Gray Hair | convnet-1 | 2.42 | 2.04 | 1.66 | 1.82 | 0.17 | 1.5 | 2.08 | **0.02** |
| | convnet-2 | 2.41 | 2.24 | 2.07 | 1.92 | 0.25 | 1.75 | 2.14 | **0.03** |
| | convnet-4 | 2.82 | N/A | 2.32 | 2.47 | 0.55 | 1.77 | 2.33 | **0.05** |
| | convnet-8 | 2.54 | N/A | 1.83 | 2.04 | 0.33 | 1.3 | 2.27 | **0.15** |
| | convnet-16 | 2.58 | N/A | 1.99 | 1.97 | 0.16 | 1.59 | 2.37 | **0.04** |
| Heavy Makeup | convnet-1 | 33.24 | N/A | 32.88 | 32.09 | 33.07 | 31.89 | 34.89 | **28.94** |
| | convnet-2 | 33.33 | N/A | N/A | 31.73 | 33.04 | N/A | N/A | **30.77** |
| | convnet-4 | 34.25 | N/A | 32.64 | 31.62 | 32.73 | 31.38 | N/A | **27.21** |
| | convnet-8 | 34.66 | 36.17 | 33.25 | 33.08 | 33.03 | 33.12 | 36.2 | **26.42** |
| | convnet-16 | 37.36 | 35.43 | 34.73 | 32.62 | 34.98 | 35.18 | 35.91 | **29.18** |
| High Cheekbones | convnet-1 | 42.29 | N/A | 43.53 | **41.0** | N/A | 41.86 | N/A | N/A |
| | convnet-2 | 43.96 | 43.86 | 41.62 | 41.11 | 42.38 | 40.82 | 43.29 | **37.8** |
| | convnet-4 | 42.79 | N/A | N/A | 43.1 | 44.1 | 43.45 | 47.4 | **41.11** |
| | convnet-8 | 42.15 | N/A | 40.87 | 40.02 | 41.99 | 41.39 | N/A | **37.68** |
| | convnet-16 | 44.06 | 42.6 | 41.34 | 39.23 | 42.28 | 42.03 | 43.09 | **35.13** |

Table 22: Results for CelebA tasks under churn at cold accuracy metric across different sizes of convolutional networks with initial sample 100. Part 2 of 4.

| Dataset | network | cold | warm | s-perturb | mixup | ls | co-dist | anchor | distill |
|---|---|---|---|---|---|---|---|---|---|
| Male | convnet-1 | 32.43 | N/A | 32.08 | 32.34 | 31.49 | N/A | 33.36 | **27.92** |
| | convnet-2 | 32.7 | 32.02 | 32.23 | 32.63 | 32.03 | 31.82 | 33.32 | **28.17** |
| | convnet-4 | 31.72 | N/A | N/A | 31.34 | 30.87 | 30.75 | 32.75 | **25.78** |
| | convnet-8 | 34.85 | 32.86 | 33.4 | 34.14 | 33.96 | 32.84 | 34.52 | **26.24** |
| | convnet-16 | 37.02 | N/A | 35.57 | 36.9 | 35.91 | 35.21 | 36.84 | **29.26** |
| Mouth Slightly Open | convnet-1 | 44.93 | 46.3 | 44.95 | 45.8 | N/A | **44.73** | 47.42 | 46.37 |
| | convnet-2 | 46.1 | 45.38 | 46.17 | 46.32 | 46.9 | 45.13 | N/A | **42.84** |
| | convnet-4 | 45.31 | N/A | 45.69 | 44.16 | 45.62 | 44.84 | 47.76 | **42.58** |
| | convnet-8 | **45.74** | 48.62 | N/A | N/A | N/A | 46.68 | N/A | 47.03 |
| | convnet-16 | **51.37** | N/A | N/A | N/A | N/A | N/A | N/A | N/A |
| Mustache | convnet-1 | 0.3 | 0.41 | 0.04 | 0.22 | 0.02 | 0.06 | N/A | **0.01** |
| | convnet-2 | 0.27 | N/A | 0.07 | 0.07 | 0.01 | 0.03 | 0.3 | **0.0** |
| | convnet-4 | 0.6 | 0.45 | 0.18 | 0.28 | 0.09 | 0.17 | 0.69 | **0.03** |
| | convnet-8 | 0.39 | 0.5 | 0.13 | 0.19 | 0.09 | 0.17 | 0.56 | **0.05** |
| | convnet-16 | 0.21 | 0.2 | 0.07 | 0.08 | 0.04 | 0.05 | 0.19 | **0.02** |
| Narrow Eyes | convnet-1 | 0.08 | 0.17 | **0.02** | 0.03 | N/A | 0.07 | N/A | **0.02** |
| | convnet-2 | 0.11 | N/A | 0.01 | 0.01 | N/A | 0.03 | 0.07 | **0.0** |
| | convnet-4 | 0.57 | 0.14 | 0.06 | 0.07 | 0.39 | 0.15 | 0.26 | **0.05** |
| | convnet-8 | 0.09 | N/A | 0.03 | 0.03 | N/A | N/A | N/A | **0.0** |
| | convnet-16 | 0.04 | N/A | **0.01** | **0.01** | 0.02 | 0.05 | N/A | **0.01** |
| No Beard | convnet-1 | 11.04 | N/A | 10.86 | 11.74 | 12.13 | 7.67 | 13.68 | **0.95** |
| | convnet-2 | 12.31 | 11.91 | 12.61 | 12.16 | 9.93 | 7.81 | 12.0 | **2.54** |
| | convnet-4 | 13.4 | 13.82 | 10.83 | 11.8 | 6.17 | 8.72 | 13.28 | **0.49** |
| | convnet-8 | 12.85 | N/A | 10.39 | 11.29 | 7.11 | 7.29 | 10.95 | **0.05** |
| | convnet-16 | 11.34 | 10.31 | 8.75 | 9.88 | 4.16 | 5.53 | 10.26 | **0.15** |
| Oval Face | convnet-1 | 9.89 | N/A | 7.95 | 8.49 | N/A | 5.3 | N/A | **0.58** |
| | convnet-2 | 13.65 | 13.6 | 7.52 | 9.02 | 10.89 | 7.86 | 14.03 | **0.63** |
| | convnet-4 | 11.1 | N/A | 7.48 | 6.62 | N/A | 6.42 | N/A | **0.63** |
| | convnet-8 | 7.54 | N/A | 4.49 | 3.66 | 6.6 | 4.44 | N/A | **0.18** |
| | convnet-16 | 8.97 | 7.4 | 3.53 | 2.63 | 3.73 | 4.29 | 7.81 | **0.2** |
| Pale Skin | convnet-1 | 1.51 | N/A | 1.15 | 0.74 | 0.03 | 0.74 | 1.2 | **0.0** |
| | convnet-2 | 1.42 | N/A | N/A | N/A | 0.23 | 0.92 | 1.76 | **0.01** |
| | convnet-4 | 1.25 | 1.32 | 0.96 | 0.76 | 0.12 | 0.52 | N/A | **0.03** |
| | convnet-8 | 1.21 | N/A | 0.73 | 0.84 | 0.11 | 0.55 | N/A | **0.05** |
| | convnet-16 | 1.9 | 1.19 | 0.69 | 0.45 | 0.11 | 0.41 | 1.05 | **0.01** |
| Pointy Nose | convnet-1 | 13.35 | 13.06 | 8.18 | 8.8 | 11.93 | 7.32 | N/A | **0.82** |
| | convnet-2 | 14.22 | N/A | 8.32 | 8.34 | N/A | 7.52 | N/A | **0.78** |
| | convnet-4 | 10.51 | N/A | 6.62 | 6.75 | N/A | 7.34 | N/A | **0.79** |
| | convnet-8 | 10.21 | 10.3 | 4.12 | 2.65 | 6.38 | 5.53 | N/A | **0.28** |
| | convnet-16 | 8.4 | 6.03 | 3.13 | 2.26 | 3.13 | 4.33 | N/A | **0.35** |
| Receding Hairline | convnet-1 | 2.44 | N/A | 2.24 | 2.13 | 0.31 | 1.09 | N/A | **0.0** |
| | convnet-2 | 3.15 | 3.49 | 2.33 | 2.14 | 0.66 | 1.69 | 2.74 | **0.03** |
| | convnet-4 | 3.2 | N/A | 2.3 | 2.43 | 0.87 | 1.54 | N/A | **0.03** |
| | convnet-8 | 2.83 | N/A | 2.22 | 1.64 | 0.34 | 1.64 | N/A | **0.01** |
| | convnet-16 | 3.03 | 2.64 | 2.0 | 1.45 | 0.37 | 1.22 | 3.2 | **0.04** |
| Rosy Cheeks | convnet-1 | 1.66 | N/A | 0.63 | 0.94 | 0.16 | 0.41 | N/A | **0.02** |
| | convnet-2 | 1.8 | 1.67 | 0.48 | 0.58 | 0.22 | 0.41 | 1.33 | **0.01** |
| | convnet-4 | 1.28 | 1.03 | 0.43 | 0.54 | 0.31 | 0.28 | N/A | **0.02** |
| | convnet-8 | 0.98 | N/A | 0.35 | 0.27 | 0.1 | 0.31 | N/A | **0.0** |
| | convnet-16 | 0.65 | 0.59 | 0.18 | 0.27 | 0.06 | 0.19 | N/A | **0.0** |

Table 23: Results for CelebA tasks under churn at cold accuracy metric across different sizes of convolutional networks with initial sample 100. Part 3 of 4.

| Dataset | network | cold | warm | s-perturb | mixup | ls | co-dist | anchor | distill |
|---|---|---|---|---|---|---|---|---|---|
| Sideburns | convnet-1 | 1.48 | N/A | 0.84 | 1.18 | 0.07 | 0.2 | 1.28 | **0.01** |
| | convnet-2 | 1.87 | N/A | 0.99 | 1.7 | 0.2 | 0.53 | 1.88 | **0.0** |
| | convnet-4 | 1.72 | N/A | 1.04 | 1.29 | 0.15 | 0.5 | 1.63 | **0.04** |
| | convnet-8 | 1.44 | N/A | 0.79 | 1.25 | 0.26 | 0.39 | N/A | **0.05** |
| | convnet-16 | 0.77 | N/A | 0.5 | 0.66 | **0.0** | 0.22 | N/A | **0.0** |
| Smiling | convnet-1 | 42.83 | 42.46 | 42.17 | 41.71 | N/A | 40.72 | N/A | **40.15** |
| | convnet-2 | 42.82 | N/A | 41.86 | 42.56 | 42.63 | **41.52** | N/A | N/A |
| | convnet-4 | 44.26 | N/A | N/A | 43.51 | N/A | **43.41** | N/A | N/A |
| | convnet-8 | 45.69 | 45.97 | 45.27 | 44.87 | 45.01 | 44.48 | 46.89 | **41.71** |
| | convnet-16 | 47.36 | 49.32 | 46.42 | 47.13 | 47.14 | 46.85 | 50.39 | **43.22** |
| Straight Hair | convnet-1 | 2.94 | 2.91 | 1.43 | 1.61 | 2.63 | 1.15 | N/A | **0.12** |
| | convnet-2 | 3.86 | N/A | 1.85 | 2.18 | 4.14 | 1.84 | N/A | **0.19** |
| | convnet-4 | 3.77 | N/A | 1.35 | 1.72 | N/A | 1.87 | N/A | **0.21** |
| | convnet-8 | 2.19 | N/A | 1.51 | 1.23 | N/A | 1.48 | N/A | **0.07** |
| | convnet-16 | 2.33 | N/A | 0.84 | 0.62 | 0.8 | 1.17 | N/A | **0.03** |
| Wavy Hair | convnet-1 | 27.48 | 24.54 | 23.44 | 20.77 | 24.68 | 21.24 | 25.77 | **17.54** |
| | convnet-2 | 24.67 | N/A | 25.29 | 23.02 | N/A | 22.82 | N/A | **22.78** |
| | convnet-4 | 26.69 | N/A | 25.02 | **21.85** | N/A | 22.59 | N/A | N/A |
| | convnet-8 | 24.65 | N/A | 24.16 | **21.01** | N/A | 23.29 | N/A | 23.5 |
| | convnet-16 | 25.68 | 26.74 | 23.53 | 17.7 | 22.7 | 21.47 | 25.47 | **16.2** |
| Wearing Earrings | convnet-1 | 6.61 | N/A | 3.82 | 4.14 | 3.84 | 2.06 | N/A | **0.01** |
| | convnet-2 | 7.24 | N/A | 4.44 | 5.05 | 4.0 | 2.86 | N/A | **0.21** |
| | convnet-4 | 5.52 | N/A | 3.83 | 3.36 | 4.28 | 1.92 | N/A | **0.08** |
| | convnet-8 | 3.85 | N/A | 2.91 | 2.3 | 2.52 | 2.56 | N/A | **0.07** |
| | convnet-16 | 4.41 | 4.65 | 1.68 | 1.61 | 1.61 | 2.12 | N/A | **0.13** |
| Wearing Hat | convnet-1 | 3.52 | 3.61 | 3.43 | 3.38 | 2.33 | 2.92 | 3.35 | **1.94** |
| | convnet-2 | 3.82 | 3.89 | 3.58 | 3.58 | 2.88 | 3.23 | 3.54 | **2.36** |
| | convnet-4 | 3.33 | 3.64 | 3.63 | 3.86 | 2.46 | 3.35 | 3.39 | **1.79** |
| | convnet-8 | 3.95 | N/A | N/A | N/A | 3.01 | N/A | N/A | **2.53** |
| | convnet-16 | 3.93 | 3.59 | 3.1 | 3.67 | **1.75** | 2.83 | 3.43 | 1.97 |
| Wearing Lipstick | convnet-1 | 33.01 | 33.37 | 32.15 | 32.32 | 32.06 | 31.24 | 32.8 | **27.09** |
| | convnet-2 | 32.99 | N/A | 32.23 | 32.45 | 32.59 | 31.67 | 33.89 | **28.27** |
| | convnet-4 | 34.36 | N/A | N/A | 33.77 | 33.88 | N/A | N/A | **29.93** |
| | convnet-8 | 36.84 | N/A | N/A | 36.27 | 35.6 | 35.16 | 38.95 | **29.11** |
| | convnet-16 | 38.4 | N/A | 37.44 | 37.24 | 37.47 | 36.44 | 37.14 | **30.81** |
| Wearing Necklace | convnet-1 | 1.03 | N/A | 0.21 | 0.3 | 0.3 | 0.16 | N/A | **0.0** |
| | convnet-2 | 1.39 | N/A | 0.37 | 0.54 | 1.25 | 0.45 | N/A | **0.02** |
| | convnet-4 | 0.94 | N/A | 0.28 | 0.15 | 0.75 | 0.47 | N/A | **0.0** |
| | convnet-8 | 1.2 | N/A | 0.59 | 0.21 | 0.31 | 0.49 | N/A | **0.05** |
| | convnet-16 | 0.9 | 0.69 | 0.49 | 0.15 | 0.19 | 0.27 | N/A | **0.04** |
| Wearing Necktie | convnet-1 | 5.1 | 5.48 | 5.12 | 4.88 | 3.14 | 4.54 | 5.62 | **2.08** |
| | convnet-2 | 5.11 | N/A | 4.99 | 5.07 | 3.23 | 4.54 | 5.09 | **1.96** |
| | convnet-4 | 5.64 | N/A | N/A | N/A | 3.94 | 4.96 | 5.61 | **2.53** |
| | convnet-8 | 5.51 | N/A | N/A | 5.15 | 3.85 | 4.78 | N/A | **1.51** |
| | convnet-16 | 5.25 | N/A | 5.11 | 4.98 | 2.39 | 4.4 | 5.08 | **1.59** |
| Young | convnet-1 | 15.03 | 14.34 | 12.91 | 13.25 | 12.65 | 10.12 | 16.0 | **2.63** |
| | convnet-2 | 15.15 | 15.94 | 13.72 | 12.88 | 10.28 | 10.82 | 15.26 | **1.05** |
| | convnet-4 | 15.53 | N/A | 13.24 | 13.1 | 14.28 | 11.15 | 14.29 | **0.72** |
| | convnet-8 | 14.18 | 15.34 | 11.96 | 11.72 | 9.47 | 10.25 | 14.33 | **0.51** |
| | convnet-16 | 13.18 | N/A | 12.24 | 11.56 | 10.2 | 10.22 | N/A | **0.36** |

Table 24: Results for CelebA tasks under churn at cold accuracy metric across different sizes of convolutional networks with initial sample 100. Part 4 of 4.

| Dataset | convnet-1 | | convnet-2 | | convnet-4 | | convnet-8 | | convnet-16 | |
|---|---|---|---|---|---|---|---|---|---|---|
| | Error | Churn | Error | Churn | Error | Churn | Error | Churn | Error | Churn |
| 5 o Clock Shadow | 0.8 | 1.1 | 0.77 | 1.07 | 0.82 | 1.14 | 0.81 | 1.13 | 0.83 | 1.13 |
| Arched Eyebrows | 0.83 | 1.73 | 0.82 | 1.73 | 0.82 | 1.78 | 0.82 | 1.89 | 0.84 | 1.88 |
| Attractive | 0.4 | 1.3 | 0.41 | 1.36 | 0.39 | 1.57 | 0.42 | 1.86 | 0.42 | 1.97 |
| Bags Under Eyes | 0.85 | 1.54 | 0.88 | 1.55 | 0.87 | 1.55 | 0.86 | 1.54 | 0.87 | 1.55 |
| Bald | 0.41 | 0.43 | 0.44 | 0.49 | 0.47 | 0.51 | 0.53 | 0.57 | 0.48 | 0.51 |
| Bangs | 0.79 | 1.13 | 0.83 | 1.15 | 0.82 | 1.14 | 0.84 | 1.18 | 0.86 | 1.26 |
| Big Lips | 0.74 | 1.46 | 0.77 | 1.54 | 0.79 | 1.6 | 0.79 | 1.55 | 0.74 | 1.46 |
| Big Nose | 0.82 | 1.6 | 0.83 | 1.63 | 0.82 | 1.65 | 0.82 | 1.68 | 0.84 | 1.71 |
| Black Hair | 0.82 | 1.42 | 0.8 | 1.39 | 0.82 | 1.48 | 0.83 | 1.56 | 0.81 | 1.52 |
| Blond Hair | 0.83 | 1.17 | 0.85 | 1.15 | 0.84 | 1.18 | 0.84 | 1.18 | 0.85 | 1.27 |
| Blurry | 0.5 | 0.57 | 0.53 | 0.58 | 0.5 | 0.56 | 0.55 | 0.6 | 0.59 | 0.66 |
| Brown Hair | 0.84 | 1.51 | 0.84 | 1.53 | 0.85 | 1.55 | 0.87 | 1.63 | 0.86 | 1.62 |
| Bushy Eyebrows | 0.84 | 1.25 | 0.83 | 1.23 | 0.82 | 1.25 | 0.84 | 1.27 | 0.79 | 1.21 |
| Chubby | 0.63 | 0.73 | 0.64 | 0.75 | 0.63 | 0.74 | 0.69 | 0.81 | 0.63 | 0.73 |
| Double Chin | 0.58 | 0.66 | 0.61 | 0.7 | 0.62 | 0.74 | 0.58 | 0.67 | 0.59 | 0.67 |
| Eyeglasses | 0.7 | 0.81 | 0.68 | 0.81 | 0.65 | 0.78 | 0.71 | 0.84 | 0.72 | 0.86 |
| Goatee | 0.69 | 0.83 | 0.66 | 0.8 | 0.68 | 0.81 | 0.73 | 0.85 | 0.65 | 0.77 |
| Gray Hair | 0.55 | 0.63 | 0.61 | 0.69 | 0.6 | 0.7 | 0.66 | 0.75 | 0.66 | 0.75 |
| Heavy Makeup | 0.62 | 1.38 | 0.62 | 1.42 | 0.63 | 1.46 | 0.64 | 1.67 | 0.68 | 1.8 |
| High Cheekbones | 0.46 | 1.35 | 0.48 | 1.43 | 0.51 | 1.58 | 0.59 | 1.82 | 0.7 | 2.1 |
| Male | 0.49 | 1.22 | 0.49 | 1.3 | 0.5 | 1.31 | 0.53 | 1.53 | 0.58 | 1.65 |
| Mouth Slightly Open | 0.4 | 1.39 | 0.43 | 1.51 | 0.47 | 1.56 | 0.63 | 2.04 | 0.78 | 2.82 |
| Mustache | 0.49 | 0.54 | 0.53 | 0.58 | 0.58 | 0.64 | 0.57 | 0.64 | 0.51 | 0.56 |
| Narrow Eyes | 0.65 | 0.87 | 0.65 | 0.86 | 0.71 | 0.94 | 0.69 | 0.92 | 0.7 | 0.91 |
| No Beard | 0.84 | 1.35 | 0.87 | 1.37 | 0.86 | 1.4 | 0.89 | 1.48 | 0.87 | 1.44 |
| Oval Face | 0.76 | 1.83 | 0.78 | 1.84 | 0.79 | 1.85 | 0.78 | 1.86 | 0.76 | 1.99 |
| Pale Skin | 0.6 | 0.68 | 0.62 | 0.71 | 0.65 | 0.74 | 0.57 | 0.68 | 0.6 | 0.69 |
| Pointy Nose | 0.76 | 1.84 | 0.78 | 1.87 | 0.79 | 1.9 | 0.78 | 1.88 | 0.75 | 1.86 |
| Receding Hairline | 0.69 | 0.87 | 0.73 | 0.91 | 0.73 | 0.92 | 0.76 | 0.97 | 0.75 | 0.97 |
| Rosy Cheeks | 0.64 | 0.77 | 0.62 | 0.74 | 0.62 | 0.74 | 0.68 | 0.8 | 0.67 | 0.78 |
| Sideburns | 0.6 | 0.71 | 0.64 | 0.78 | 0.63 | 0.76 | 0.71 | 0.85 | 0.65 | 0.78 |
| Smiling | 0.39 | 1.27 | 0.39 | 1.39 | 0.41 | 1.44 | 0.48 | 1.66 | 0.71 | 2.03 |
| Straight Hair | 0.76 | 1.37 | 0.79 | 1.41 | 0.82 | 1.47 | 0.8 | 1.5 | 0.74 | 1.34 |
| Wavy Hair | 0.73 | 1.53 | 0.74 | 1.55 | 0.73 | 1.69 | 0.75 | 1.78 | 0.73 | 1.83 |
| Wearing Earrings | 0.81 | 1.42 | 0.85 | 1.48 | 0.82 | 1.43 | 0.83 | 1.43 | 0.82 | 1.42 |
| Wearing Hat | 0.6 | 0.68 | 0.64 | 0.73 | 0.62 | 0.7 | 0.66 | 0.76 | 0.65 | 0.76 |
| Wearing Lipstick | 0.4 | 1.1 | 0.39 | 1.16 | 0.41 | 1.32 | 0.43 | 1.51 | 0.44 | 1.67 |
| Wearing Necklace | 0.75 | 1.01 | 0.77 | 1.05 | 0.77 | 1.02 | 0.8 | 1.07 | 0.74 | 0.99 |
| Wearing Necktie | 0.7 | 0.86 | 0.67 | 0.83 | 0.72 | 0.87 | 0.77 | 0.93 | 0.75 | 0.93 |
| Young | 0.82 | 1.46 | 0.85 | 1.5 | 0.88 | 1.57 | 0.85 | 1.54 | 0.87 | 1.66 |

Table 25: CelebA Error Bands with initial sample 100: Average standard errors for error and churn across baselines for each dataset and network across 100 runs.

| Dataset | network | cold | warm | s-perturb | mixup | ls | co-dist | anchor | distill |
|---|---|---|---|---|---|---|---|---|---|
| 5 o Clock Shadow | convnet-1 | 7.17 | N/A | 5.75 | 5.27 | 4.75 | 5.39 | N/A | **1.0** |
| | convnet-2 | 7.26 | N/A | N/A | 6.28 | 4.75 | 5.4 | N/A | **1.11** |
| | convnet-4 | 7.64 | N/A | 6.71 | 6.29 | 5.41 | 5.79 | N/A | **1.29** |
| | convnet-8 | 6.52 | N/A | N/A | 6.0 | 5.51 | 5.86 | N/A | **2.16** |
| | convnet-16 | 5.81 | N/A | 5.09 | 5.15 | 4.29 | 4.96 | N/A | **1.19** |
| Arched Eyebrows | convnet-1 | 19.42 | N/A | N/A | 16.14 | N/A | 15.52 | N/A | **12.26** |
| | convnet-2 | 19.72 | 17.63 | 17.96 | 16.18 | 19.56 | 16.19 | N/A | **6.66** |
| | convnet-4 | 21.16 | N/A | N/A | 18.0 | N/A | 17.15 | N/A | **12.34** |
| | convnet-8 | 20.23 | N/A | 18.22 | 16.77 | 16.93 | 17.3 | N/A | **3.84** |
| | convnet-16 | 18.21 | N/A | 17.59 | 16.43 | 16.83 | 16.37 | N/A | **11.43** |
| Attractive | convnet-1 | 22.41 | N/A | **20.62** | N/A | N/A | N/A | N/A | N/A |
| | convnet-2 | 25.48 | 22.66 | 22.71 | 23.22 | N/A | **20.95** | 23.52 | N/A |
| | convnet-4 | 23.51 | N/A | 21.25 | N/A | 22.17 | 20.04 | N/A | **13.73** |
| | convnet-8 | 24.05 | 22.27 | 22.04 | 22.29 | 22.73 | 21.09 | 22.41 | **9.39** |
| | convnet-16 | 24.17 | N/A | 21.85 | 21.22 | 22.03 | 20.32 | 21.95 | **12.02** |
| Bags Under Eyes | convnet-1 | 13.52 | N/A | 11.88 | 10.28 | N/A | 10.07 | N/A | **2.11** |
| | convnet-2 | 14.87 | N/A | 12.15 | 12.12 | 13.11 | 11.32 | N/A | **3.73** |
| | convnet-4 | 13.15 | N/A | 12.19 | 11.77 | 11.56 | 10.52 | N/A | **2.44** |
| | convnet-8 | 12.87 | N/A | 11.75 | 11.18 | 9.97 | 10.96 | N/A | **2.27** |
| | convnet-16 | 12.13 | N/A | 10.53 | 9.29 | 7.24 | 8.47 | N/A | **1.88** |
| Bald | convnet-1 | 1.25 | 1.32 | 1.05 | 0.99 | 0.63 | 0.9 | 0.71 | **0.27** |
| | convnet-2 | 1.23 | N/A | 0.98 | 1.06 | 0.63 | 0.95 | 0.69 | **0.15** |
| | convnet-4 | 1.31 | N/A | 1.34 | N/A | 0.92 | 1.22 | N/A | **0.34** |
| | convnet-8 | 1.24 | 1.25 | 0.91 | 0.96 | 0.59 | 0.86 | 0.72 | **0.3** |
| | convnet-16 | 1.03 | N/A | 0.74 | 0.86 | 0.56 | 0.82 | 0.56 | **0.25** |
| Bangs | convnet-1 | 6.92 | 6.31 | 6.28 | 6.08 | 6.28 | 5.74 | 5.89 | **3.23** |
| | convnet-2 | 7.44 | 6.76 | 6.52 | 6.28 | 6.34 | 6.21 | 6.46 | **2.92** |
| | convnet-4 | 7.78 | 7.21 | 7.08 | 6.87 | 7.04 | 6.38 | 6.63 | **3.48** |
| | convnet-8 | 8.36 | 8.07 | 7.7 | 7.84 | 7.65 | 7.39 | 7.53 | **4.0** |
| | convnet-16 | 7.45 | N/A | 7.45 | 6.74 | N/A | 5.75 | N/A | **5.42** |
| Big Lips | convnet-1 | 7.46 | N/A | 5.89 | N/A | N/A | N/A | N/A | **1.24** |
| | convnet-2 | 7.39 | N/A | 5.86 | N/A | N/A | N/A | N/A | **1.44** |
| | convnet-4 | 6.08 | N/A | 4.7 | N/A | N/A | N/A | N/A | **1.34** |
| | convnet-8 | **6.27** | N/A | N/A | N/A | N/A | N/A | N/A | N/A |
| | convnet-16 | 4.68 | N/A | 4.04 | 4.32 | **3.77** | N/A | N/A | N/A |
| Big Nose | convnet-1 | 14.84 | N/A | 12.46 | 13.11 | N/A | 12.06 | N/A | **2.66** |
| | convnet-2 | **15.35** | N/A | N/A | N/A | N/A | N/A | N/A | N/A |
| | convnet-4 | 15.2 | N/A | 13.56 | 13.64 | N/A | 12.48 | N/A | **2.6** |
| | convnet-8 | 14.0 | N/A | 13.31 | 13.45 | 12.86 | 12.73 | N/A | **2.74** |
| | convnet-16 | 13.81 | N/A | 13.19 | 13.04 | 12.65 | 12.5 | N/A | **3.02** |
| Black Hair | convnet-1 | 15.1 | N/A | 12.96 | N/A | 13.72 | 12.45 | 13.88 | **6.63** |
| | convnet-2 | 14.7 | N/A | N/A | N/A | N/A | 13.08 | N/A | **10.36** |
| | convnet-4 | 15.32 | N/A | N/A | N/A | **14.68** | N/A | N/A | N/A |
| | convnet-8 | 14.2 | N/A | 14.35 | N/A | N/A | 13.16 | 13.28 | **8.52** |
| | convnet-16 | 14.41 | 14.52 | 14.02 | N/A | 14.04 | 12.81 | 14.12 | **8.58** |
| Blond Hair | convnet-1 | 7.95 | 6.95 | 6.57 | 6.47 | 6.57 | 6.07 | 5.69 | **3.28** |
| | convnet-2 | 8.36 | 7.89 | 7.3 | 7.24 | 7.59 | 6.93 | 6.89 | **2.99** |
| | convnet-4 | 8.33 | 7.97 | 7.81 | 7.21 | 7.71 | 7.11 | 7.71 | **3.37** |
| | convnet-8 | 9.08 | 8.69 | 8.57 | 8.14 | 8.58 | 7.8 | 8.16 | **3.18** |
| | convnet-16 | 9.37 | N/A | 9.0 | 8.56 | 8.99 | 8.1 | 8.73 | **5.74** |

Table 26: Results for CelebA tasks under churn at cold accuracy metric across different sizes of convolutional networks. Part 1 of 4.

| Dataset | network | cold | warm | s-perturb | mixup | ls | co-dist | anchor | distill |
|---|---|---|---|---|---|---|---|---|---|
| | convnet-1 | **0.03** | N/A | **0.03** | N/A | N/A | N/A | N/A | N/A |
| | convnet-2 | **0.22** | N/A | **0.22** | N/A | N/A | N/A | N/A | N/A |
| Blurry | convnet-4 | **0.21** | N/A | **0.21** | 0.27 | N/A | N/A | N/A | N/A |
| | convnet-8 | **0.28** | N/A | **0.28** | N/A | N/A | N/A | N/A | N/A |
| | convnet-16 | 0.22 | N/A | **0.2** | N/A | 0.2 | N/A | N/A | N/A |
| | convnet-1 | 14.06 | N/A | 13.06 | 10.9 | 13.23 | 11.82 | N/A | **6.16** |
| | convnet-2 | 13.06 | N/A | N/A | N/A | N/A | N/A | N/A | **6.37** |
| Brown Hair | convnet-4 | 13.46 | N/A | 13.05 | 11.23 | N/A | 11.31 | N/A | **7.81** |
| | convnet-8 | 13.2 | 12.61 | N/A | 11.05 | 13.2 | 11.86 | N/A | **3.66** |
| | convnet-16 | 13.79 | N/A | 12.79 | 10.76 | 12.9 | 11.53 | N/A | **3.63** |
| | convnet-1 | 7.31 | N/A | 6.19 | 6.07 | 5.52 | 5.51 | N/A | **1.02** |
| | convnet-2 | 7.02 | N/A | 6.23 | 6.38 | 5.65 | 6.32 | 7.4 | **1.0** |
| Bushy Eyebrows | convnet-4 | 7.01 | N/A | 6.4 | 6.48 | N/A | 6.56 | N/A | **1.34** |
| | convnet-8 | 6.71 | N/A | 6.09 | 6.44 | 5.8 | N/A | N/A | **2.16** |
| | convnet-16 | 5.15 | N/A | 5.24 | N/A | **4.49** | N/A | N/A | N/A |
| | convnet-1 | 2.21 | N/A | 1.87 | 1.91 | 1.22 | 1.78 | N/A | **0.4** |
| | convnet-2 | 2.33 | N/A | 2.11 | 2.23 | 1.56 | 1.94 | N/A | **0.43** |
| Chubby | convnet-4 | 2.57 | N/A | 2.12 | 2.17 | 1.62 | 2.18 | N/A | **0.41** |
| | convnet-8 | 2.04 | N/A | 1.6 | 2.08 | 1.32 | 1.7 | N/A | **0.49** |
| | convnet-16 | 1.81 | N/A | 1.66 | 1.7 | 1.12 | 1.58 | N/A | **0.43** |
| | convnet-1 | 2.04 | N/A | 1.43 | 1.6 | 1.0 | 1.55 | N/A | **0.37** |
| | convnet-2 | 1.39 | N/A | 1.45 | N/A | 0.64 | 1.22 | N/A | **0.18** |
| Double Chin | convnet-4 | 2.04 | N/A | 1.57 | 2.14 | 1.15 | 1.63 | N/A | **0.47** |
| | convnet-8 | 2.38 | N/A | 1.7 | 2.08 | 1.38 | 1.86 | N/A | **0.55** |
| | convnet-16 | 1.26 | N/A | 1.22 | N/A | 0.7 | N/A | N/A | **0.35** |
| | convnet-1 | 3.14 | N/A | 2.94 | 2.6 | 2.5 | 2.42 | 2.67 | **1.96** |
| | convnet-2 | 2.91 | N/A | 2.78 | 2.73 | 2.46 | 2.49 | 2.71 | **1.46** |
| Eyeglasses | convnet-4 | 3.07 | N/A | 2.75 | 2.65 | 2.44 | 2.39 | 2.73 | **0.77** |
| | convnet-8 | 3.28 | N/A | 2.91 | 2.92 | 2.83 | **2.61** | 2.85 | N/A |
| | convnet-16 | 3.11 | N/A | N/A | N/A | 2.58 | **2.51** | N/A | N/A |
| | convnet-1 | 3.19 | N/A | N/A | N/A | 1.93 | N/A | N/A | **1.01** |
| | convnet-2 | 3.46 | N/A | 3.28 | 3.48 | 2.07 | 2.46 | N/A | **0.58** |
| Goatee | convnet-4 | 3.22 | N/A | 2.79 | 3.01 | 1.53 | 2.39 | N/A | **0.47** |
| | convnet-8 | 3.05 | N/A | 2.54 | 2.69 | 1.24 | 1.93 | 2.97 | **0.39** |
| | convnet-16 | 2.66 | N/A | 2.46 | 2.45 | 1.13 | 1.95 | N/A | **0.4** |
| | convnet-1 | 2.63 | N/A | 2.25 | 2.32 | 1.89 | 2.04 | N/A | **0.76** |
| | convnet-2 | 2.35 | N/A | 2.31 | 2.04 | 2.03 | 2.23 | 2.51 | **0.49** |
| Gray Hair | convnet-4 | 2.67 | N/A | 2.43 | 2.34 | 2.18 | 2.2 | 2.75 | **0.88** |
| | convnet-8 | 3.06 | N/A | 3.18 | 2.95 | 2.83 | 2.89 | 2.85 | **1.42** |
| | convnet-16 | 3.21 | 2.66 | 2.71 | 2.63 | 1.98 | 2.19 | 1.93 | **0.67** |
| | convnet-1 | 17.61 | N/A | 16.25 | 16.27 | N/A | **14.45** | N/A | N/A |
| | convnet-2 | 17.56 | N/A | 16.22 | 15.81 | N/A | **14.73** | N/A | N/A |
| Heavy Makeup | convnet-4 | 19.8 | N/A | 18.69 | 17.95 | N/A | **16.71** | 18.03 | N/A |
| | convnet-8 | 20.46 | N/A | 20.03 | 20.16 | N/A | **17.65** | N/A | N/A |
| | convnet-16 | 22.4 | N/A | 20.09 | 19.48 | 21.11 | 18.73 | 20.07 | **12.52** |
| | convnet-1 | 20.08 | N/A | **17.11** | N/A | N/A | N/A | N/A | N/A |
| | convnet-2 | 20.35 | N/A | 18.38 | N/A | N/A | **16.79** | N/A | N/A |
| High Cheekbones | convnet-4 | 23.14 | N/A | 20.56 | N/A | N/A | **18.68** | N/A | N/A |
| | convnet-8 | 24.9 | N/A | **22.56** | N/A | N/A | N/A | N/A | N/A |
| | convnet-16 | 29.46 | N/A | 27.22 | 26.76 | N/A | **25.42** | N/A | N/A |

Table 27: Results for CelebA tasks under churn at cold accuracy metric across different sizes of convolutional networks. Part 2 of 4.

| Dataset | network | cold | warm | s-perturb | mixup | ls | co-dist | anchor | distill |
|---|---|---|---|---|---|---|---|---|---|
| Male | convnet-1 | 14.08 | N/A | 10.88 | 11.53 | 10.88 | **10.68** | 10.74 | N/A |
| | convnet-2 | 13.93 | N/A | **12.23** | 12.7 | N/A | N/A | N/A | N/A |
| | convnet-4 | 15.16 | N/A | 13.89 | 13.32 | 12.99 | 12.48 | 12.5 | **10.07** |
| | convnet-8 | 16.31 | N/A | 15.26 | 15.33 | 14.92 | 14.62 | N/A | **12.87** |
| | convnet-16 | 16.86 | 15.56 | 15.79 | 15.77 | 15.91 | 14.98 | 14.92 | **10.73** |
| Mouth Slightly Open | convnet-1 | 22.17 | N/A | 17.62 | N/A | N/A | **16.96** | N/A | N/A |
| | convnet-2 | 22.29 | N/A | **20.38** | N/A | N/A | N/A | N/A | N/A |
| | convnet-4 | **23.01** | N/A | N/A | N/A | N/A | N/A | N/A | N/A |
| | convnet-8 | **26.52** | N/A | N/A | N/A | N/A | N/A | N/A | N/A |
| | convnet-16 | 31.25 | 29.07 | 29.08 | 28.74 | 28.95 | 27.64 | 30.03 | **23.06** |
| Mustache | convnet-1 | 0.7 | N/A | 0.44 | 0.64 | 0.21 | 0.52 | N/A | **0.07** |
| | convnet-2 | 0.61 | N/A | 0.66 | N/A | 0.42 | 0.8 | N/A | **0.17** |
| | convnet-4 | 0.77 | N/A | 0.4 | 0.86 | 0.38 | 0.68 | N/A | **0.11** |
| | convnet-8 | 0.62 | N/A | 0.5 | N/A | 0.41 | 0.65 | 0.41 | **0.18** |
| | convnet-16 | 0.54 | N/A | 0.38 | N/A | 0.34 | N/A | N/A | **0.14** |
| Narrow Eyes | convnet-1 | 0.97 | N/A | 0.36 | 0.54 | 0.74 | 0.66 | N/A | **0.09** |
| | convnet-2 | 1.0 | N/A | 0.75 | 1.07 | N/A | N/A | N/A | **0.34** |
| | convnet-4 | 0.48 | N/A | **0.32** | N/A | 0.56 | N/A | N/A | N/A |
| | convnet-8 | **0.48** | N/A | N/A | N/A | N/A | N/A | N/A | N/A |
| | convnet-16 | 0.19 | N/A | **0.18** | N/A | N/A | N/A | N/A | N/A |
| No Beard | convnet-1 | 12.32 | N/A | 10.85 | 11.5 | N/A | 9.95 | 11.34 | **7.4** |
| | convnet-2 | 12.82 | 11.52 | 10.9 | 11.05 | 10.91 | 9.95 | 11.34 | **4.02** |
| | convnet-4 | 12.21 | N/A | 11.92 | 11.64 | N/A | 10.59 | N/A | **6.4** |
| | convnet-8 | 13.59 | N/A | 12.55 | 11.99 | 12.61 | 10.91 | 13.0 | **6.16** |
| | convnet-16 | 13.75 | 12.59 | 12.35 | 12.1 | 12.19 | 11.45 | 12.77 | **6.25** |
| Oval Face | convnet-1 | 14.15 | N/A | 12.26 | N/A | N/A | N/A | N/A | **2.43** |
| | convnet-2 | 16.24 | N/A | 14.91 | 14.84 | N/A | N/A | N/A | **3.21** |
| | convnet-4 | 13.96 | N/A | 12.65 | 13.23 | N/A | 14.23 | N/A | **2.77** |
| | convnet-8 | 12.14 | N/A | 11.65 | 10.94 | 10.82 | N/A | N/A | **2.54** |
| | convnet-16 | 10.69 | N/A | N/A | 8.81 | N/A | N/A | N/A | **1.91** |
| Pale Skin | convnet-1 | 2.09 | N/A | 2.08 | 1.87 | 1.5 | 1.98 | N/A | **1.26** |
| | convnet-2 | 1.93 | 2.08 | 1.51 | 1.66 | 1.04 | 1.58 | 2.27 | **0.34** |
| | convnet-4 | 2.13 | N/A | 1.87 | 1.94 | 1.42 | 1.9 | N/A | **0.46** |
| | convnet-8 | 1.98 | 1.88 | 1.4 | 1.41 | 0.83 | 1.52 | 1.89 | **0.29** |
| | convnet-16 | 1.44 | N/A | N/A | N/A | **1.0** | 1.56 | N/A | N/A |
| Pointy Nose | convnet-1 | 15.42 | N/A | 13.99 | N/A | N/A | N/A | N/A | **2.65** |
| | convnet-2 | 16.07 | N/A | 14.29 | **13.62** | N/A | N/A | N/A | N/A |
| | convnet-4 | 13.81 | N/A | 12.23 | 12.73 | N/A | N/A | N/A | **3.06** |
| | convnet-8 | 11.64 | N/A | 9.99 | N/A | 11.61 | N/A | N/A | **2.3** |
| | convnet-16 | 9.84 | N/A | 10.12 | 9.1 | 9.44 | N/A | N/A | **3.94** |
| Receding Hairline | convnet-1 | 4.18 | N/A | 3.37 | 3.07 | 2.55 | 2.96 | 3.44 | **0.63** |
| | convnet-2 | 4.42 | N/A | 4.32 | 4.37 | 3.7 | 3.89 | N/A | **1.08** |
| | convnet-4 | 4.62 | N/A | 4.17 | N/A | 3.31 | 3.76 | N/A | **1.44** |
| | convnet-8 | 4.5 | N/A | 4.42 | 4.38 | 3.83 | 4.24 | N/A | **1.83** |
| | convnet-16 | 3.76 | N/A | 3.38 | 2.97 | 2.5 | N/A | N/A | **0.89** |
| Rosy Cheeks | convnet-1 | 2.54 | N/A | 2.09 | 2.05 | 1.16 | 1.97 | N/A | **0.38** |
| | convnet-2 | 2.38 | N/A | 2.3 | 2.14 | 1.25 | 2.28 | N/A | **0.35** |
| | convnet-4 | 3.0 | N/A | 2.43 | 2.51 | 1.9 | 2.41 | N/A | **0.57** |
| | convnet-8 | 2.23 | N/A | 1.86 | 1.86 | 1.44 | N/A | N/A | **0.45** |
| | convnet-16 | 1.58 | N/A | 1.37 | 1.48 | 0.84 | N/A | N/A | **0.33** |

Table 28: Results for CelebA tasks under churn at cold accuracy metric across different sizes of convolutional networks. Part 3 of 4.

| Dataset | network | cold | warm | s-perturb | mixup | ls | co-dist | anchor | distill |
|---------|---------|------|------|-----------|-------|-----|---------|--------|---------|
| Sideburns | convnet-1 | 2.94 | N/A | 2.86 | 2.44 | 1.1 | 2.26 | N/A | **0.39** |
| | convnet-2 | 2.81 | 2.55 | 2.72 | 2.28 | 1.15 | 2.03 | 2.85 | **0.4** |
| | convnet-4 | 3.64 | N/A | 2.43 | 2.78 | 1.39 | 1.98 | 2.89 | **0.38** |
| | convnet-8 | 2.74 | N/A | 2.48 | 2.56 | 1.12 | 1.72 | N/A | **0.24** |
| | convnet-16 | 3.68 | 3.45 | 3.0 | 3.01 | 1.97 | 2.68 | 3.34 | **0.73** |
| Smiling | convnet-1 | 15.69 | N/A | 13.19 | N/A | N/A | 12.1 | 11.68 | **7.9** |
| | convnet-2 | 14.6 | N/A | 12.49 | N/A | N/A | **12.1** | N/A | N/A |
| | convnet-4 | 15.51 | N/A | 13.95 | N/A | N/A | 12.93 | 12.93 | **10.51** |
| | convnet-8 | 18.75 | N/A | 15.82 | 16.83 | N/A | **15.42** | N/A | N/A |
| | convnet-16 | 23.22 | N/A | 20.93 | 20.69 | 20.7 | **19.72** | 21.27 | N/A |
| Straight Hair | convnet-1 | 5.75 | N/A | 3.71 | 4.29 | 5.56 | 4.46 | N/A | **0.85** |
| | convnet-2 | 6.46 | N/A | 5.28 | 6.14 | N/A | N/A | N/A | **1.08** |
| | convnet-4 | 6.05 | N/A | 5.02 | N/A | N/A | N/A | N/A | **1.28** |
| | convnet-8 | 4.52 | N/A | 4.55 | N/A | N/A | N/A | N/A | **1.2** |
| | convnet-16 | 5.47 | N/A | 5.58 | N/A | **4.68** | N/A | N/A | N/A |
| Wavy Hair | convnet-1 | 19.16 | N/A | 17.36 | 17.53 | N/A | 16.15 | N/A | **9.73** |
| | convnet-2 | 20.05 | N/A | 19.23 | 19.77 | N/A | **18.29** | N/A | N/A |
| | convnet-4 | 19.38 | N/A | 17.93 | 17.01 | 20.82 | 16.92 | N/A | **7.53** |
| | convnet-8 | 19.13 | N/A | 17.86 | 17.5 | 18.6 | 17.5 | 18.9 | **12.31** |
| | convnet-16 | 19.23 | N/A | 19.7 | 17.36 | 18.97 | 16.75 | N/A | **11.37** |
| Wearing Earrings | convnet-1 | 12.07 | 10.52 | 9.43 | 8.34 | 7.36 | 9.14 | 11.0 | **1.78** |
| | convnet-2 | 10.98 | N/A | 9.78 | 9.8 | N/A | 8.77 | N/A | **1.97** |
| | convnet-4 | 10.41 | N/A | 9.21 | 9.22 | 8.72 | 8.79 | N/A | **2.15** |
| | convnet-8 | 7.05 | N/A | 7.31 | 7.54 | 6.44 | N/A | N/A | **1.71** |
| | convnet-16 | 7.73 | N/A | 7.58 | 6.49 | 5.52 | N/A | N/A | **1.53** |
| Wearing Hat | convnet-1 | 2.73 | 2.35 | 2.26 | 2.3 | 2.11 | 2.1 | 2.09 | **1.23** |
| | convnet-2 | 2.33 | N/A | 2.25 | 2.39 | 2.06 | 2.22 | 2.18 | **1.44** |
| | convnet-4 | 2.57 | 2.58 | 2.42 | 2.57 | 2.15 | 2.2 | 2.17 | **0.93** |
| | convnet-8 | 3.0 | 2.83 | 2.71 | 2.89 | 2.49 | 2.32 | 2.54 | **1.01** |
| | convnet-16 | 3.09 | N/A | 3.03 | N/A | 2.83 | 2.81 | 2.73 | **2.42** |
| Wearing Lipstick | convnet-1 | 15.83 | N/A | **14.95** | N/A | N/A | N/A | N/A | N/A |
| | convnet-2 | 14.95 | 13.66 | 13.89 | 13.55 | 13.95 | 13.03 | 12.26 | **8.63** |
| | convnet-4 | 17.23 | N/A | 15.16 | 15.08 | 15.28 | 14.37 | N/A | **11.92** |
| | convnet-8 | 18.3 | N/A | 16.93 | 16.46 | N/A | **15.27** | N/A | N/A |
| | convnet-16 | 20.15 | 17.99 | 17.87 | 17.28 | 18.2 | 16.93 | 17.86 | **12.74** |
| Wearing Necklace | convnet-1 | 2.08 | N/A | **1.62** | N/A | 1.67 | N/A | N/A | N/A |
| | convnet-2 | 2.37 | N/A | **1.87** | N/A | 2.21 | N/A | N/A | N/A |
| | convnet-4 | 1.81 | N/A | **1.63** | N/A | N/A | N/A | N/A | N/A |
| | convnet-8 | 2.04 | N/A | N/A | N/A | **1.76** | N/A | N/A | N/A |
| | convnet-16 | 1.46 | N/A | 1.42 | N/A | **1.06** | N/A | N/A | N/A |
| Wearing Necktie | convnet-1 | 4.21 | N/A | 3.76 | 3.83 | 3.45 | 3.66 | 3.57 | **3.26** |
| | convnet-2 | 4.03 | N/A | 3.71 | 3.8 | 3.82 | 3.39 | 3.64 | **1.06** |
| | convnet-4 | 4.45 | N/A | 4.28 | N/A | 4.24 | **4.03** | 4.37 | N/A |
| | convnet-8 | 4.75 | 4.04 | 3.88 | 4.02 | 3.94 | 3.51 | 4.13 | **2.01** |
| | convnet-16 | 4.78 | 4.06 | 4.03 | 3.96 | 4.2 | 3.95 | 4.0 | **1.72** |
| Young | convnet-1 | 13.17 | N/A | N/A | N/A | N/A | **10.53** | N/A | N/A |
| | convnet-2 | 13.9 | N/A | 12.1 | 12.29 | N/A | 11.5 | N/A | **4.24** |
| | convnet-4 | 13.59 | N/A | 12.1 | 11.88 | N/A | N/A | N/A | **5.21** |
| | convnet-8 | 13.73 | N/A | 12.73 | 12.24 | N/A | **11.89** | N/A | N/A |
| | convnet-16 | 12.62 | N/A | 12.06 | 11.93 | N/A | 11.22 | N/A | **6.54** |

Table 29: Results for CelebA tasks under churn at cold accuracy metric across different sizes of convolutional networks. Part 4 of 4.

| Dataset | convnet-1 | | convnet-2 | | convnet-4 | | convnet-8 | | convnet-16 | |
|---|---|---|---|---|---|---|---|---|---|---|
| | Error | Churn | Error | Churn | Error | Churn | Error | Churn | Error | Churn |
| 5 o Clock Shadow | 0.79 | 0.98 | 0.78 | 0.98 | 0.78 | 0.98 | 0.83 | 1.06 | 0.83 | 1.07 |
| Arched Eyebrows | 0.82 | 1.2 | 0.81 | 1.23 | 0.84 | 1.21 | 0.84 | 1.38 | 0.81 | 1.36 |
| Attractive | 0.37 | 0.79 | 0.36 | 0.92 | 0.37 | 0.83 | 0.38 | 0.97 | 0.4 | 0.85 |
| Bags Under Eyes | 0.84 | 1.2 | 0.87 | 1.25 | 0.86 | 1.24 | 0.86 | 1.32 | 0.87 | 1.38 |
| Bald | 0.38 | 0.42 | 0.43 | 0.47 | 0.48 | 0.54 | 0.44 | 0.48 | 0.54 | 0.58 |
| Bangs | 0.77 | 0.83 | 0.8 | 0.88 | 0.8 | 0.85 | 0.83 | 0.93 | 0.82 | 0.91 |
| Big Lips | 0.8 | 1.51 | 0.82 | 1.55 | 0.81 | 1.53 | 0.8 | 1.53 | 0.76 | 1.56 |
| Big Nose | 0.81 | 1.23 | 0.82 | 1.18 | 0.82 | 1.25 | 0.82 | 1.36 | 0.81 | 1.37 |
| Black Hair | 0.78 | 0.96 | 0.79 | 1.01 | 0.78 | 0.97 | 0.78 | 0.98 | 0.8 | 1.04 |
| Blond Hair | 0.8 | 0.86 | 0.8 | 0.91 | 0.82 | 0.95 | 0.82 | 0.94 | 0.83 | 0.96 |
| Blurry | 0.59 | 0.66 | 0.55 | 0.66 | 0.57 | 0.67 | 0.58 | 0.67 | 0.55 | 0.63 |
| Brown Hair | 0.83 | 1.2 | 0.83 | 1.14 | 0.84 | 1.17 | 0.87 | 1.21 | 0.83 | 1.3 |
| Bushy Eyebrows | 0.83 | 1.09 | 0.82 | 1.12 | 0.86 | 1.16 | 0.85 | 1.22 | 0.82 | 1.14 |
| Chubby | 0.62 | 0.72 | 0.65 | 0.77 | 0.65 | 0.78 | 0.64 | 0.76 | 0.64 | 0.77 |
| Double Chin | 0.58 | 0.68 | 0.6 | 0.69 | 0.59 | 0.69 | 0.6 | 0.72 | 0.59 | 0.69 |
| Eyeglasses | 0.6 | 0.68 | 0.64 | 0.7 | 0.63 | 0.68 | 0.61 | 0.69 | 0.66 | 0.73 |
| Goatee | 0.67 | 0.8 | 0.66 | 0.78 | 0.69 | 0.81 | 0.69 | 0.82 | 0.66 | 0.8 |
| Gray Hair | 0.54 | 0.59 | 0.54 | 0.59 | 0.61 | 0.68 | 0.62 | 0.72 | 0.59 | 0.67 |
| Heavy Makeup | 0.56 | 0.74 | 0.58 | 0.75 | 0.59 | 0.83 | 0.6 | 0.92 | 0.59 | 0.92 |
| High Cheekbones | 0.45 | 0.76 | 0.45 | 0.77 | 0.49 | 1.09 | 0.47 | 1.05 | 0.5 | 1.31 |
| Male | 0.45 | 0.54 | 0.47 | 0.61 | 0.45 | 0.65 | 0.48 | 0.64 | 0.47 | 0.71 |
| Mouth Slightly Open | 0.39 | 0.67 | 0.4 | 0.83 | 0.38 | 0.92 | 0.45 | 1.44 | 0.57 | 1.78 |
| Mustache | 0.5 | 0.55 | 0.54 | 0.61 | 0.55 | 0.62 | 0.5 | 0.57 | 0.54 | 0.61 |
| Narrow Eyes | 0.73 | 0.97 | 0.76 | 1.06 | 0.76 | 1.02 | 0.76 | 1.04 | 0.73 | 0.97 |
| No Beard | 0.83 | 1.01 | 0.83 | 1.07 | 0.84 | 1.11 | 0.85 | 1.11 | 0.86 | 1.19 |
| Oval Face | 0.79 | 1.54 | 0.83 | 1.51 | 0.79 | 1.57 | 0.77 | 1.59 | 0.79 | 1.85 |
| Pale Skin | 0.63 | 0.71 | 0.59 | 0.68 | 0.63 | 0.74 | 0.61 | 0.72 | 0.61 | 0.73 |
| Pointy Nose | 0.79 | 1.46 | 0.79 | 1.45 | 0.78 | 1.55 | 0.77 | 1.66 | 0.75 | 1.62 |
| Receding Hairline | 0.66 | 0.78 | 0.7 | 0.85 | 0.75 | 0.92 | 0.71 | 0.87 | 0.72 | 0.91 |
| Rosy Cheeks | 0.67 | 0.8 | 0.69 | 0.83 | 0.69 | 0.89 | 0.69 | 0.86 | 0.66 | 0.79 |
| Sideburns | 0.66 | 0.77 | 0.62 | 0.73 | 0.65 | 0.77 | 0.63 | 0.78 | 0.62 | 0.96 |
| Smiling | 0.35 | 0.59 | 0.34 | 0.51 | 0.35 | 0.69 | 0.39 | 0.85 | 0.42 | 1.35 |
| Straight Hair | 0.81 | 1.38 | 0.83 | 1.48 | 0.83 | 1.42 | 0.81 | 1.39 | 0.82 | 1.76 |
| Wavy Hair | 0.72 | 0.98 | 0.73 | 1.06 | 0.72 | 1.06 | 0.71 | 1.09 | 0.72 | 1.14 |
| Wearing Earrings | 0.87 | 1.32 | 0.84 | 1.27 | 0.87 | 1.41 | 0.84 | 1.36 | 0.85 | 1.38 |
| Wearing Hat | 0.53 | 0.55 | 0.57 | 0.58 | 0.62 | 0.65 | 0.59 | 0.63 | 0.63 | 0.67 |
| Wearing Lipstick | 0.4 | 0.56 | 0.39 | 0.61 | 0.4 | 0.66 | 0.38 | 0.62 | 0.39 | 0.7 |
| Wearing Necklace | 0.75 | 1.05 | 0.78 | 1.06 | 0.8 | 1.11 | 0.78 | 1.08 | 0.76 | 1.06 |
| Wearing Necktie | 0.66 | 0.7 | 0.71 | 0.75 | 0.71 | 0.78 | 0.73 | 0.78 | 0.71 | 0.78 |
| Young | 0.85 | 1.11 | 0.85 | 1.18 | 0.85 | 1.19 | 0.87 | 1.21 | 0.82 | 1.17 |

Table 30: CelebA Error Bands: Average standard errors for error and churn across baselines for each dataset and network across 100 runs.

| Dataset | network | cold | warm | s-perturb | mixup | ls | co-dist | anchor | distill |
|---|---|---|---|---|---|---|---|---|---|
| 5 o Clock Shadow | convnet-1 | 6.65 | N/A | 6.51 | 5.47 | N/A | N/A | N/A | **1.65** |
| | convnet-2 | 6.5 | N/A | 6.29 | N/A | N/A | N/A | N/A | **1.87** |
| | convnet-4 | 6.33 | N/A | 5.85 | 5.6 | N/A | N/A | N/A | **1.79** |
| | convnet-8 | 6.1 | N/A | N/A | N/A | N/A | N/A | N/A | **1.61** |
| | convnet-16 | 6.67 | N/A | 6.32 | 5.52 | N/A | 5.55 | N/A | **1.67** |
| Arched Eyebrows | convnet-1 | 15.6 | N/A | 12.64 | 12.44 | N/A | N/A | N/A | **3.73** |
| | convnet-2 | 15.8 | N/A | 14.01 | N/A | N/A | N/A | N/A | **5.24** |
| | convnet-4 | 14.87 | N/A | 13.34 | N/A | N/A | N/A | N/A | **3.92** |
| | convnet-8 | 15.52 | N/A | N/A | N/A | N/A | N/A | N/A | **5.16** |
| | convnet-16 | 15.72 | N/A | 12.99 | 12.91 | N/A | 12.17 | N/A | **3.8** |
| Attractive | convnet-1 | 16.61 | N/A | N/A | N/A | N/A | N/A | N/A | **5.33** |
| | convnet-2 | 16.67 | N/A | 14.58 | N/A | N/A | N/A | N/A | **4.25** |
| | convnet-4 | 16.01 | N/A | 15.36 | N/A | N/A | N/A | N/A | **4.15** |
| | convnet-8 | 16.63 | N/A | 14.49 | N/A | N/A | N/A | N/A | **4.02** |
| | convnet-16 | 16.45 | N/A | 14.96 | N/A | N/A | N/A | N/A | **4.05** |
| Bags Under Eyes | convnet-1 | 10.22 | N/A | 8.62 | N/A | N/A | N/A | N/A | **2.48** |
| | convnet-2 | 10.07 | N/A | 8.58 | N/A | N/A | N/A | N/A | **2.5** |
| | convnet-4 | 9.14 | N/A | N/A | N/A | N/A | N/A | N/A | **3.54** |
| | convnet-8 | 9.5 | N/A | 8.52 | N/A | N/A | N/A | N/A | **2.4** |
| | convnet-16 | **9.47** | N/A | N/A | N/A | N/A | N/A | N/A | N/A |
| Bald | convnet-1 | 1.61 | N/A | 1.3 | 1.32 | 1.15 | 1.2 | 1.13 | **0.44** |
| | convnet-2 | 1.58 | N/A | 1.35 | 1.3 | 1.24 | 1.29 | 1.44 | **0.5** |
| | convnet-4 | 1.38 | N/A | 1.24 | 1.27 | 1.1 | 1.22 | 1.02 | **0.39** |
| | convnet-8 | 1.42 | N/A | 1.33 | 1.25 | 1.17 | 1.21 | 1.1 | **0.45** |
| | convnet-16 | 1.14 | N/A | 1.17 | 1.15 | 0.93 | 1.06 | 0.79 | **0.33** |
| Bangs | convnet-1 | 5.69 | N/A | 4.91 | N/A | N/A | 4.21 | N/A | **1.76** |
| | convnet-2 | 5.64 | N/A | N/A | N/A | N/A | 4.38 | N/A | **1.82** |
| | convnet-4 | 5.51 | N/A | N/A | N/A | N/A | N/A | N/A | **2.58** |
| | convnet-8 | 5.48 | N/A | 5.28 | N/A | N/A | 4.58 | N/A | **2.21** |
| | convnet-16 | 5.52 | N/A | N/A | N/A | N/A | N/A | N/A | **2.02** |
| Big Lips | convnet-1 | 8.07 | N/A | 7.58 | N/A | N/A | N/A | N/A | **1.98** |
| | convnet-2 | 7.91 | N/A | 7.31 | N/A | N/A | N/A | N/A | **1.79** |
| | convnet-4 | **7.72** | N/A | N/A | N/A | N/A | N/A | N/A | N/A |
| | convnet-8 | 6.37 | N/A | N/A | N/A | N/A | N/A | N/A | **1.51** |
| | convnet-16 | 4.48 | N/A | N/A | N/A | N/A | N/A | N/A | **1.0** |
| Big Nose | convnet-1 | 12.13 | N/A | 10.63 | N/A | N/A | N/A | N/A | **4.06** |
| | convnet-2 | 12.06 | N/A | 10.75 | N/A | N/A | N/A | N/A | **2.86** |
| | convnet-4 | **11.45** | N/A | N/A | N/A | N/A | N/A | N/A | N/A |
| | convnet-8 | 11.16 | N/A | N/A | N/A | N/A | N/A | N/A | **2.8** |
| | convnet-16 | 11.82 | N/A | 10.64 | N/A | N/A | N/A | N/A | **2.81** |
| Black Hair | convnet-1 | 11.61 | N/A | 10.19 | N/A | N/A | 8.78 | N/A | **2.98** |
| | convnet-2 | 10.9 | N/A | 10.05 | N/A | N/A | N/A | N/A | **3.02** |
| | convnet-4 | 10.37 | N/A | 10.09 | N/A | N/A | N/A | N/A | **2.96** |
| | convnet-8 | 10.61 | N/A | 10.57 | N/A | N/A | N/A | N/A | **2.97** |
| | convnet-16 | 10.68 | N/A | 10.27 | N/A | N/A | N/A | N/A | **2.9** |
| Blond Hair | convnet-1 | 5.74 | N/A | N/A | N/A | N/A | N/A | N/A | **2.09** |
| | convnet-2 | 5.62 | N/A | 5.37 | N/A | N/A | 4.52 | N/A | **1.88** |
| | convnet-4 | 5.51 | N/A | 5.39 | 4.9 | N/A | 4.49 | N/A | **1.97** |
| | convnet-8 | 5.62 | N/A | 4.95 | 4.88 | 4.77 | 4.52 | 4.54 | **1.93** |
| | convnet-16 | 5.48 | N/A | 5.07 | 5.04 | 4.86 | 4.64 | 4.73 | **1.94** |

Table 31: Results for CelebA tasks under churn at cold accuracy metric across different sizes of convolutional networks with initial sample 10000. Part 1 of 4.

| Dataset | network | cold | warm | s-perturb | mixup | ls | co-dist | anchor | distill |
|---------|---------|------|------|-----------|-------|-----|---------|--------|---------|
| Blurry | convnet-1 | 0.08 | N/A | **0.06** | N/A | N/A | N/A | N/A | N/A |
| | convnet-2 | **0.08** | N/A | 0.09 | N/A | N/A | N/A | N/A | N/A |
| | convnet-4 | **0.04** | N/A | N/A | N/A | N/A | N/A | N/A | N/A |
| | convnet-8 | **0.08** | N/A | N/A | N/A | N/A | N/A | N/A | N/A |
| | convnet-16 | **0.04** | N/A | N/A | N/A | N/A | N/A | N/A | N/A |
| Brown Hair | convnet-1 | 11.37 | N/A | 10.55 | N/A | N/A | N/A | N/A | **3.63** |
| | convnet-2 | 11.18 | N/A | 11.02 | N/A | N/A | N/A | N/A | **2.88** |
| | convnet-4 | 10.91 | N/A | N/A | N/A | N/A | N/A | N/A | **3.0** |
| | convnet-8 | 10.87 | N/A | 10.56 | N/A | N/A | N/A | N/A | **2.85** |
| | convnet-16 | 10.93 | N/A | 10.55 | N/A | N/A | N/A | N/A | **2.64** |
| Bushy Eyebrows | convnet-1 | 7.34 | N/A | 6.85 | N/A | N/A | N/A | N/A | **2.38** |
| | convnet-2 | 7.24 | N/A | 6.19 | N/A | N/A | N/A | N/A | **1.77** |
| | convnet-4 | 7.18 | N/A | 7.07 | N/A | N/A | N/A | N/A | **2.38** |
| | convnet-8 | 6.96 | N/A | 6.86 | N/A | N/A | N/A | N/A | **1.77** |
| | convnet-16 | 6.79 | N/A | 6.65 | N/A | N/A | N/A | N/A | **1.69** |
| Chubby | convnet-1 | 3.01 | N/A | 2.7 | N/A | 2.28 | N/A | N/A | **0.73** |
| | convnet-2 | 2.99 | N/A | 2.78 | N/A | N/A | N/A | N/A | **0.85** |
| | convnet-4 | 2.82 | N/A | 2.77 | N/A | N/A | N/A | N/A | **0.83** |
| | convnet-8 | 2.47 | N/A | 2.49 | N/A | N/A | N/A | N/A | **0.74** |
| | convnet-16 | 2.36 | N/A | N/A | N/A | N/A | N/A | N/A | **0.66** |
| Double Chin | convnet-1 | 2.4 | N/A | 2.18 | N/A | 1.99 | N/A | N/A | **0.62** |
| | convnet-2 | 2.13 | N/A | N/A | N/A | 1.95 | N/A | N/A | **0.63** |
| | convnet-4 | 2.07 | N/A | 1.94 | N/A | N/A | N/A | N/A | **0.63** |
| | convnet-8 | 2.11 | N/A | 1.9 | N/A | N/A | N/A | N/A | **0.58** |
| | convnet-16 | 1.94 | N/A | 2.02 | N/A | N/A | N/A | N/A | **0.59** |
| Eyeglasses | convnet-1 | 2.52 | 2.0 | 2.09 | 1.98 | 1.87 | 1.8 | 1.76 | **0.9** |
| | convnet-2 | 2.44 | N/A | 2.25 | 2.01 | 1.92 | 1.83 | N/A | **0.84** |
| | convnet-4 | 2.33 | N/A | 1.97 | 2.0 | 1.87 | 1.79 | N/A | **0.81** |
| | convnet-8 | 2.34 | N/A | 2.24 | 2.2 | N/A | 1.81 | N/A | **1.08** |
| | convnet-16 | 2.25 | 2.01 | 1.93 | 2.05 | 1.82 | 1.75 | 1.93 | **0.79** |
| Goatee | convnet-1 | 3.55 | N/A | 3.38 | N/A | N/A | N/A | N/A | **1.28** |
| | convnet-2 | 3.46 | N/A | N/A | N/A | N/A | N/A | N/A | **1.7** |
| | convnet-4 | 3.56 | N/A | N/A | N/A | N/A | N/A | N/A | **1.56** |
| | convnet-8 | 3.27 | N/A | 3.07 | N/A | N/A | N/A | N/A | **0.95** |
| | convnet-16 | 3.23 | N/A | 3.09 | N/A | N/A | N/A | N/A | **0.96** |
| Gray Hair | convnet-1 | 2.62 | N/A | 2.41 | 2.04 | N/A | 2.05 | N/A | **0.85** |
| | convnet-2 | 2.57 | N/A | 2.39 | 2.16 | 2.3 | 2.04 | N/A | **0.87** |
| | convnet-4 | 2.49 | N/A | 2.45 | 2.2 | N/A | 2.12 | N/A | **0.82** |
| | convnet-8 | 2.5 | N/A | 2.37 | 2.29 | N/A | 2.2 | 2.32 | **0.85** |
| | convnet-16 | 2.4 | N/A | 2.22 | 2.01 | 2.16 | 1.98 | 2.14 | **0.77** |
| Heavy Makeup | convnet-1 | 11.2 | N/A | 9.59 | 8.79 | N/A | 7.8 | N/A | **3.04** |
| | convnet-2 | 10.72 | N/A | 9.53 | N/A | N/A | N/A | N/A | **3.92** |
| | convnet-4 | 10.53 | N/A | 9.35 | N/A | N/A | 8.12 | N/A | **3.24** |
| | convnet-8 | 10.67 | N/A | 9.93 | N/A | N/A | 8.18 | N/A | **3.17** |
| | convnet-16 | 10.96 | N/A | 9.76 | N/A | N/A | 8.07 | N/A | **3.4** |
| High Cheekbones | convnet-1 | 12.86 | N/A | 10.43 | N/A | N/A | 9.04 | N/A | **3.04** |
| | convnet-2 | 12.42 | N/A | 10.41 | N/A | N/A | N/A | N/A | **3.04** |
| | convnet-4 | 12.1 | N/A | 10.4 | N/A | N/A | N/A | N/A | **3.04** |
| | convnet-8 | 12.37 | N/A | 9.83 | N/A | N/A | 8.65 | N/A | **2.87** |
| | convnet-16 | 13.01 | N/A | 11.45 | N/A | N/A | N/A | N/A | **4.27** |

Table 32: Results for CelebA tasks under churn at cold accuracy metric across different sizes of convolutional networks with initial sample 10000. Part 2 of 4.

| Dataset | network | cold | warm | s-perturb | mixup | ls | co-dist | anchor | distill |
|---------|---------|------|------|-----------|-------|-----|---------|--------|---------|
| Male | convnet-1 | 8.42 | 6.21 | 6.31 | 6.2 | 6.18 | 5.79 | 5.5 | **3.08** |
| | convnet-2 | 8.11 | N/A | 6.39 | 6.43 | 6.14 | 6.03 | 5.51 | **3.27** |
| | convnet-4 | 7.65 | 6.08 | 6.15 | 6.03 | 6.03 | 5.79 | 5.37 | **3.18** |
| | convnet-8 | 8.04 | 6.52 | 6.61 | 6.34 | 6.38 | 6.17 | 5.67 | **3.09** |
| | convnet-16 | 8.06 | 6.45 | 6.6 | 6.5 | 6.49 | 6.25 | 5.83 | **3.21** |
| Mouth Slightly Open | convnet-1 | 12.43 | N/A | **10.22** | N/A | N/A | N/A | N/A | N/A |
| | convnet-2 | 12.16 | N/A | 9.61 | N/A | N/A | 7.94 | N/A | **2.86** |
| | convnet-4 | 12.54 | N/A | 10.41 | N/A | N/A | N/A | N/A | **2.82** |
| | convnet-8 | 12.31 | N/A | 9.84 | N/A | N/A | 7.82 | N/A | **3.36** |
| | convnet-16 | 14.93 | N/A | **10.6** | N/A | N/A | N/A | N/A | N/A |
| Mustache | convnet-1 | 1.94 | N/A | N/A | N/A | 1.49 | N/A | N/A | **0.51** |
| | convnet-2 | 1.86 | N/A | N/A | N/A | 1.49 | N/A | N/A | **0.52** |
| | convnet-4 | 2.09 | N/A | 1.88 | N/A | 1.83 | N/A | N/A | **0.58** |
| | convnet-8 | 1.98 | N/A | N/A | N/A | **1.56** | N/A | N/A | N/A |
| | convnet-16 | 1.64 | N/A | 1.56 | N/A | 1.33 | N/A | N/A | **0.49** |
| Narrow Eyes | convnet-1 | 4.11 | N/A | 3.43 | N/A | N/A | N/A | N/A | **0.94** |
| | convnet-2 | 3.58 | N/A | N/A | N/A | N/A | N/A | N/A | **0.97** |
| | convnet-4 | 3.14 | N/A | 2.63 | N/A | N/A | N/A | N/A | **0.69** |
| | convnet-8 | 2.0 | N/A | N/A | N/A | N/A | N/A | N/A | **0.44** |
| | convnet-16 | 1.06 | N/A | 0.9 | N/A | N/A | N/A | N/A | **0.18** |
| No Beard | convnet-1 | 8.44 | N/A | 6.59 | N/A | N/A | 6.04 | N/A | **2.22** |
| | convnet-2 | 8.27 | N/A | 7.63 | N/A | N/A | 6.26 | N/A | **2.27** |
| | convnet-4 | 7.97 | N/A | 6.93 | N/A | N/A | 6.18 | N/A | **2.36** |
| | convnet-8 | 8.0 | N/A | 7.26 | N/A | N/A | 6.26 | N/A | **2.31** |
| | convnet-16 | 8.34 | N/A | 7.29 | 7.2 | N/A | 6.2 | 6.79 | **2.25** |
| Oval Face | convnet-1 | 14.2 | N/A | 13.96 | N/A | N/A | N/A | N/A | **3.5** |
| | convnet-2 | 14.39 | N/A | 13.74 | N/A | N/A | N/A | N/A | **3.66** |
| | convnet-4 | **14.24** | N/A | N/A | N/A | N/A | N/A | N/A | N/A |
| | convnet-8 | 14.14 | N/A | 12.29 | N/A | N/A | N/A | N/A | **3.09** |
| | convnet-16 | 12.36 | N/A | 12.19 | N/A | N/A | N/A | N/A | **2.75** |
| Pale Skin | convnet-1 | 2.74 | N/A | 2.17 | 2.3 | 2.25 | 2.03 | N/A | **0.65** |
| | convnet-2 | 2.29 | N/A | 2.3 | N/A | N/A | N/A | N/A | **0.58** |
| | convnet-4 | 2.44 | N/A | 2.22 | N/A | 2.32 | 2.13 | N/A | **0.61** |
| | convnet-8 | 2.38 | N/A | 2.17 | 2.2 | N/A | 2.05 | N/A | **0.75** |
| | convnet-16 | 2.23 | N/A | 2.3 | N/A | N/A | 2.11 | N/A | **0.58** |
| Pointy Nose | convnet-1 | 15.58 | N/A | 15.39 | N/A | N/A | N/A | N/A | **3.7** |
| | convnet-2 | 15.4 | N/A | 15.14 | N/A | N/A | N/A | N/A | **3.68** |
| | convnet-4 | 15.23 | N/A | 13.25 | N/A | N/A | N/A | N/A | **3.44** |
| | convnet-8 | **13.53** | N/A | N/A | N/A | N/A | N/A | N/A | N/A |
| | convnet-16 | 11.39 | N/A | N/A | N/A | N/A | N/A | N/A | **2.86** |
| Receding Hairline | convnet-1 | 4.57 | N/A | 4.2 | N/A | N/A | N/A | N/A | **1.25** |
| | convnet-2 | 4.28 | N/A | 4.1 | N/A | N/A | N/A | N/A | **1.15** |
| | convnet-4 | 4.4 | N/A | 4.56 | N/A | N/A | N/A | N/A | **1.67** |
| | convnet-8 | 4.41 | N/A | 4.26 | N/A | N/A | N/A | N/A | **1.63** |
| | convnet-16 | 4.18 | N/A | N/A | N/A | N/A | N/A | N/A | **1.21** |
| Rosy Cheeks | convnet-1 | 4.15 | N/A | 3.84 | 3.23 | 3.69 | N/A | N/A | **1.23** |
| | convnet-2 | 4.49 | N/A | 4.26 | 3.66 | 3.88 | 3.61 | N/A | **1.23** |
| | convnet-4 | 4.28 | N/A | 4.0 | 3.37 | N/A | 3.46 | N/A | **1.16** |
| | convnet-8 | 3.86 | N/A | 3.71 | 3.31 | 3.44 | N/A | N/A | **1.05** |
| | convnet-16 | 3.83 | N/A | 3.65 | 3.16 | N/A | 3.38 | N/A | **0.94** |

Table 33: Results for CelebA tasks under churn at cold accuracy metric across different sizes of convolutional networks with initial sample 10000. Part 3 of 4.

| Dataset | network | cold | warm | s-perturb | mixup | ls | co-dist | anchor | distill |
|---|---|---|---|---|---|---|---|---|---|
| Sideburns | convnet-1 | 3.19 | N/A | N/A | 2.52 | N/A | 2.36 | N/A | **0.9** |
| | convnet-2 | 3.07 | N/A | 2.95 | N/A | N/A | N/A | N/A | **1.14** |
| | convnet-4 | 3.13 | N/A | 2.74 | 2.62 | N/A | 2.48 | N/A | **0.96** |
| | convnet-8 | 2.93 | N/A | 2.75 | 2.64 | N/A | 2.38 | N/A | **0.89** |
| | convnet-16 | 2.91 | N/A | 2.71 | 2.62 | N/A | 2.37 | N/A | **0.79** |
| Smiling | convnet-1 | 9.61 | N/A | 7.58 | 7.66 | N/A | 7.18 | N/A | **4.44** |
| | convnet-2 | 9.43 | N/A | 8.35 | N/A | N/A | 7.0 | N/A | **3.04** |
| | convnet-4 | 9.4 | N/A | 9.14 | N/A | N/A | 6.91 | N/A | **3.09** |
| | convnet-8 | 9.81 | N/A | 7.69 | 7.93 | N/A | 6.84 | 7.44 | **2.58** |
| | convnet-16 | 10.19 | 7.42 | 7.73 | 7.9 | 7.58 | 7.18 | 7.4 | **2.81** |
| Straight Hair | convnet-1 | 6.64 | N/A | 6.34 | N/A | N/A | N/A | N/A | **1.58** |
| | convnet-2 | 6.96 | N/A | N/A | N/A | N/A | N/A | N/A | **1.62** |
| | convnet-4 | **7.33** | N/A | N/A | N/A | N/A | N/A | N/A | N/A |
| | convnet-8 | 6.47 | N/A | N/A | N/A | N/A | N/A | N/A | **1.54** |
| | convnet-16 | 5.95 | N/A | N/A | N/A | N/A | N/A | N/A | **1.24** |
| Wavy Hair | convnet-1 | **16.13** | N/A | N/A | N/A | N/A | N/A | N/A | N/A |
| | convnet-2 | 16.31 | N/A | N/A | N/A | N/A | N/A | N/A | **5.63** |
| | convnet-4 | 15.8 | N/A | 14.23 | 14.27 | N/A | 13.68 | N/A | **3.97** |
| | convnet-8 | 15.76 | N/A | N/A | N/A | N/A | N/A | N/A | **3.84** |
| | convnet-16 | 15.25 | N/A | 13.89 | N/A | N/A | N/A | N/A | **3.81** |
| Wearing Earrings | convnet-1 | 12.37 | N/A | 10.47 | 9.11 | N/A | N/A | N/A | **2.94** |
| | convnet-2 | 12.09 | N/A | 11.23 | N/A | N/A | N/A | N/A | **3.23** |
| | convnet-4 | 12.05 | N/A | 10.72 | 9.95 | N/A | N/A | N/A | **3.37** |
| | convnet-8 | 11.55 | N/A | 10.6 | 9.83 | N/A | 9.68 | N/A | **3.35** |
| | convnet-16 | 11.33 | N/A | 10.42 | 9.61 | N/A | 9.44 | N/A | **2.95** |
| Wearing Hat | convnet-1 | 1.9 | N/A | 1.58 | 1.61 | 1.52 | 1.43 | 1.39 | **0.64** |
| | convnet-2 | 1.95 | N/A | 1.81 | 1.82 | 1.71 | 1.66 | 1.58 | **0.73** |
| | convnet-4 | 1.86 | N/A | N/A | N/A | N/A | 1.53 | N/A | **0.9** |
| | convnet-8 | 1.86 | N/A | N/A | N/A | N/A | 1.54 | N/A | **0.85** |
| | convnet-16 | 1.88 | 1.66 | 1.77 | 1.78 | 1.6 | 1.6 | 1.59 | **0.71** |
| Wearing Lipstick | convnet-1 | 9.1 | N/A | 7.6 | 7.14 | N/A | 6.47 | N/A | **3.29** |
| | convnet-2 | 9.07 | N/A | 7.96 | 7.55 | N/A | 6.77 | N/A | **2.83** |
| | convnet-4 | 8.46 | N/A | N/A | 7.54 | N/A | 6.84 | N/A | **2.86** |
| | convnet-8 | 8.67 | N/A | N/A | N/A | N/A | 6.8 | N/A | **2.93** |
| | convnet-16 | 9.18 | N/A | 7.83 | N/A | N/A | 6.97 | N/A | **2.99** |
| Wearing Necklace | convnet-1 | 0.94 | N/A | **0.75** | N/A | N/A | N/A | N/A | N/A |
| | convnet-2 | 1.64 | N/A | **1.45** | N/A | N/A | N/A | N/A | N/A |
| | convnet-4 | 1.36 | N/A | 1.14 | N/A | N/A | N/A | N/A | **0.41** |
| | convnet-8 | 1.51 | N/A | 1.45 | N/A | N/A | N/A | N/A | **0.41** |
| | convnet-16 | 0.99 | N/A | N/A | N/A | N/A | N/A | N/A | **0.25** |
| Wearing Necktie | convnet-1 | 3.34 | N/A | 3.03 | 2.86 | N/A | 2.8 | N/A | **0.98** |
| | convnet-2 | 3.04 | N/A | N/A | N/A | N/A | N/A | N/A | **0.97** |
| | convnet-4 | 3.43 | N/A | 2.99 | 2.97 | 3.02 | 2.85 | N/A | **1.05** |
| | convnet-8 | 3.28 | N/A | 3.06 | 3.07 | N/A | 2.75 | N/A | **1.04** |
| | convnet-16 | 3.26 | N/A | N/A | N/A | N/A | 2.87 | N/A | **1.53** |
| Young | convnet-1 | 10.65 | N/A | 9.59 | 8.32 | N/A | N/A | N/A | **2.52** |
| | convnet-2 | 10.51 | N/A | 9.54 | N/A | N/A | N/A | N/A | **2.9** |
| | convnet-4 | 10.08 | N/A | 9.21 | N/A | N/A | N/A | N/A | **2.81** |
| | convnet-8 | 10.02 | N/A | 9.42 | N/A | N/A | N/A | N/A | **2.79** |
| | convnet-16 | 9.67 | N/A | N/A | N/A | N/A | 8.3 | N/A | **2.63** |

Table 34: Results for CelebA tasks under churn at cold accuracy metric across different sizes of convolutional networks with initial sample 10000. Part 4 of 4.

| | convnet-1 | | convnet-2 | | convnet-4 | | convnet-8 | | convnet-16 | |
|---|---|---|---|---|---|---|---|---|---|---|
| Dataset | Error | Churn | Error | Churn | Error | Churn | Error | Churn | Error | Churn |
| 5 o Clock Shadow | 0.67 | 0.77 | 0.68 | 0.78 | 0.65 | 0.75 | 0.67 | 0.79 | 0.7 | 0.82 |
| Arched Eyebrows | 0.59 | 0.75 | 0.61 | 0.79 | 0.61 | 0.78 | 0.61 | 0.77 | 0.61 | 0.79 |
| Attractive | 0.14 | 0.36 | 0.14 | 0.35 | 0.15 | 0.38 | 0.14 | 0.37 | 0.14 | 0.4 |
| Bags Under Eyes | 0.67 | 0.92 | 0.69 | 0.94 | 0.69 | 0.95 | 0.69 | 0.94 | 0.68 | 0.95 |
| Bald | 0.33 | 0.35 | 0.36 | 0.39 | 0.39 | 0.42 | 0.34 | 0.37 | 0.37 | 0.4 |
| Bangs | 0.66 | 0.7 | 0.67 | 0.73 | 0.64 | 0.68 | 0.65 | 0.71 | 0.68 | 0.73 |
| Big Lips | 0.67 | 1.18 | 0.65 | 1.19 | 0.68 | 1.23 | 0.67 | 1.27 | 0.67 | 1.31 |
| Big Nose | 0.64 | 0.9 | 0.65 | 0.92 | 0.65 | 0.9 | 0.65 | 0.94 | 0.65 | 0.94 |
| Black Hair | 0.6 | 0.67 | 0.61 | 0.69 | 0.61 | 0.69 | 0.59 | 0.68 | 0.6 | 0.67 |
| Blond Hair | 0.72 | 0.77 | 0.7 | 0.74 | 0.7 | 0.74 | 0.68 | 0.71 | 0.69 | 0.72 |
| Blurry | 0.54 | 0.61 | 0.56 | 0.64 | 0.55 | 0.62 | 0.51 | 0.58 | 0.54 | 0.6 |
| Brown Hair | 0.65 | 0.82 | 0.66 | 0.81 | 0.67 | 0.83 | 0.66 | 0.82 | 0.67 | 0.84 |
| Bushy Eyebrows | 0.67 | 0.82 | 0.69 | 0.84 | 0.7 | 0.84 | 0.68 | 0.84 | 0.73 | 0.9 |
| Chubby | 0.57 | 0.64 | 0.58 | 0.65 | 0.54 | 0.6 | 0.58 | 0.65 | 0.56 | 0.64 |
| Double Chin | 0.53 | 0.58 | 0.53 | 0.58 | 0.53 | 0.58 | 0.55 | 0.61 | 0.49 | 0.54 |
| Eyeglasses | 0.56 | 0.58 | 0.57 | 0.59 | 0.53 | 0.55 | 0.54 | 0.57 | 0.55 | 0.58 |
| Goatee | 0.59 | 0.63 | 0.56 | 0.6 | 0.57 | 0.61 | 0.57 | 0.61 | 0.58 | 0.63 |
| Gray Hair | 0.48 | 0.51 | 0.52 | 0.54 | 0.54 | 0.57 | 0.52 | 0.55 | 0.5 | 0.53 |
| Heavy Makeup | 0.36 | 0.39 | 0.36 | 0.38 | 0.37 | 0.37 | 0.36 | 0.4 | 0.37 | 0.38 |
| High Cheekbones | 0.21 | 0.32 | 0.21 | 0.35 | 0.22 | 0.34 | 0.21 | 0.33 | 0.23 | 0.49 |
| Male | 0.26 | 0.29 | 0.26 | 0.3 | 0.26 | 0.29 | 0.26 | 0.3 | 0.27 | 0.3 |
| Mouth Slightly Open | 0.16 | 0.28 | 0.15 | 0.29 | 0.16 | 0.29 | 0.15 | 0.28 | 0.57 | 0.77 |
| Mustache | 0.51 | 0.56 | 0.45 | 0.49 | 0.47 | 0.52 | 0.45 | 0.51 | 0.45 | 0.5 |
| Narrow Eyes | 0.71 | 0.9 | 0.71 | 0.9 | 0.72 | 0.94 | 0.69 | 0.93 | 0.71 | 0.96 |
| No Beard | 0.68 | 0.77 | 0.67 | 0.76 | 0.66 | 0.74 | 0.66 | 0.74 | 0.66 | 0.77 |
| Oval Face | 0.6 | 1.05 | 0.61 | 1.03 | 0.61 | 1.06 | 0.61 | 1.12 | 0.61 | 1.2 |
| Pale Skin | 0.55 | 0.6 | 0.48 | 0.55 | 0.54 | 0.61 | 0.56 | 0.64 | 0.51 | 0.58 |
| Pointy Nose | 0.61 | 1.09 | 0.61 | 1.05 | 0.62 | 1.11 | 0.61 | 1.11 | 0.63 | 1.26 |
| Receding Hairline | 0.66 | 0.76 | 0.63 | 0.72 | 0.64 | 0.73 | 0.62 | 0.72 | 0.65 | 0.73 |
| Rosy Cheeks | 0.55 | 0.61 | 0.58 | 0.65 | 0.63 | 0.7 | 0.6 | 0.67 | 0.58 | 0.67 |
| Sideburns | 0.58 | 0.62 | 0.56 | 0.61 | 0.55 | 0.59 | 0.52 | 0.57 | 0.53 | 0.58 |
| Smiling | 0.13 | 0.25 | 0.14 | 0.23 | 0.13 | 0.23 | 0.14 | 0.26 | 0.16 | 0.49 |
| Straight Hair | 0.71 | 1.14 | 0.71 | 1.13 | 0.72 | 1.14 | 0.72 | 1.16 | 0.72 | 1.19 |
| Wavy Hair | 0.52 | 0.69 | 0.52 | 0.68 | 0.52 | 0.69 | 0.52 | 0.73 | 0.51 | 0.71 |
| Wearing Earrings | 0.69 | 0.89 | 0.69 | 0.89 | 0.69 | 0.9 | 0.7 | 0.91 | 0.71 | 0.94 |
| Wearing Hat | 0.5 | 0.51 | 0.56 | 0.6 | 0.54 | 0.56 | 0.51 | 0.53 | 0.52 | 0.54 |
| Wearing Lipstick | 0.17 | 0.21 | 0.16 | 0.21 | 0.16 | 0.21 | 0.16 | 0.2 | 0.16 | 0.21 |
| Wearing Necklace | 0.7 | 0.97 | 0.73 | 1.0 | 0.73 | 0.99 | 0.72 | 0.98 | 0.73 | 1.0 |
| Wearing Necktie | 0.62 | 0.65 | 0.58 | 0.61 | 0.59 | 0.62 | 0.59 | 0.61 | 0.61 | 0.7 |
| Young | 0.65 | 0.87 | 0.66 | 0.88 | 0.66 | 0.88 | 0.67 | 0.9 | 0.66 | 0.9 |

Table 35: CelebA Error Bands with initial sample 10000: Average standard errors for error and churn across baselines for each dataset and network across 100 runs.

| Dataset | Architecture | cold | warm | sperturb | mixup | ls | codist | anchor | distill (ours) |
|---------|-------------|------|------|----------|-------|------|--------|--------|----------------|
| cifar10 | ResNet-50 | 49.85 | 46.4 | 46.6 | 46.45 | 44.4 | N/A | 45.35 | **42.85** |
| cifar100 | ResNet-50 | 89.05 | 85.9 | 83.7 | 85.05 | 81.95 | N/A | 84.45 | **72.05** |
| cifar10 | ResNet-101 | 50.65 | 48.2 | 46.25 | 48.45 | N/A | N/A | 46.8 | **43.85** |
| cifar100 | ResNet-101 | 88.9 | 87.8 | 86.75 | 87.75 | 86.4 | N/A | 86.25 | **77.75** |
| cifar10 | ResNet-152 | 46.9 | 47.7 | 47.7 | 47.1 | 50.0 | N/A | 46.4 | **43.4** |
| cifar100 | ResNet-152 | 89.4 | 86.9 | 85.9 | 86.3 | 84.1 | N/A | 83.8 | **78.6** |

Table 36: Results for CIFAR10 and CIFAR100 under churn at cold accuracy metric across ResNet-50, ResNet-101 and ResNet-152. Initial sample size and batch size is fixed at 1000..

| Initial Sample | network | cold | warm | s-perturb | mixup | ls | co-dist | anchor | distill |
|----------------|---------|------|------|-----------|-------|------|---------|--------|---------|
| | transformer-1 | 38.34 | 36.72 | 37.36 | N/A | **36.35** | N/A | N/A | N/A |
| | transformer-2 | 37.33 | N/A | N/A | N/A | **35.99** | N/A | N/A | N/A |
| 100 | transformer-4 | 40.78 | N/A | 39.48 | N/A | 38.07 | N/A | N/A | **34.38** |
| | transformer-8 | 44.32 | N/A | 45.67 | N/A | 42.65 | N/A | 43.9 | **38.96** |
| | transformer-16 | 50.87 | 46.87 | 46.99 | N/A | 47.27 | 45.49 | 45.38 | **40.05** |
| | transformer-1 | 15.69 | N/A | 15.31 | N/A | N/A | 13.41 | 10.33 | **7.78** |
| | transformer-2 | 16.62 | 14.35 | 15.1 | N/A | 13.0 | 15.1 | 13.11 | **7.0** |
| 1000 | transformer-4 | 18.79 | N/A | N/A | N/A | N/A | N/A | 16.72 | **14.24** |
| | transformer-8 | 20.11 | N/A | 19.79 | N/A | 17.56 | 18.3 | 17.51 | **12.93** |
| | transformer-16 | 24.06 | N/A | 22.4 | N/A | 20.27 | 21.36 | 20.43 | **15.75** |
| | transformer-1 | 8.87 | N/A | N/A | N/A | N/A | N/A | 8.48 | **6.4** |
| | transformer-2 | 9.0 | N/A | N/A | N/A | N/A | N/A | N/A | **6.83** |
| 10000 | transformer-4 | 9.24 | N/A | N/A | N/A | N/A | N/A | N/A | **6.55** |
| | transformer-8 | 9.95 | N/A | N/A | N/A | 6.74 | N/A | N/A | **6.74** |
| | transformer-16 | **12.73** | N/A | N/A | N/A | N/A | N/A | N/A | N/A |

Table 37: Results for IMDB under churn at cold accuracy metric across different sizes of transformer networks and initial sample sizes. Batch size is fixed at 1000.

| | transformer-1 | | transformer-2 | | transformer-4 | | transformer-8 | | transformer-16 | |
|----------------|------|-------|------|-------|------|-------|------|-------|------|-------|
| Iniitial Sample | Error | Churn | Error | Churn | Error | Churn | Error | Churn | Error | Churn |
| 100 | 0.51 | 1.31 | 0.53 | 1.44 | 0.57 | 1.36 | 0.62 | 1.74 | 0.71 | 2.27 |
| 1000 | 0.45 | 0.61 | 0.48 | 0.67 | 0.52 | 0.9 | 0.61 | 1.0 | 0.74 | 1.57 |
| 10000 | 0.18 | 0.33 | 0.19 | 0.31 | 0.23 | 0.32 | 0.3 | 0.4 | 0.84 | 1.56 |

Table 38: IMDB Error Bands: Mean standard errors for error and churn across baselines for each dataset and network across 100 runs.

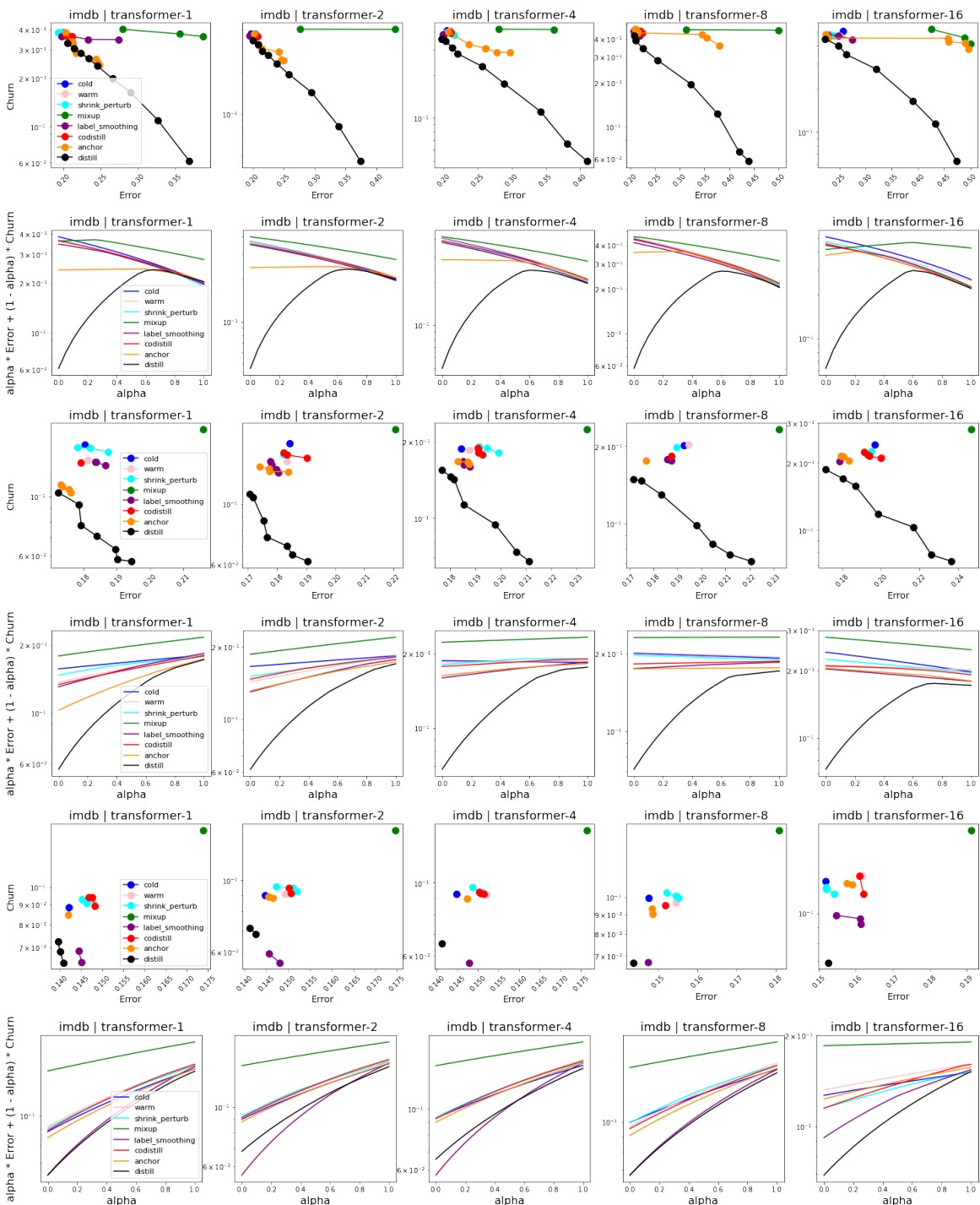

Figure 4: IMDB dataset with transformer. Pareto frontier for each baseline and costs of each method, where the cost is a convex combination between the error and the churn, as we vary the weight between churn and accuracy. **Top two**: Initial batch size 100. **Middle**: Initial batch size 1000. **Bottom**: Initial batch size 10000.

