# OpenReview forum: "Churn Reduction via Distillation"
_ICLR.cc/2022/Conference — ICLR 2022 Spotlight_

### Official Review · Reviewer_pZBb · 2021-10-31

**Correctness:** 4
**Technical Novelty And Significance:** 4
**Empirical Novelty And Significance:** 4
**Recommendation:** 8
**Confidence:** 2

**Main Review:**

Strengths
- The authors provide theoretical guarantees to justify the use of their algorithm compared to comparable approaches
- The authors provide empirical justifications of the proposed algorithm using many datasets and baselines

Weaknesses
-	I was not able to find a major weakness in this paper

Typos
-	5.2 : « performs the best » or « performs well » probably


**Summary Of The Paper:**

This paper explores the relationship between the low-churn problem and distillation. The authors show that there is an equivalence between those two methods and that distillation performs particularly well on low-churn dataset tasks. The authors propose a novel churn reduction algorithm based on distillation which involves the training of a classifier by minimizing a distilled loss and solving a convex program. The authors  provide theoretical guarantees for the proposed algorithm and explain the advantages of the proposed approach compared to the anchor loss, another churn-reduction method.

The authors validate empirically their approach on 12 OpenML datasets, 10 MNIST variants, CIFAR10, CIFAR100 and IMDB.


**Summary Of The Review:**

This paper provides strong theoretical and empirical results regarding the effectiveness of the proposed distillation based algorithm for the churn reduction problem.

---

> ### Author Response · Authors · 2021-11-23
> **Response to Reviewer pZBb**
>
> We thank the reviewer for appreciating our theoretical and empirical results.

---

> > ### Comment · Reviewer_pZBb · 2021-11-28
> > **Response Read**
> >
> > Thanks for the response. I am satisfied with the authors' response and I keep my score the same (8).

---

### Official Review · Reviewer_TJ4g · 2021-11-05

**Correctness:** 3
**Technical Novelty And Significance:** 3
**Empirical Novelty And Significance:** 4
**Recommendation:** 8
**Confidence:** 4

**Details Of Ethics Concerns:**

While I don't see an ethical concern for this work on its own, it may be of interest to the authors if they can think about how their distillation process can be adjusted if the base model was somehow unfair

**Main Review:**

Some of the strongest aspect of the paper is as follows:

- Distillation is a training process that has been applied to many different settings with remarkable success. The current paper introduces this technique as a simpler solution for a problem of interest, especially in production settings.
- The results seem to support the importance of their method. The authors also conducted detailed analysis of multiple settings that makes the claims stronger
- The theoretical underpinnings presented in this paper by essentially casting distillation as a simpler dual problem for the more computationally complex constrained optimization primal is well appreciated. The following propositions are also of interest and the bounds can lead to newer theories and/or better usage of such methods in practice
- Overall the paper is well presented, especially the methods section.

The paper can be improved upon by addressing a few points as below:
- The authors presented the claim that their process corresponds to `a`  value of $\lambda$ that constrains the churn in the primal setup. Algorithm 1 also points to a last step where the corresponding primal solution is found from the dual solutions. However, while possibly trivial, the methods to use this distillation process for an ``acceptable churn`` is missing. I.e how would one go about using this method for re-training if they a certain amount of churn can be tolerated. From the introduction and motivation presented in the paper, it seems such a use case maybe of interest and the paper can perhaps include this setting
- From a presentation perspective, the confidence bounds in the appendix could be brought up in the main paper
- Another minor aspect of the presentation, the problem of churn and its impact on problem settings should be presented with citations in the introduction section. There is some citation on these in the related work however the problem introduction also needs some justification e.g. literature on churn being hurtful in production settings etc.

Some minor points that can be addressed:
- The traces to arrive at Proposition 1 may be included in the main paper. Its one of the most interesting and foundational aspects of the paper and could be better presented to make the main paper self sufficient
- In section 3, $y$ is used without definition. Also, `low predictive churn` is used without defining what does low mean

**Summary Of The Paper:**

In real-life applications of predictive models, predictions often form a step of the process. Changes in the predictive model often need to be validated end to end using methods such as A/B tests before they can be used in production. Thus for certain class of problems, there is a need to control the churn or the difference in the predictive model due to retraining of the model towards a more robust pipeline. The authors present an approach to control this churn via a simple distillation method. They also show that their distillation method, under certain assumptions, is equivalent to a constrained optimization problem that explicitly constrains the "churn" without the added complexity of constrained optimization. They validated their method by conducting experiments on a number of baselines on a wide range of datasets and model architectures.

**Summary Of The Review:**

This is a very interesting paper that adds both empirical and theoretical findings to the body of literature. The theoretical underpinnings of distillation as a constrained optimization applicable for churn reduction sheds new light on a very popular method and can lead to new applications. The presented results also support the efficacy of the methods. Overall this paper is a good example of an ICLR paper. I would encourage the authors to address the presentation points mentioned in the main review for a more impactful submission.

---

> ### Author Response · Authors · 2021-11-23
> **Response to Reviewer TJ4g**
>
> We thank the reviewer for the thoughtful and detailed feedback and for appreciating the theoretical and empirical insights.
>
> Post-processing in Algorithm 1: The algorithm learns one classifier for each value of {\lambda_1, …, \lambda_L}, and finds a convex combination of the classifiers that solves the primal problem (this is a simple convex program in L variables, which we’ll be happy to explicitly state in the appendix). As we note in the last paragraph of Section 3, in practice, we avoid this post-processing step, and instead pick from the L classifiers learned, the one that achieves the least empirical risk while satisfying the churn constraint. That is, we evaluate the classification error and the churn for each of the L classifiers on the test set, and from among those that are within the acceptable error limit, we pick the one with the least churn. In our experiments, we try out \lambda-values from {0.1, 0.2, …, 0.9}.
>
> Confidence bounds: Nice suggestion! We’ll move this to the main text.
>
> Citations in introduction: Yes, we’ll include some citations of related work in the introduction.
>
> Proposition 1: We’ll be happy to provide a proof sketch in the main text.
>
> “constraining it to have low predictive churn”: We’ll replace this with “constraining its churn to be within an allowable limit”.

---

### Official Review · Reviewer_nqfu · 2021-11-08

**Correctness:** 3
**Technical Novelty And Significance:** 2
**Empirical Novelty And Significance:** 2
**Recommendation:** 5
**Confidence:** 4

**Main Review:**

=================================================================

[Main Strengths]

This paper's main strength is that the authors' lengthy proof of the performance guarantee (not sure how tight the bound is, though), and the key implementation codes are provided in the Appendix.

=================================================================

[Main Weaknesses]

The primary limitation of this study is that, while the underlying problem framework is intriguing, it does not appear to make a significant impact, much like the various flavors of domain adaptations. I encourage authors to make additional improvements, particularly for the experiments described in Section 5, which is to show that how the distillation handles the distribution (or domain) shift issues, which will be highly intriguing and will definitely necessitate more detailed experiment settings.

=================================================================

[Technical Comments]

1) The title is a little misleading, as churn prediction often means “predicting which clients are most likely to cancel a subscription".
2) Another essential related works that I recommend the authors cite is unsupervised domain adaptation with practical application (for example, https://arxiv.org/abs/1905.02530).

=================================================================

**Summary Of The Paper:**

This paper examines an interesting problem, so-called churn prediction provides some nice proofs, builds distillation-based algorithms, and demonstrated the performance to multi-class problems in experiments.


**Summary Of The Review:**

The use of distillation for churn prediction is intriguing, but it does not appear to make a major difference, similar to the many flavors of domain adaptation.

---

> ### Author Response · Authors · 2021-11-23
> **Response to Reviewer nqfu**
>
> We thank the reviewer for appreciating the theoretical contributions and their feedback.
>
> **Connection to domain adaptation**: We assert that the connection between domain adaptations and our problem of churn reduction is loose. The former involves learning a model for a target distribution given data from the source distribution and (usually) limited information from the target distribution. The latter involves learning a model on a given dataset in such a way that the predictions don’t differ by much from a base model and there does not need to be a notion of source vs target data in our setting. Furthermore, it may not even be useful to try to match a model’s predictions if it was trained for a different task (as is often the case in domain adaptation).
>
> The reviewer brings up a good point that if the base model were trained on a source dataset different from the target dataset that a new model is trained on, then it would be interesting to investigate the domain adaptation from training the new model using the base model’s predictions. However, in this paper, we don’t assume that there is such a shift between the data distribution the base model vs the new model is trained on– in fact, this paper along with previous papers show that even training on the same data using the same architecture and training procedure can lead to vastly different models (even if all of the models attain high accuracy). We thank the reviewer for bringing this up and we will clarify this further in the paper.
>
> **Impact**: We would like to emphasize that churn is a highly relevant problem in practical machine learning systems where models are continuously launched and replace previous models– in such situations, unnecessary differences in predictions between the new model and the previous model can have undesirable downstream consequences for the end user. We believe that our distillation-based approach offers a simple and effective solution to reducing churn in real-world production models. Moreover, through experiments on 20+ benchmark datasets, we’ve shown statistically significant and substantial gains of our distillation-based approach over several prior baselines, and this we believe points to the potential impact our proposal can have.
>
> **Title**: We are using the notion of churn defined in previous works e.g. [Fard et al. 2016] and [Cotter et. al. 2019]. The reviewer is right that “churn” also refers to various “customer churn prediction” datasets that can be found on e.g. Kaggle and OpenML. We’ll be happy to clarify this in the paper.
>
> **Unsupervised domain adaptation**: We thank the reviewer for bringing up this related work in unsupervised domain adaptation– this may be relevant in situations where there is a distinction between the data distributions the base model and the new model are trained on. We agree that this could add an additional source of “churn” between the base model and the new model; however we stress that in such settings, one may be better off re-training the model on the target task (or applying standard domain adaptation strategies) rather than forcing it to match a base model trained on a different task.

---

### Decision · Program_Chairs · 2022-01-20

**Decision:**

Accept (Spotlight)

**Comment:**

The paper introduces a procedure to control the churn (i.e. differences in the predictive model due o retraining) using distillation.

This is a strong paper, with novel technique which is clearly presented, and is backed by sound theory. The experimental results were also deemed extremely convincing by reviewers TJ4g and pZBb.

Reviewer nqfu raised a question about the similarities between churn reduction and domain adaptation. The authors have addressed this by pointing out similarities to their work but also noting that, in the settings mentioned by the reviewer, alternative approaches such as completely retraining the model might be more appropriate. This part of the rebuttal is convincing.

Reviewer TJ4g has pointed out several points of improvement, to which the authors have responded adequately.

All in all, this paper is ready for and deserving of acceptance.